# A cell state-specific metabolic vulnerability to GPX4-dependent ferroptosis in glioblastoma

Matei A Banu [1,12], Athanassios Dovas [2,12], Michael G Argenziano [1,12], Wenting Zhao [3,12],
Colin P Sperring[1], Henar Cuervo Grajal [4], Zhouzerui Liu[3], Dominique MO Higgins[5], Misha Amini[1],
Brianna Pereira[2], Ling F Ye[6], Aayushi Mahajan[1,2], Nelson Humala[1,2], Julia L Furnari[1,2], Pavan S Upadhyayula[1],
Fereshteh Zandkarimi[7], Trang TT Nguyen [2], Damian Teasley[1], Peter B Wu[8], Li Hai[9], Charles Karan[9],
Tyrone Dowdy[10], Aida Razavilar[2], Markus D Siegelin [2], Jan Kitajewski[11], Mioara Larion [10],
Jeffrey N Bruce [1], Brent R Stockwell [7], Peter A Sims [3✉] & Peter Canoll [1,2✉]

## Abstract

**Glioma cells hijack developmental programs to control cell state. Here, we uncover a glioma cell state-specific metabolic liability that can be therapeutically targeted. To model cell conditions at brain tumor inception, we generated genetically engineered murine gliomas, with deletion of p53 alone (p53) or with constitutively active Notch signaling (N1IC), a pathway critical in controlling astrocyte differentiation during brain development. N1IC tumors harbored quiescent astrocyte-like transformed cell populations while p53 tumors were predominantly comprised of proliferating progenitor-like cell states. Further, N1IC transformed cells exhibited increased mitochondrial lipid peroxidation, high ROS production and depletion of reduced glutathione. This altered mitochondrial phenotype rendered the astrocyte-like, quiescent populations more sensitive to pharmacologic or genetic inhibition of the lipid hydroperoxidase GPX4 and induction of ferroptosis. Treatment of patient-derived early-passage cell lines and glioma slice cultures generated from surgical samples with a GPX4 inhibitor induced selective depletion of quiescent astrocyte-like glioma cell populations with similar metabolic profiles. Collectively, these findings reveal a specific therapeutic vulnerability to ferroptosis linked to mitochondrial redox imbalance in a subpopulation of quiescent astrocyte-like glioma cells resistant to standard forms of treatment.**

**Keywords** Glioma; Astrocytic; Quiescent; Mitochondrial-metabolism; Ferroptosis
**Subject Categories** Autophagy & Cell Death; Cancer; Metabolism

## Introduction

Intratumoral heterogeneity remains a central therapeutic hurdle in glioblastoma (GBM) (Neftel et al, 2019; Wang et al, 2022; Yuan et al, 2018; Zhao et al, 2021). Previous studies have identified diverse tumor cell states in high grade gliomas (Johnson et al, 2021; Liu et al, 2022; Neftel et al, 2019; Yuan et al, 2018). These glioma states differ in their resemblance to neural or glial lineages as well as their proliferative status (Neftel et al, 2019; Xie et al, 2022). To drive a particular state, glioma cells hijack specific neurodevelopmental transcriptional programs and master regulators (Neftel et al, 2019). We recently discovered that glioma states also differ in their therapeutic vulnerabilities, with proliferative populations demonstrating sensitivity to mitotic poisons (Zhao et al, 2021). These findings are consistent with clinical experience showing that cycling cells are effectively targeted by standard chemotherapy and radiation (Barthel et al, 2019; Liau et al, 2017; Spinazzi et al, 2022). In contrast, quiescent populations with mesenchymal or astrocytic features are relatively insensitive to standard forms of treatment and, while abundant in primary GBM, are even more abundant in recurrent GBM (Artegiani et al, 2017; Chen et al, 2012; Wang et al, 2022; Xie et al, 2022). Importantly, a subset of these persister tumor cells re-enter the cell cycle even under cytotoxic pressure and repopulate the proliferating cell pool (Oren et al, 2021; Xie et al, 2022). Quiescent tumor cells are therefore the likely culprit for resistance and tumor recurrence in GBM following standard of care treatment (Couturier et al, 2020; Hoang-Minh et al, 2018; Xie et al, 2022). There is now strong evidence that astrocyte-like glioma cells give rise to mesenchymal populations (Schiffman et al, 2023) which are associated with an aggressive phenotype, recurrence, and poor survival. In our previous studies with both bulk (Gill et al, 2014) and scRNA-seq (Yuan et al, 2018),

[1]Department of Neurological Surgery, Columbia University Irving Medical Center, New York, NY, USA. [2]Department of Pathology and Cell Biology, Columbia University Irving Medical Center, New York, NY, USA. [3]Department of System Biology, Columbia University Irving Medical Center, New York, NY, USA. [4]Spanish National Center for Cardiovascular Research, Madrid, Spain. [5]Department of Neurological Surgery, University of North Carolina School of Medicine, Chapel Hill, NC, USA. [6]Department of Medicine, Columbia University Irving Medical Center, New York, NY, USA. [7]Department of Biological Sciences, Department of Chemistry and Herbert Irving Comprehensive Cancer Center, Columbia University, New York, NY, USA. [8]Department of Neurological Surgery, UCLA Geffen School of Medicine, Los Angeles, CA, USA. [9]Sulzberger Columbia Genome Center, Columbia University, New York, NY, USA. [10]NCI Neuro-oncology Branch, Bethesda, MD, USA. [11]University of Illinois Cancer Center, Department of Physiology and Biophysics, University of Illinois Chicago, Chicago, IL, USA. [12]These authors contributed equally: Matei A Banu, Athanassios Dovas, Michael G Argenziano, Wenting Zhao.
✉E-mail: pas2182@cumc.columbia.edu; pc561@cumc.columbia.edu

we observed that the mesenchymal phenotype was strongly associated with recurrent tumors and that the principal remaining glioma cell type with neural lineage resemblance was astrocyte-like. Furthermore, more recent studies using multiplexed single-cell lineage tracing have phylogenetically associated the astrocyte-like and mesenchymal states (Schiffman et al, 2023; Wang et al, 2019). Thus, targeting specific vulnerabilities in the quiescent astrocyte-like population can potentially delay tumor relapse and lead to durable therapeutic responses (Hangauer et al, 2017; Xie et al, 2022). Cell state-specific druggable targets remain, however, an unmet need in GBM.

In the CNS, cell identity and function are tightly linked to metabolism. During neural development, activation of quiescent progenitors is followed by a rapid shift in mitochondrial metabolism (Llorens-Bobadilla et al, 2015). Mitochondrial redox activity and bioenergetics regulate cell fate decisions during neurogenesis and tumor initiation (Bonnay et al, 2020; Ojha et al, 2022). Astrocytic mitochondrial ROS play key roles in redox balance and function in the brain microenvironment (Vicente-Gutierrez et al, 2019). Furthermore, a recent study has identified mitochondrial transfer from tumor-associated astrocytes as a mechanism to regulate proliferation in GBM (Watson et al, 2023). In contrast, blockage of mitochondrial oxidative phosphorylation stalls glioma cells in quiescence through impaired NAD+ regeneration (Bonnay et al, 2020). In gliomas, energetic stress constrains evolutionary trajectories through activation of neurodevelopmental regulatory switches (Barthel et al, 2019). Furthermore, along with mesenchymal transformation, metabolic reprogramming is a key feature in recurrent GBM (Garofano et al, 2021). Long-term exposure of glioma cells to cytotoxic therapies such as temozolomide induces a quiescent cell state with a switch towards mitochondrial oxidative phosphorylation (Rabe et al, 2020). Such metabolic versatility leads to aggressive tumor behavior and poor prognosis (Garofano et al, 2021). Consequently, tumor cell subpopulations with specific metabolic affinities and potentially unique therapeutic vulnerabilities have been recently identified in glioma (Al-Dalahmah et al, 2023; Garofano et al, 2021; Hoang-Minh et al, 2018; Rusu et al, 2019) including subpopulations of glioma cells responsive to electron transport chain (ETC) inhibitors (Garofano et al, 2021). Mitochondria are dynamic organelles capable of functional remodeling based on cellular cues. Intriguingly, developmental transcriptional programs have been found to modulate electron flow through complex I in glioma cells, further linking cell state to mitochondrial phenotype (Ojha et al, 2022). Thus, we hypothesized that individual cell states depend on highly specialized metabolic programs at the mitochondrial level which could reveal novel state-specific druggable vulnerabilities.

There are significant challenges in modeling the astrocyte-like glioma cell state due to quiescence and plasticity, outcompeted by proliferative populations, or undergoing mesenchymal drift in models that require expansion and propagation of tumor cells (LeBlanc et al, 2022; Pine et al, 2020). Therefore, to explore the metabolic dependencies and therapeutic vulnerabilities of glioma states we used several different models, ranging from genetically engineered glioma models to patient derived cell lines and explants. First, we used a p53-deleted, PDGF-B overexpressing genetic murine glioma model with or without Notch activation as a model system. Notch signaling is a master regulator of cell fate decisions and astrocyte differentiation during central nervous system (CNS)

development (Benner et al, 2013; Wang et al, 2020; Zamboni et al, 2020). We leveraged this model to perform functional and metabolomic studies and identified state-specific metabolic programs. NPC-like glioma cells relied on amino acid metabolism and high mitochondrial oxygen consumption. Astrocyte-like glioma cells exhibited profound alterations in mitochondrial metabolism, increased ROS production and high lipid peroxidation. Based on these programs, we then identified an astrocytic state that is selectively vulnerable to inhibition of the lipid hydroperoxidase GPX4 and ferroptosis, a regulated non-necroptotic form of cell death driven by iron-dependent lipid peroxidation. High sensitivity to GPX4 inhibition of astrocyte-like glioma populations was directly linked to altered mitochondrial activity at complex I. Treating early passage patient derived cell lines and acute slice cultures generated from surgical specimens of GBM selectively targeted quiescent astrocyte-like glioma cells, highlighting the potential clinical significance of our findings.

# Results

## Activated Notch induces a slow-growing astrocytic phenotype in a PDGFB/p53−/− murine glioma model

While genetically engineered retrovirally induced murine glioma models recapitulate the key histopathological features of human GBM, they do not recapitulate the diversity of glioma states, limiting their utility in pre-clinical studies of cell state-specific druggable targets (Couturier et al, 2020; Weng et al, 2019). Notch activation in neural (NPC) or oligodendrocyte (OPC) progenitor cells represses progenitor programs, initiating and maintaining astrocytic differentiation (Benner et al, 2013; Dray et al, 2021; Engler et al, 2018; Wang et al, 2020; Zamboni et al, 2020). Given its critical role in regulating proliferation and lineage trajectories in the CNS, we hypothesized that Notch activation could have similar effects on glioma phenotype. To test this, we generated two genetic murine glioma models by injecting an HA tagged PDGFB-IRES-Cre expressing retrovirus to target progenitor cells in the subcortical white matter of p53$^{fl/fl}$ mice (Eyler et al, 2020), with one of the models harboring a constitutively-active form of Notch1 in transformed cells at tumor inception (Fig. 1A). Compared to the p53 control model, the N1IC model showed significantly longer survival, ranging from 29 to 151 days post-injection (dpi) (Fig. 1B), and longer latency in tumor formation, as confirmed by weekly bioluminescence imaging (Fig. 1C). Serial MRI scans also demonstrated aggressive tumors in the p53 model at approximately four weeks post-injection while N1IC mice developed radiographically detectable tumors by 60 dpi at the earliest (Appendix Fig. S1a). Histologically, both models exhibited high grade features at end stage (Appendix Fig. S1b), however, the N1IC model showed significantly lower proliferation in the transformed and recruited populations, as measured by Ki67 labeling index (Fig. 1D). Quiescent cell populations with high Notch activity have been identified in primary, treatment naïve GBMs (Liau et al, 2017). Notably, slow cycling persister cells driven by Notch signaling also emerge in GBM after prolonged anti-proliferative drug exposure (Liau et al, 2017).

We next characterized the transcriptional states of the transformed cells in the two models using scRNA-seq. N1IC

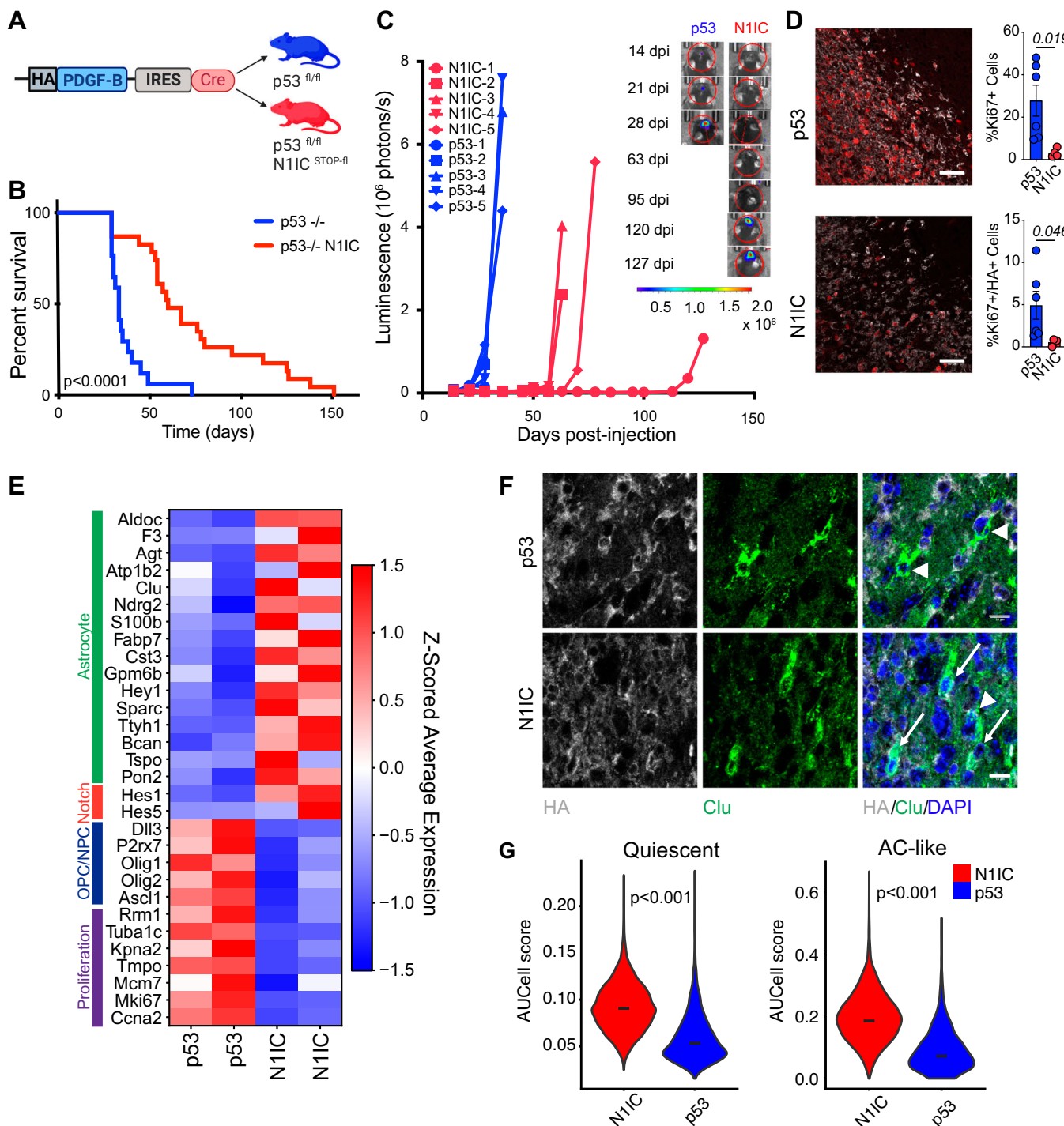

glioma cells expressed higher levels of astrocytic markers while p53 transformed cells expressed higher levels of proliferation markers as well as NPC and OPC markers (Fig. 1E, Dataset EV1). Canonical Notch downstream targets were highly expressed in the transformed N1IC population, including *Hey1, Hes1,* and *Hes5,* transcription factors with key roles in driving quiescence and astrocytic cell fate in the CNS (Engler et al, 2018). The inhibitory Notch ligand *Dll3,* critical in maintenance of undifferentiated neural progenitors (Zhao et al, 2009) was highly expressed in p53

tumor cells (Fig. 1E). N1IC tumor cells activated Notch-dependent transcriptional programs regulating lineage identity in the persister cell population identified in human GBM (Liau et al, 2017) (Appendix Fig. S1c,d, Table EV1). Early NPC or OPC transcription factors were downregulated in N1IC tumors, including *Sox6, Olig1,* and *Myt1.* In contrast, astrocytic master regulators such as *Id3* and *Runx3* were upregulated in the transformed N1IC population (Fig. 1E; Appendix Fig. S1d, Table EV1). Using immunofluorescence and taking advantage of the HA-tag that marks the

**Figure 1.   Notch-driven murine glioma model of a quiescent astrocyte-like tumor cell state.**

(A) Schematic depicting genetically engineered murine glioma models with or without Notch activation. HA tagged PDGF-B IRES Cre retrovirus was injected in the corpus callosum of p53$^{fl/fl}$ or p53$^{fl/fl}$ N1IC$^{STOP-fl}$ mice at 6 weeks. (B) Survival curve for the two glioma models. Survival was significantly longer in p53$-/-$ N1IC mice compared to p53$-/-$ only mice, *p* calculated by Mantel–Cox Log-rank test. Note the significant variability in survival for N1IC mice (29–151 days survival). (C) Serial IVIS imaging demonstrating increased latency in signal detection and tumor formation in N1IC tumors compared to p53 tumors with subsequent sharp increase in bioluminescence and aggressive tumor growth. Inset demonstrates representative IVIS images for one mouse from each model. Also refer to Appendix Fig. S1. (D) Representative immunofluorescence images of Ki67 (red) and HA (gray) in N1IC and p53 endstage tumors. Quantification of Ki67+ cells and Ki67/HA double + cells demonstrating a decreased proliferative transformed population in N1IC tumors as well as an increased recruited proliferative population in p53 tumors. Bar graph shows mean proportions ± SEM. *P*-values calculated by Welch's t test, data pooled from *n* = 6 p53/*n* = 4 N1IC animals. Scale bar, 50 μm. (E) Heatmap showing the expression of selected glioma cell state markers, proliferation genes as well as Notch canonical downstream targets in the p53 and N1IC transformed populations, as derived from scRNA-seq of the retrovirus induced tumors (*n* = 2 from each model). Also refer to Appendix Fig. S1. (F) Representative immunofluorescence of Clu (green) and HA (gray) demonstrating double positive transformed astrocytic tumor cells only in the N1IC model (arrows) with presence of Clu+/HA- non-transformed tumor-associated astrocytes in both models (arrow heads). Scale bar, 10 μm. (G) Violin plots of AUCell scores of the AC-like cell state and quiescence gene signatures, as derived from single cell RNAseq, in tumor cells from the two models (*n* = 2 mice per genotype). *P*-value calculated by Welch's t test. Source data are available online for this figure.

transformed cells (Methods), we confirmed the presence of this Clu+/HA+ astrocytic glioma cell population in Notch tumors, while Clu was only seen in HA-negative tumor-associated, untransformed astrocytes in the p53 model (Fig. 1F). We lastly mapped out the landscape of tumor cell states based on proliferation status in the two models. While a significant proportion of the N1IC transformed tumor cells remained proliferative, the majority were quiescent and had AC-like features (Fig. 1G). Thus, even under constitutive Notch activation a subset of transformed glioma cells can re-enter the cell cycle and promote tumor growth, similar to subpopulations of persister cells identified in other cancer types (Oren et al, 2021). Based on these findings, we demonstrate that p53 and N1IC tumors are valid models to study diverse transcriptional tumor cell states in GBM.

## Multi-omic studies reveal cell state-specific metabolic programs

In the CNS, lineage identity is tightly linked to specialized metabolic programs (Llorens-Bobadilla et al, 2015). In glioma, few studies have focused on correlations between transcriptional cell state and metabolism (Bonnay et al, 2020; Garofano et al, 2021; Ojha et al, 2022). We probed this potential link by integrating transcriptomic and metabolomic analysis on murine tumors combined with functional studies on primary cell cultures derived from the two models (Fig. 2A). First, we performed gene ontology analysis using the scRNA-seq data of the transformed cells from the two murine models. This revealed profound differences in metabolic programs at the transcriptional level, with AC-like N1IC glioma cells showing significant enrichment in genes associated with mitochondrial metabolism (electron transport chain/ETC, oxidative phosphorylation, tricyclic acid cycle/TCA, fatty acid β-oxidation/FAO), lipid peroxidation and redox balance. Notably, AC-like neoplastic populations were enriched in programs involved in oxidative stress responses, including taurine/hypotaurine metabolism and glutathione peroxidase activity. In contrast, NPC-like populations demonstrated enrichment in amino acid metabolism (seleno-aminoacids, alanine, aspartate, glutamate) and nitrogen catabolism (Fig. 2B, Dataset EV2).

Next, we performed global metabolite profiling via LC-MS analysis on endstage tumors from the two models (Methods). Partial least-squares discrimination analysis (PLS-DA) of 750 polar and non-polar metabolites separated the two models based on

metabolic features (Fig. 2C, Dataset EV3). Notably, metabolites resulting from oxidative stress and oxidation/reduction reactions, such as monohydroperoxi polyunsaturated fatty acids and oxylipins, specifically epoxyeicosatrienoic acids (EETs) and the corresponding dihydroxy derivatives (DHET), were significantly enriched in the AC-like model, in line with the transcriptional programs identified based on scRNA-seq (Fig. 2B,D). Galactose, a metabolite regulating oxidative phosphorylation, was also higher in the astrocytic murine model. In contrast, p53 tumors had more abundant arginine and alanine in concordance with transcriptional programs active in the transformed populations driving amino acid metabolism. Furthermore, the rapidly proliferating NPC model had higher levels of fructose 1,6 bisphosphonate, demonstrating a higher glycolytic flux (Fig. 2D, Dataset EV3). Overall, these corroborated multi-omic findings demonstrate significant differences in metabolic programs between the two glioma cell state models, in line with their phenotype and transcriptional programs.

To further dissect cell state-specific functional and metabolic differences in transformed populations, we next isolated primary cell cultures from end-stage tumors. N1IC tumor cells retained the N1IC transgene and showed increased expression of Notch downstream targets *Hey1* and *Hes5* (Appendix Fig. S2a,b). The Notch1 intracellular domain (N1ICD) was highly expressed and exhibited nuclear translocation in cells isolated from the Notch model (Appendix Fig. S2c,d). Primary cultures were comprised of pure transformed populations with ubiquitous expression of the HA tag (Appendix Fig. S2c). We performed liquid chromatography-mass spectrometry (LC-MS/MS) untargeted metabolic and lipidomic profiling on these primary cell cultures from the two models (Fig. 2E–G). PLS-DA analysis of 203 quantified polar metabolites clearly separated the p53 and N1IC cell lines, with 25 metabolites differentially abundant between the two populations (Fig. 2E,F, Dataset EV3). Notably, metabolic ontology analysis (Methods) in N1IC cells revealed enrichment in oxidative stress response and redox balance pathways, such as cysteine/methionine metabolism (L-serine, L-cystine) and taurine/hypotaurine metabolism (Fig. 2F; Appendix Fig. S3a, Dataset EV3). N1IC cells were enriched in D-sedoheptulose 7-phosphate and gluconic acid, metabolites in the pentose phosphate pathway, a pathway highly active in astrocytes driving NADPH regeneration for reductive recycling of glutathione (GSH) under oxidative stress (Vicente-Gutierrez et al, 2019) (Fig. 2F; Appendix Fig. S3a). Joint transcriptomic-metabolomic pathway analysis in the N1IC model

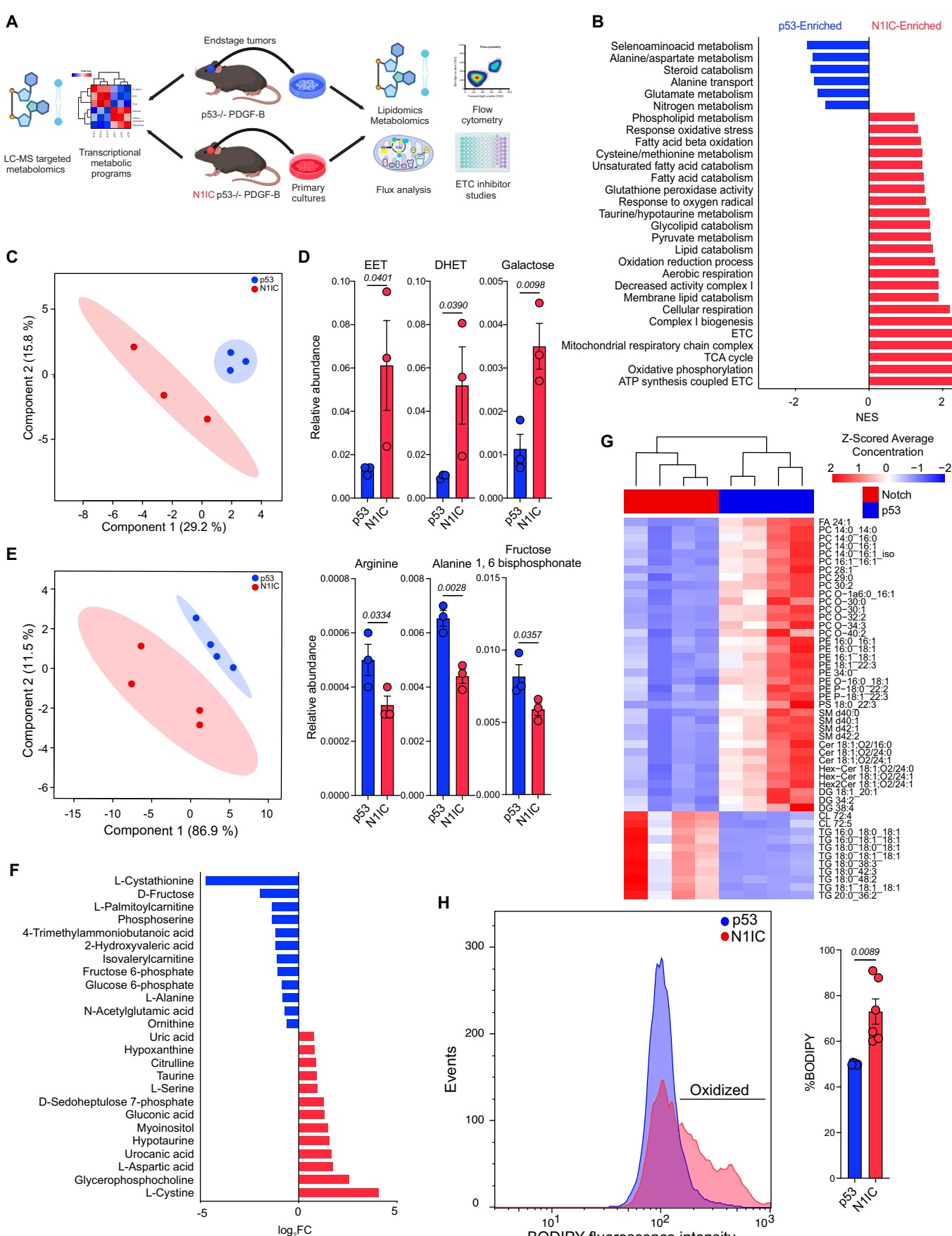

**Figure 2. Multi-omic analysis reveals differences in metabolic programs between N1IC and p53 models.**

(A) Schematic diagram depicting workflow for multi-omic analysis including gene ontology for transcriptional metabolic programs using scRNA-seq and metabolomic analysis using LC-MS in murine tumors as well as functional studies, metabolic and lipidomic LC-MS analysis of primary cell cultures isolated from the two models. Also refer to Appendix Fig. S2 for further characterization of isolated cell lines. (B) Bar graph depicting significant metabolic gene ontologies in the two models via GSEA. NES – normalized enrichment score. Full list of transcriptional metabolic programs provided in Dataset EV2. (C) PLS-DA analysis based on LC-MS metabolomic/lipidomic studies performed on p53 and N1IC endstage tumors demonstrating separation of models based on metabolic features. Shaded area represents 95% CI. Full list of analyzed metabolites provided in Dataset EV3. $n = 3$ independent tumors per model. (D) Bar graphs depicting mean ± SEM relative abundance of select metabolites in the two murine models. $P$-values calculated via unpaired one-tailed $t$-test. $n = 3$ independent tumors per model. (E) PLS-DA analysis of LC-MS untargeted metabolomics performed on p53 and N1IC primary cell cultures demonstrating separation of models based on metabolic profile. Shaded area represents 95% CI. $n = 4$ replicates per model. Full list of analyzed metabolites provided in Dataset EV3. (F) Bar graph demonstrating $\log_2$FC of significant differentially enriched metabolites in N1IC and p53 cells. $P$-values provided in Dataset EV3. Also refer to Appendix Fig. S3 for in depth description of N1IC metabolic pathways based on integrated transcriptomic and metabolomic data. (G) Heatmap of differentially enriched lipid species in p53 and N1IC cell cultures in both the positive and negative mode. Scale bar represents Z-scored average concentration of distinct lipid species normalized by protein concentration. FDR-corrected $P$-values (Welch's $t$-test) and lipid ontology analysis provided in Dataset EV4 and Appendix Fig. S3. (H) Ridge plot depicting flow cytometry of BODIPY-C11 fluorescence demonstrating higher lipid peroxidation in N1IC vs. p53 primary cell culture at baseline. Right: Quantification of lipid peroxidation in p53 and N1IC cell lines. $P$-value calculated by Welch's $t$-test. Data pooled from $n = 6$ independent experiments.

(Methods) revealed compensatory transcriptional activation of mitochondrial metabolic programs with upregulation of genes encoding enzymes in the TCA cycle (*Mdh1, Sdhc1, Aco2*) or the pyruvate pathway (*Pck2*). N1IC cells also exhibited ineffective cysteine/methionine metabolism, with degradation of cysteine to taurine/hypotaurine via upregulated cysteine sulfinic acid decarboxylase (*Csad*) instead of utilization for GSH synthesis (Appendix Fig. S3b,c, Dataset EV3). In contrast, p53 cells were enriched in glycolysis and galactose pathway metabolites (D-fructose, glucose 6-posphate, fructose 6-phosphate), programs shown to play a major role in the energy balance of rapidly proliferating glioma cells (Hoang-Minh et al, 2018). Furthermore, p53 cells had high levels of the L-cystathionine metabolite and increased expression of glutamate-cysteine ligase genes (*Gclm*), both involved in GSH synthesis via the transsulfuration pathway. In line with the transcriptomic ontologies and tumor metabolome, p53 cells demonstrated high levels of L-alanine, which has been shown to inhibit glial fate of NPCs in the CNS (Radu et al, 2019) (Fig. 2D,F; Appendix Fig. S3a).

Lipid profiling also revealed differences in the two models, with p53 cells showing a higher abundance and variety of lipid species compared to the N1IC cells (Fig. 2G). Eleven lipid types were differentially enriched in N1IC cells compared to 37 species in p53 cells. The lipid profile segregated by model, in both the negative and positive ion mode (Appendix Fig. S3d). Notably, p53 cells had higher levels of multiple functionally important lipid groups, including glycerophosphocholines (PC), glycerophosphoethanolamines (PE), ceramides (Cer) and sphingolipids (SM) which are essential in stabilizing the membrane lipid bilayer (Fig. 2G; Appendix Fig. S3e, Dataset EV4). In contrast, the most elevated lipid species in N1IC cells were triacylglycerols (TG), with depletion of most other lipid species containing fatty acids. Interestingly, the only differentially abundant polyunsaturated fatty acyl (PUFAs) containing phospholipid species were in the p53 cell lines. We hypothesized that differences in transcriptomic, metabolomic, and lipidomic data result from higher baseline lipid peroxidation in N1IC cells. To verify this, we used the fluorescent dye BODIPY-C11 and quantified baseline lipid peroxides in the two cell types via flow cytometry (Fig. 2H). N1IC cells had a higher oxidized fraction compared to p53 cells suggesting that N1IC cells may be undergoing baseline low-level ferroptosis, a form of programmed cell death based on iron-dependent oxidation of phospholipids with polyunsaturated fatty acyl tails.

## Mitochondrial complex I dysfunction leads to increased ROS production in the astrocyte-like N1IC state

Aside from key biosynthetic and energetic functions, mitochondria play permissive roles in cell fate decisions during neural development via ROS production. Given the metabolic alterations and redox imbalance noted in N1IC cells, we further examined the mitochondrial phenotype and electron transport chain (ETC) function in the two models. Notably, based on scRNA-seq data, genes regulating mitochondrial complex I activity, especially genes encoding NADH dehydrogenase subunits, were significantly upregulated in transformed populations from the N1IC AC-like mouse model. *Keap1*, an important sensor of oxidative and electrophilic stress regulating *Nrf2* activity was also highly upregulated. In turn, several Nrf2 target genes (*Txnrd1, Srxn1, Gclm, Cul3*) were downregulated in the N1IC AC-like model. The *Sdhc* gene encoding a subunit of the succinate dehydrogenase enzyme, important in regulating electron flow through the ETC, was significantly upregulated in the AC-like cell state model, as were several thioredoxin genes involved in critical redox reactions (*Txn1, Txn2*) (Fig. 3A). Altered complex I activity has been recently shown to increase NADH levels, modulating ROS production by altering electron flow in the ETC (Noch et al, 2024; Ojha et al, 2022; Weiss-Sadan et al, 2023). Notably, NADH/NAD levels have been directly linked to quiescence in GBM (Bonnay et al, 2020). Reductive substrates were enriched in the AC-like murine model compared to the NPC-like model, with significantly increased NADH/NAD ratio and marginally elevated NADPH/NADP and CoQH2/CoQ4 or CoQH2/CoQ9 ratios (Fig. 3B, Dataset EV3). These findings pointed towards a potentially altered function of the ETC in the AC-like model (Noch et al, 2024; Ojha et al, 2022; Weiss-Sadan et al, 2023). Consequently, N1IC cells demonstrated significantly higher total ROS production compared to p53 cells as well as higher baseline mitochondrial-specific lipid peroxidation, measured via mitoCLOX flow cytometry (Fig. 3C). In turn, the reduced form of the antioxidant glutathione (GSH) and the GSH/GSSG ratio were significantly decreased in both the AC-like N1IC murine tumors and primary N1IC cell cultures, further pointing towards significant redox imbalance and exhaustion of ROS quenching mechanisms in the astrocytic model (Fig. 3D).

Higher ROS production can directly impact mitochondrial integrity and function. Maladaptive ROS induced ROS release, regulated by the mitochondrial permeability transition pore

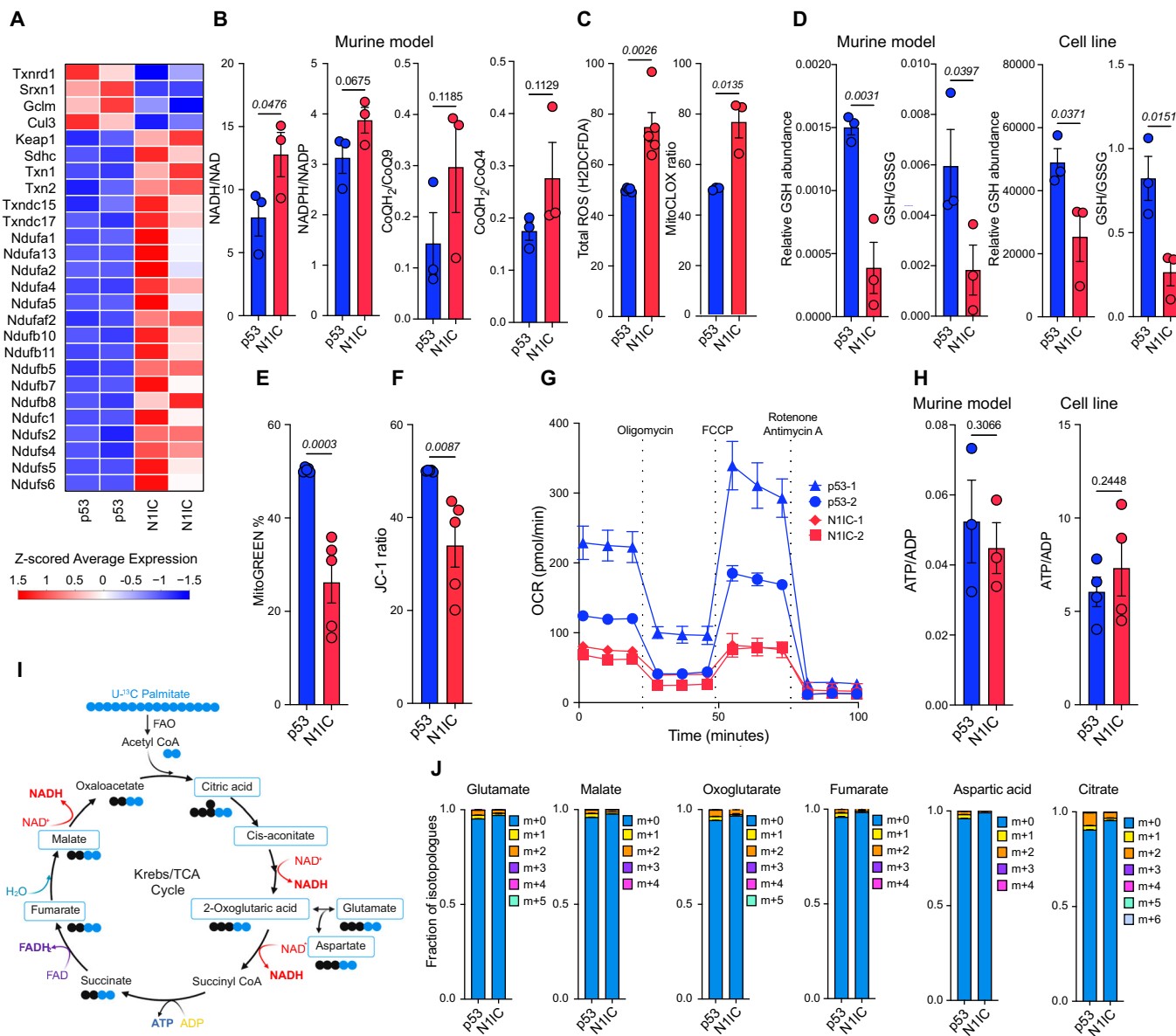

**Figure 3. Altered complex I activity leads to increased ROS and mitochondrial lipid peroxidation in the N1IC AC-like model.**

(A) Heatmap showing differentially expressed genes involved in complex I activity or redox metabolism identified by scRNA-seq in the p53 and N1IC transformed cell populations. $n = 2$ samples for each model. (B) Bar graph depicting mean ratio ± SEM of reductive substrates based on LC-MS analysis of tumor tissue from the p53 and N1IC murine models. $n = 3$ samples for each model. $P$-values calculated by unpaired one-tailed t-test. Also refer to Dataset EV3. (C) Bar graph depicting mean ± SEM H2DCFDA and MitoCLOX ratio demonstrating increased ROS and mitochondrial lipid peroxidation in N1IC primary cultures compared to p53. Data pooled from $n = 5$ (H2DCFDA) and $n = 3$ (MitoCLOX) independent experiments. $P$-values calculated by unpaired two-tailed t-test. (D) Bar graphs depicting mean GSH relative abundance and GSH/GSSG ratio ± SEM in the murine tumors and primary cell cultures from p53 and N1IC models. $P$-values calculated by unpaired one-tailed t-test. $n = 3$ independent tumor samples for each model (murine model, left) and $n = 3$ independent measurements in a pair of cell lines (cell line, right). (E) Bar graph of MitoGREEN median gate ± SEM demonstrating difference in mitochondrial numbers in p53 and N1IC primary cell cultures. $P$-values calculated by unpaired two-tailed t-test. Data pooled from $n = 5$ independent experiments. (F) Bar graph depicting mean ± SEM JC-1 ratio demonstrating decreased mitochondrial membrane potential in N1IC compared to p53 primary cell cultures. Data pooled from $n = 5$ independent experiments. $P$-values calculated by unpaired two-tailed t-test. Also refer to Appendix Fig. S4. (G) Seahorse OCR analysis of two different p53 and N1IC primary cell cultures demonstrating higher energetic metabolism in p53 tumor cells compared to N1IC tumor cells. Each point represents mean, error bars represent SEM of minimum $n = 3$ replicates. Also refer to Appendix Fig. S4. (H) Bar graph depicting mean ATP/ADP ratio ± SEM in murine tumors and primary cell cultures demonstrating no difference in energetic balance between the p53 and N1IC models. $n = 3$ (tumors) or $n = 4$ (cell culture) independent samples for each model. $P$-values calculated by unpaired two-tailed t-test. (I) Schematic depiction of C$^{13}$ palmitate flux analysis to measure fatty acid oxidation (FAO) in primary cell cultures. Boxed metabolites were measured and are depicted in Panel (J). (J) Quantification of TCA cycle metabolites incorporating C$^{13}$ from FAO of tagged palmitic acid demonstrates increased uptake in the p53 primary cell cultures. Stacked bar graphs depict mean ± SEM from $n = 3$ independent measurements. Source data are available online for this figure.

(mPTP), induces oxidative damage at the level of the mitochondrial membrane. Indeed, N1IC cells had significantly lower mitochondrial numbers compared to p53 cells (Fig. 3E). Thus, we further investigated mitochondrial function and basal energy metabolism. First, we measured baseline mitochondrial membrane potential ($\Delta\Psi_m$) in the two cell models using the JC-1 and MitoRED mitochondrial dyes and found significant membrane depolarization and less dye uptake in N1IC cells (Fig. 3F; Appendix Fig. S4a). Notably, JC-1 is a ratiometric dye ensuring difference in $\Delta\Psi_m$ is independent of mitochondrial numbers in different cell line models. We also measured the oxygen consumption rate (OCR) and found significantly lower basal and maximal respiration as well as lower respiratory capacity in the N1IC cells compared to p53 cells (Fig. 3G; Appendix Fig. S4b). Notably, there were no significant differences between models in non-mitochondrial OCR.

Seahorse extracellular acidification rate (ECAR) analysis suggested significantly higher glucose usage in the rapidly proliferating p53 NPC-like cell state (Appendix Fig. S4c), in line with recent studies (Garofano et al, 2021; Hoang-Minh et al, 2018). This is further supported by enrichment in glycolytic metabolites glucose 6-P and fructose 6-P in p53 cells (Fig. 2F). The slow-cycling N1IC cells had decreased energetic requirements and lower ATP production (Appendix Fig. S4b). Nonetheless, despite the altered mitochondrial state in N1IC cells, there were no significant differences in ATP/ADP ratio between tumor or cell line models, possibly due to decreased energetic requirements in the quiescent AC-like populations (Fig. 3H). Given the recently uncovered roles of cysteine in modulating mitochondrial function and energy metabolism in glioma (Noch et al, 2024; Upadhyayula et al, 2023), we also used cysteine and methionine deprivation and observed significant decrease in p53 maximal respiration but minimal effect on basal OCR. Intriguingly, there were no effects of cysteine/methionine deprivation on energy metabolism in the Notch model, already chronically cysteine deprived secondary to ineffective metabolism (Appendix Figs. S3c and S4d,e). We further investigated mitochondrial function by comparing FAO in the two cell models via C$^{13}$-palmitate flux analysis. In line with the lower OCR, we found that N1IC cells have lower FAO, with reduced uptake of labeled C$^{13}$ from palmitate oxidation in TCA cycle intermediates such as glutamate, fumarate, aspartate, or citrate (Fig. 3I,J). To compensate, N1IC transformed populations upregulate transcriptional FAO metabolic programs (Fig. 2B). Overall, AC-like glioma populations appear to have an altered mitochondrial state with decreased $\Delta\Psi_m$, decreased OCR, decreased FAO, decreased glycolysis but with increased mitochondrial lipid peroxidation and ROS production. Thus, we conclude that the altered mitochondria are the main driver of redox imbalance noted in the AC-like model.

## Differential sensitivity to Gpx4 inhibition and ferroptosis of the astrocyte-like N1IC cell state

We next investigated if these metabolic differences can be leveraged to design cell state-targeted therapies. As detailed above, N1IC cells harbor an altered mitochondrial state with increased mitochondrial and cellular lipid peroxidation as well as increased global ROS production (Figs. 2 and 3). Furthermore, high ROS levels and ineffective cysteine metabolism can lead to depletion of glutathione

stores (Fig. 3D), a principal ROS quenching mechanism (Appendix Fig. S3c). The selenocysteine enzyme glutathione peroxidase 4 (GPX4), using GSH as a cofactor, plays a central role in protecting cells from oxidative stress, particularly under thiol deprivation conditions (Dixon et al, 2012; Jiang et al, 2021). Given these metabolic differences, we compared the sensitivity of N1IC and p53 cells to induction of ferroptosis with pharmacologic (RSL3) and genetic (siRNA) inhibition of GPX4. In vitro viability assays showed that N1IC cell lines were preferentially sensitive to RSL3 compared to p53 cell lines (Fig. 4A). We investigated if this differential sensitivity was secondary to ferroptosis or off-target effects of RSL3 using three independent modalities. First, cell death induced by GPX4 inhibition in N1IC cells was significantly reduced, although not completely abrogated, with addition of the ferroptosis rescue drug Ferrostatin-1 (Appendix Fig. S5a). Ferrostatin-1 decreased efficacy of GPX4 inhibition in both N1IC and p53 cell lines (Fig. 4B). In contrast, addition of the apoptosis inhibitor Z-VAD-FMK or the necroptosis inhibitor Necrostatin-1 did not modify sensitivity to RSL3 in the N1IC cell line (Appendix Fig. S5a). Notably, treatment of a N1IC cell line with Ferrostatin-1 in the absence of RSL3 did not affect cell viability, suggesting that low level baseline lipid peroxidation and ferroptosis encountered in this cell state do not lead to significant levels of cell death (Appendix Fig. S5b).

Second, in both cell models RSL3 induced upregulation of canonical, albeit not entirely specific, ferroptosis markers at a transcriptional level, including glutathione-specific γ-glutamylcyclotransferase 1 (*Chac1*), prostaglandin-endoperoxide synthase 2 (*Ptgs2*) and the system x$_c^-$ antiporter (*Slc7a11*) (Fig. 4C). Importantly, all three markers are upregulated in vehicle-treated N1IC cells compared to p53 cells, further suggesting low level baseline ferroptosis in the AC-like cell state. All three markers are significantly higher in RSL3-treated N1IC cells, demonstrating a higher ferroptotic response in the AC-like cell state. Third, exposure of N1IC cells to RSL3 led to a significant increase in BODIPY-C11 fluorescence, a marker of increased lipid peroxidation, starting at approximately 10 min after drug treatment (Appendix Fig. S5c). Taken together, these results demonstrate that in both p53 and N1IC cell lines RSL3 inhibition of GPX4 induces ferroptosis, with the N1IC cells showing significantly higher sensitivity to ferroptosis.

Next, we asked if increased ferroptosis sensitivity in N1IC cells is specifically linked to GPX4 dependency. N1IC cell lines demonstrated significantly higher sensitivity and lower area under the curve (AUC) for both ML162, a chloroacetamide, and ML210, a nitroisoxazole (Appendix Fig. S5d,e). ML162 and ML210 are GPX4 inhibitors with substantially different chemical structures, and, therefore, distinct off-target effects. Importantly, inhibition of GPX4 with ML162 and ML210 induced ferroptosis, which was rescued with addition of Ferrostatin-1 (Appendix Fig. S5d,e). We also tested the effects of GPX4 genetic inhibition in the two cell models using siRNA. In line with response to pharmacologic inhibition, N1IC cells were significantly more sensitive to GPX4 knockdown compared to p53 cells (Fig. 4D). GPX4 knockdown induced lipid peroxidation and cell death after 48 h and these effects were partially reversed with addition of Ferrostatin-1 (Fig. 4D; Appendix Fig. S5f–h). Thus, differential sensitivity to ferroptosis in the two models is directly linked to degree of GPX4 dependency.

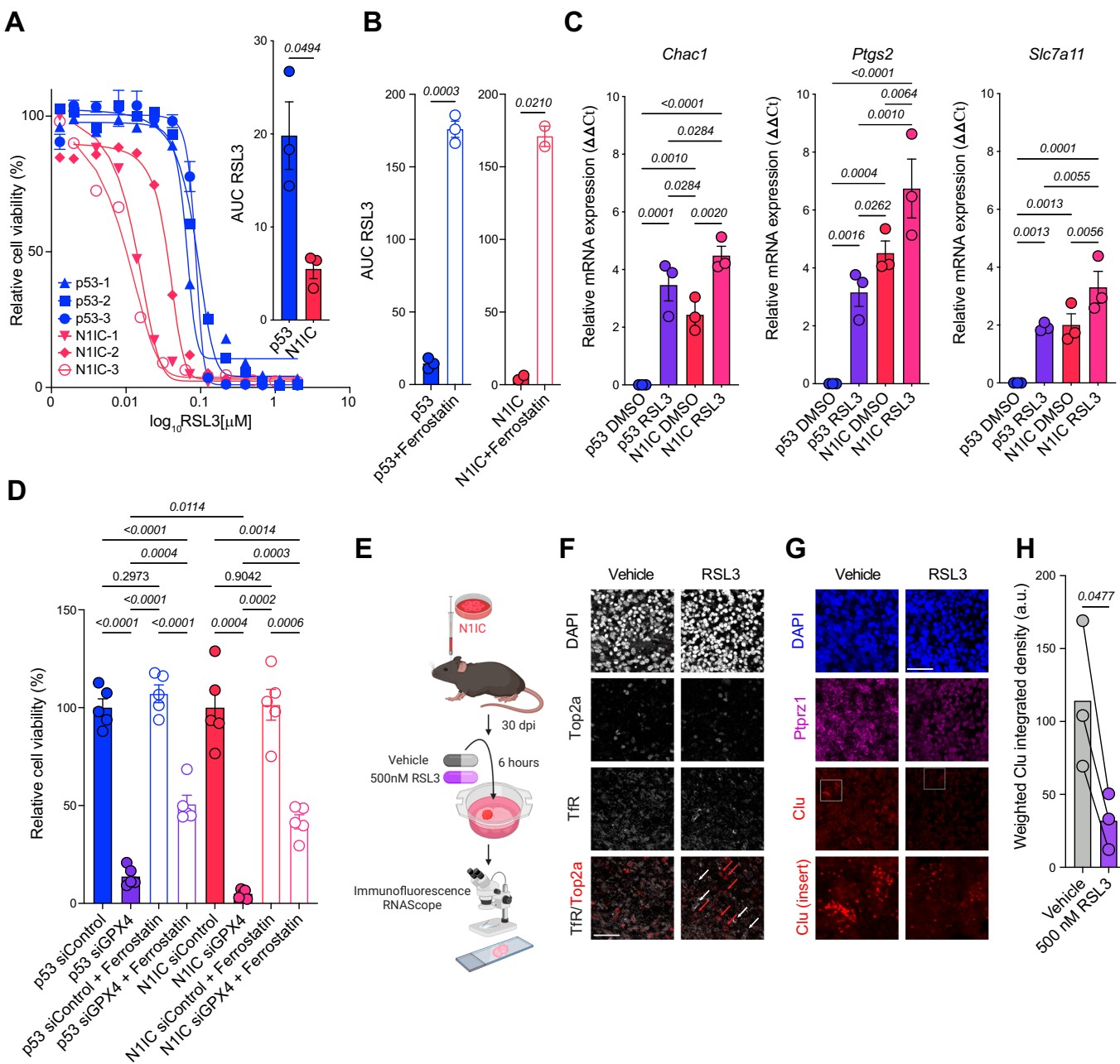

**Figure 4. Differential sensitivity to Gpx4 inhibition and ferroptosis of the quiescent AC-like cell state.**

(A) RSL3 drug screen on $n = 3$ p53 and N1IC independent primary cell cultures, error bars represent SEM from $n = 3$ replicates. Inset: Bar graph depicting mean area under the curve (AUC) ± SEM for panel a, comparison by Welch's t test. (B) AUC from RSL3 dose–response curves with and without 2 μM Ferrostatin-1 in $n = 3$ p53 cell lines and $n = 2$ N1IC cell lines. P-values calculated via paired t-test. Also refer to Appendix Fig. S5. (C) PCR ΔΔCT ± SEM values of canonical ferroptosis markers in p53 and N1IC primary cell cultures treated with 500 nM RSL3 or DMSO control for 2 h. Data pooled from $n = 3$ independent experiments. Normalized to actin, p53 DMSO used as reference. Displayed Q-values calculated by ANOVA with FDR correction via Benjamini, Krieger, and Yekutieli. (D) Bar graph depicting mean cell numbers ± SEM assessed in primary cell cultures from the two models undergoing transfection with control or GPX4 siRNA for 48 h demonstrating increased cell death in the N1IC model and partial rescue with Ferrostatin-1. $n = 5$ fields per condition. Displayed P-values calculated by one-way Brown-Forsythe and Welch ANOVA tests. Also refer to Appendix Fig. S5. (E) Experimental setup schematic representation of murine organotypic slice cultures. 3 slices per condition were generated from 3 different tumor-bearing mice. Mice were sacrificed at 30 dpi. Treatment with 500 nM RSL3 or DMSO vehicle for 6 h. (F) Double immunofluorescence of proliferation marker Top2a and ferroptosis marker transferrin receptor (TfR) demonstrating lack of TfR staining in vehicle treated slices and upregulation of TfR after 500 nM RSL3 in Top2a negative non-proliferating cells. Red arrows mark Top2a+ cells, white arrows mark TfR+ cells. Scale bar, 50 μm. (G) RNAScope of Ptprz1 (tumor marker—magenta) and Clu (astrocytic marker—red) after 500 nM RSL3 or DMSO vehicle demonstrating depletion of the Clu+ transformed cell population. Scale bar, 50 μm, insets 40 × 40 μm. (H) Quantification of weighted integrated density on $n = 3$ independent slice cultures. Bar graph depicting mean with individual paired values. P-value via paired t-test. Source data are available online for this figure.

Intriguingly, both cell models were resistant to IKE, a potent system $x_c^-$/Slc7a11 inhibitor and ferroptosis inducer. Doses of up to 20 mM were insufficient to achieve IC50 and addition of Ferrostatin-1 did not significantly alter AUC (Appendix Fig. S6a,b). To better understand this finding, we first verified if murine glioma cells expressed Slc7a11. We found minimal differences in Slc7a11 at the transcriptional level between cell models (Appendix Fig. S6c) and Western blot analysis confirmed that all cell lines expressed relatively high levels of the transporter (Appendix Fig. S6d). Notably, exposing N1IC cells to 100 nM or 1 µM IKE led to decrease in GSH levels but without significant changes in GSH/GSSG ratio (Appendix Fig. S6e), suggesting that our glioma models are insensitive to system $x_c^-$ inhibitors because of alternative methods of generating cysteine, such as the transsulfuration pathway. Cystathionine, an important intermediate in the trans-sulfuration pathway, was detected in both cell models (Dataset EV3). Other possibilities such as additional pools of glutathione or other thiols that circumvent the need for system $x_c^-$ cannot be excluded and further studies are needed to explore this finding. Thus, inhibition of Slc7a11 did not induce ferroptosis in either N1IC or p53 cell lines, further demonstrating the specific sensitivity to inhibition of GPX4, particularly in the N1IC model.

Tumor microenvironment and intercellular interactions can significantly impact ferroptosis (Wu et al, 2019). To test the effects of ferroptosis on N1IC tumor cells in the context of the complex cellular architecture of gliomas, we generated acute slice cultures from N1IC cell-transplanted murine tumors at 30 dpi, treated the slices with either vehicle or RSL3 and performed immunofluorescence and RNAscope studies (Fig. 4E). GPX4-driven ferroptosis specifically targeted the non-cycling tumor cell states, with upregulation of the canonical ferroptosis marker TfR (Feng et al, 2020) solely in Top2a negative cells (Fig. 4F). Furthermore, RSL3 specifically targeted the AC-like transformed population with significant depletion of Ptprz1+ Clu+ cells after 6 h of 500 nM RSL3 (Fig. 4G,H). Overall, these findings support a cell state-specific metabolic vulnerability to GPX4 inhibition and ferroptosis in quiescent AC-like glioma cell populations.

## Mitochondrial lipid peroxidation and complex I activity control sensitivity to Gpx4 inhibition and ferroptosis

Mitochondria generate most of the energy and ROS in cells, playing a central role in programmed cell death. Therefore, we sought to dissect the link between the altered mitochondrial state and increased sensitivity to GPX4-driven ferroptosis in the AC-like model by modulating the electron flow through the ETC (Fig. 5A; Appendix Fig. S7a–e). Notably, RSL3 treatment induced significant upregulation of mitochondrial lipid peroxidation in N1IC cells, a potential key first step in ferroptosis (Fig. 5B). Treating N1IC cells with IACS-010759, a specific complex I inhibitor, decreased total ROS production as well as cellular and mitochondrial lipid peroxidation (Fig. 5C), and decreased RSL3-induced cell death (Fig. 5D), demonstrating that mitochondrial complex I activity is important in initiating ferroptosis. In contrast, uncoupling oxidative phosphorylation with FCCP did not impact ROS or lipid peroxidation in N1IC cells (Fig. 5E), but induced significant cell death, either in combination with RSL3 or on its own (Fig. 5F, Appendix Fig. S7d), presumably via mechanisms that are independent of ROS or lipid peroxidation. Notably, both IACS

and FCCP reduce ATP production by blocking electron flow in the ETC, but have opposite effects on response to RSL3, either as co-treatment or administered prior to RSL3 treatment (Appendix Fig. S7f). Furthermore, directly inhibiting ATP synthase activity with oligomycin did not impact response to RSL3 (Fig. 5G). Together, these experiments demonstrate that baseline ROS levels and the capacity to increase ROS production, but not ATP levels, influence sensitivity to GPX4 inhibition in the N1IC model. Lastly, to further identify the exact site in the ETC modulating response to ferroptosis in N1IC cells we also inhibited complex III, the other major site of ROS production in the mitochondria, with antimycin and did not find any difference in RSL3 sensitivity (Fig. 5G). Thus, altered activity and ROS production at complex I in the N1IC model drive increased response to GPX4 inhibition and ferroptosis.

Mitochondrial translocator protein (Tspo) plays important roles in ROS generation and mitochondrial respiration (Gatliff et al, 2017), modulating mPTP opening and thereby amplifying ROS production via ROS induced ROS release. Tspo is significantly upregulated in the N1IC model (Fig. 1E, Table EV1) and can further drive redox imbalance induced by complex I dysfunction. Therefore, we also tested if Tspo is a potential driver of increased mitochondrial lipid peroxidation and response to GPX4 inhibitors in N1IC cells. To this end, we pharmacologically inhibited Tspo with PK11195 for 24 h and noted dose-dependent decrease in mitochondrial lipid peroxidation, a small decrease in $\Delta\Psi_m$ and no impact on BODIPY levels or total ROS production (Fig. 5H; Appendix Fig. S7g,h). Cell viability assays showed that blocking Tspo prior to pharmacologic GPX4 inhibition, but not co-treatment, significantly reduced sensitivity to ferroptosis in dose-dependent manner, with almost complete rescue of cell death at 10 µM PK11195 (Fig. 5I). These findings suggest that high Tspo affects sensitivity to ferroptosis via effects on baseline mitochondrial lipid peroxidation. Importantly, unlike complex I, Tspo is not needed to initiate ferroptosis via GPX4 inhibition in the N1IC model.

To further assess the role of mitochondrial lipid peroxidation in driving vulnerability to ferroptosis in N1IC cells we used a battery of ROS scavengers and ferroptosis inhibitors with different mechanisms of action. Deferoxamine (DFO), an iron chelator, decreased mitochondrial lipid peroxidation while Ferrostatin-1, a ROS scavenger that acts at the ER level, had no significant effect (Appendix Fig. S7i). Nonetheless, both Ferrostatin-1 and DFO changed total cellular lipid peroxidation although neither significantly affected total cellular ROS (Appendix Fig. S7j,k). Mitochondrial ROS scavengers decreased both cellular lipid peroxidation and total ROS production (Fig. 5K) and were potent inhibitors of ferroptosis, decreasing RSL3-induced cell death (Fig. 5L). We conclude that iron-dependent mitochondrial ROS generation and lipid peroxidation are key steps in ferroptosis. Because mitochondrial ROS have been implicated in other forms of cell death, specifically apoptosis, we co-treated cells with a mitochondrial scavenger (SKQ1) and the apoptosis inducer staurosporine (STS). Surprisingly, inhibiting mitochondrial ROS production did not decrease apoptosis-driven cell death, suggesting that mitochondrial activity specifically drives response to ferroptosis in N1IC cells (Fig. 5M).

## RSL3 targets quiescent astrocyte-like transformed cell populations in human glioma

To assess the clinical significance of our models, we next investigated the presence of N1IC-like tumor cell populations in

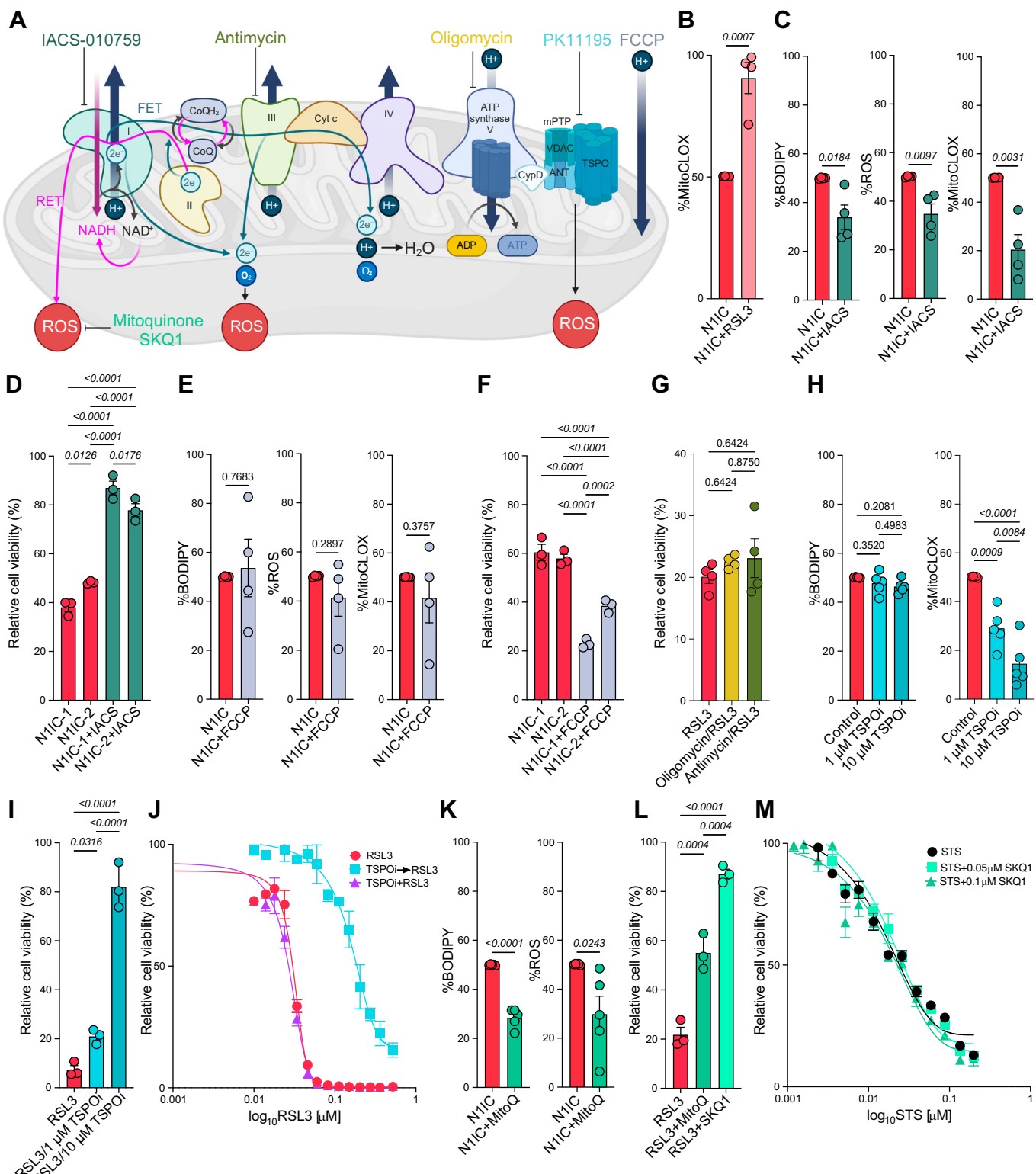

IDH1-wild type GBM patient samples(Yuan et al, 2018). Quiescent tumor cell populations, predominantly adopting the AC-like cell state, were present in all eight patient samples (Appendix Fig. S8a). TSPO, which was directly linked to GPX4-driven ferroptosis in the murine model, was highly upregulated in quiescent tumor cell populations (Appendix Fig. S8a–c). We derived signatures from differentially up and downregulated genes in the N1IC tumor cell population (Appendix Fig. S8d, e) and compiled Spearman correlations between glioma cell state scores (Neftel et al, 2019) and the two N1IC gene signatures (Methods). The "N1IC_up"

◄ **Figure 5.  Mechanistic link between complex I-driven mitochondrial lipid peroxidation and sensitivity to Gpx4-dependent ferroptosis in the AC-like cell state.**

(A) Schematic depiction of electron flow (forward – FET, reverse – RET) in the electron transport chain, sites of ROS production in the ETC and pharmacologic inhibitors used to modulate the ETC in conjunction with GPX4 inhibition. Mitoquinone and SKQ1 are mitochondrial specific ROS scavengers. PK11195 is a TSPO inhibitor (TSPOi) and modulates mPTP opening to control ROS production. Schematic generated with BioRender. (B) Bar graph depicting mean ± SEM MitoCLOX with or without RSL3 treatment (500 nM, 30 min) demonstrating increased mitochondrial lipid peroxidation after GPX4 inhibition. Data pooled from $n = 4$ independent experiments. *P* calculated via unpaired two-tailed t-test. (C) Bar graphs depicting mean ± SEM for BODIPY-C11, H2DCFDA/ROS and MitoCLOX levels with or without inhibition of complex I via IACS-010759 demonstrating decreased total/mitochondrial lipid peroxidation and total ROS production. Data pooled from $n = 4$ independent experiments. *P*-values calculated via unpaired two-tailed t-test. (D) Bar graphs depicting mean cell viability ± SEM after treatment with 20 nM RSL3 alone (red) or RSL3 + IACS-010759 (dark green) in two different N1IC cell lines demonstrating decreased response to ferroptosis after complex I inhibition. $n = 3$ biological replicates. Displayed Q-values calculated by ANOVA with FDR correction via Benjamini, Krieger, and Yekutieli. (E) Bar graphs depicting mean ± SEM for BODIPY-C11, H2DCFDA/ROS and mitoCLOX levels with or without uncoupling of oxidative phosphorylation via FCCP demonstrating no effects on total/mitochondrial lipid peroxidation or total ROS production. Data pooled from $n = 4$ independent experiments. *P*-values calculated via unpaired two-tailed t-test. (F) Bar graphs depicting mean cell viability ± SEM after treatment with 20 nM RSL3 alone (red) or RSL3 + FCCP (light blue) in two different N1IC cell lines demonstrating increased cell death after GPX4 inhibition and uncoupling. $n = 3$ biological replicates. Displayed Q-values calculated by ANOVA with FDR correction via Benjamini, Krieger, and Yekutieli. (G) Bar graphs depicting mean cell viability ± SEM after treatment with DMSO followed by 20 nM RSL3 (red), 0.5 µM Oligomycin followed by 20 nM RSL3 (yellow) or 5 µM Antimycin followed by 20 nM RSL3 (green) demonstrating no effects of ATP synthase or complex III inhibition on response to ferroptosis. $n = 4$ biological replicates. Data normalized to DMSO, Oligomycin or Antimycin alone, respectively. Q-values calculated by ANOVA with FDR correction via Benjamini, Krieger, and Yekutieli. (H) Bar graphs depicting mean ± SEM for BODIPY-C11 and MitoCLOX levels with or without TSPO inhibition with 1 and 10 µM PK11195 (TSPOi) demonstrating dose-dependent decrease in mitochondrial but not in cellular lipid peroxidation. Data pooled from $n = 5$ independent experiments. Displayed Q-values calculated by ANOVA with FDR correction via Benjamini, Krieger, and Yekutieli. Also refer to Appendix Fig. S7. (I) Bar graphs depicting mean cell viability ± SEM after treatment with DMSO followed by RSL3 (red), 1 µM or 10 µM PK11195 followed by RSL3 (turquoise) demonstrating dose-dependent rescue of ferroptosis with TSPO inhibition. $n = 3$ biological replicates. Displayed Q-values calculated by ANOVA with FDR correction via Benjamini, Krieger, and Yekutieli. (J) Dose–response curves in N1IC cells treated with DMSO followed by RSL3 (red), 10 µM PK11195 followed by RSL3 (turquoise) and RSL3 combined with 10 µM PK11195 (purple) demonstrating rescue of cell death with sequential TSPO inhibition only. Error bars represent SEM from $n = 3$ replicates. (K) Bar graphs depicting mean ± SEM for BODIPY-C11 and H2DCFDA levels with or without MitoQ treatment demonstrating decreased lipid peroxidation and total ROS production with mitochondrial specific ROS scavengers. Data pooled from $n = 5$ independent experiments. *P*-values calculated via unpaired two-tailed t-test. (L) Bar graphs depicting mean cell viability ± SEM after treatment with RSL3 alone (red), RSL3 with 0.25 µM MitoQ and RSL3 with 0.1 µM SKQ1 (green) demonstrating rescue of ferroptosis with two different mitochondrial ROS scavengers. $n = 3$ biological replicates. Displayed Q-values calculated by ANOVA with FDR correction via Benjamini, Krieger, and Yekutieli. (M) Dose–response curves in N1IC cells treated with apoptosis inducer STS alone (black), STS combined with 0.05 µM SKQ1 and STS combined with 0.1 µM SKQ1(green) demonstrating no effect on apoptosis driven cell death with mitochondrial ROS scavengers. Error bars represent SEM from $n = 3$ replicates. Source data are available online for this figure.

signature most closely correlated with the AC and MES1/2 gene signatures, while the "N1IC_down" signature clustered with highly proliferative NPC-like states (Appendix Fig. S8d). Notably, in a cohort of five patients undergoing chronic intratumoral delivery of topotecan, a mitotic poison, post-treatment tissue analysis revealed significant enrichment in the "N1IC_up" gene signature (Spinazzi et al, 2022) (Appendix Fig. S8e). These findings demonstrate that the N1IC murine model recapitulates tumor cell populations present in treatment naïve GBM as well as populations expanded after exposure to chronic cytotoxic therapies. To explore cell-state-specific metabolic programs in human glioma, we also calculated the Spearman correlation coefficient between metabolic pathways and cell state signatures using the transcriptomic data from eight human GBM samples (Yuan et al, 2018) (Methods). In line with our previous findings, the AC-like signature highly correlated with multiple metabolic pathways identified in the N1IC model, including fatty acid (FA) transport, superoxide metabolism, and ROS metabolism. Conversely, NPC-like and proliferating cell state scores negatively correlated with these metabolic programs but positively correlated with other pathways, such as regulation of cellular amino acid metabolism (Appendix Fig. S8f, Dataset EV5). Thus, we demonstrate that quiescent AC-like cell states with specific metabolic programs identified in the N1IC murine model are ubiquitously present in human GBM samples.

To dissect ferroptosis effects at the single cell level in human glioma models, we first used a patient-derived glioma neurosphere cell line (TS543). Importantly, at early passages, we can identify a subpopulation of cells with the astrocyte-like phenotype by scRNA-seq, however, this population is eventually selected out or transformed at later passages (Fig. 6A). scRNA-seq on early

passage TS543 neurospheres treated with RSL3 for one or three days demonstrated that the only glioma cell state signature significantly depleted at both time points was the astrocyte-like signature (Fig. 6B). The "N1IC_up" gene signature was also significantly depleted by RSL3 treatment of TS543 neurospheres at both time points (Fig. 6C). Furthermore, we noted downregulation of astrocytic markers including *AQP4, S100B, FABP7* and *F3* after 1-day treatment and *CST3, HOPX, SPARC, CLU* after the 3-day treatment (Fig. 6D, Dataset EV6). Importantly, the 3-day RSL3 treatment also led to depletion of cell populations with the transcriptional mitochondrial phenotype encountered in the N1IC AC-like model, with significantly decreased normalized enrichment scores for ETC, oxidative phosphorylation, and aerobic respiration (Fig. 6E).

We also analyzed glioma cell data from the Cancer Cell Line Encyclopedia (CCLE, Methods) to verify if Notch signaling alone was predictive of response to GPX4 inhibitors using in vitro models. Glioma cell states and proliferation scores were poorly defined in patient derived cell lines, as shown by other recent studies, likely due to lack of important microenvironmental cues (Pine et al, 2020). Only the NPC1 state correlated significantly with Notch signaling (Appendix Fig. S9a, Dataset EV7). Transcriptional Notch activity correlated poorly with the AUC for several GPX4 inhibitors, including RSL3, ML162, the dual GPX4-CoQ$_{10}$ inhibitor FIN56, and the dual GPX4-mevalonate pathway inhibitor Lovastatin. Intriguingly, TSPO expression anticorrelated with RSL3 AUC, suggesting a key role of mitochondrial ROS in driving response to ferroptosis across multiple glioma models (Appendix Fig. S9b, Dataset EV7). Lastly, there were no Notch-dependent differences in metabolites resulting from redox imbalance,

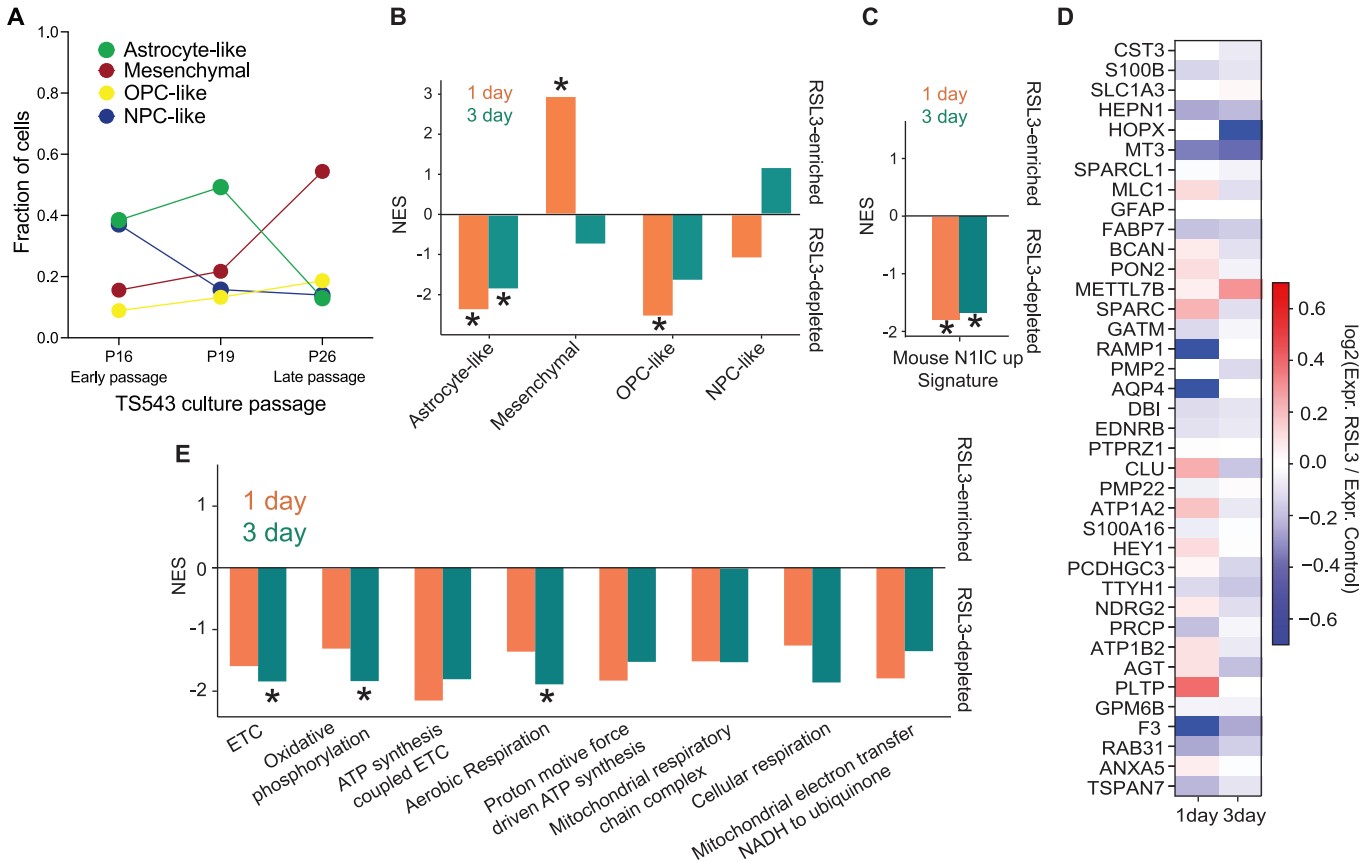

**Figure 6. GPX4 inhibition targets an AC-like subpopulation in an early passage patient derived cell line.**

(A) The phenotypic cell states of TS543 culture change over time. The cell state of each cell from scRNA-seq data of 3 different TS543 culture passages was determined using gene signatures described in Neftel et al (2019). Number of cells at each cell state were counted and used to calculate the cell fraction at each culture passage. (B) Normalized enrichment scores (NES) for glioma cell state-specific gene signatures comparing the vehicle-treated and RSL3-treated TS543 glioma neurospheres treated for one (orange) or three (green) days showing consistent depletion of the astrocyte-like cell state. (C) Same as (B). but for the mouse "N1IC_up" astrocyte-like gene signature showing consistent depletion for both treatment times. (D) Heatmap showing differential expression fold-changes comparing vehicle- and RSL3-treated TS543 neurospheres at one and three days for markers of the astrocyte-like signature. (E) Same as (A). but for mitochondrial metabolic gene signatures showing consistent depletion for both treatment times and significant depletion of ETC, oxidative phosphorylation, and aerobic respiration at day 3. For all panels, * indicates FDR < 0.05. Source data are available online for this figure.

including GSH/GSSG ratio, NAD+ or NADP+ levels (Appendix Fig. S9c). Overall, these findings demonstrate the significant limitations of cell culture models in studying cell state-specific sensitivity to GPX4 inhibition in glioma.

To better understand the lack of correlation between Notch signaling and RSL3 sensitivity in vitro we overexpressed N1IC in the TS543 neurospheres and performed bulk RNA-seq. Surprisingly, we found that activating Notch in a transformed cell line was insufficient to guide the TS543 neurospheres towards the AC-like phenotype, instead inducing MES transformation (Appendix Fig. S9d, Dataset EV8). Furthermore, Notch overexpression also activated different metabolic programs, with inhibition of the mitochondrial ETC and oxidative phosphorylation signatures encountered in the N1IC model (Appendix Fig. S9e, Dataset EV8). These findings demonstrate that timing and context of Notch activation are essential in inducing an astrocytic cell fate. Lastly, we tested if baseline Notch signaling controls susceptibility to ferroptosis. Genetic inhibition of the canonical Notch pathway using a dominant negative MAML construct did not induce

changes either in cell state markers (Appendix Fig. S9f) or in response to ferroptosis inducers, including GPX4 and SLC7A11 inhibitors (Appendix Fig. S9g–j). These experiments further demonstrate that the AC-like phenotype and not Notch signaling per se controls response to ferroptosis.

Finally, we generated xenograft (PDX) mouse models by intracranial injection of TS543 glioma neurospheres and profiled multiple PDX replicates by scRNA-seq at 21 days post-injection (Appendix Fig. S10a). As expected, the largest population of cells in the tumor tissue was comprised of TS543 human glioma cells, which were easily discernable from the murine cells in the microenvironment (Appendix Fig. S10b). When we computationally isolated the glioma cells and performed unsupervised clustering, we identified a minute but discrete cluster consistent with an astrocyte-like population, which was consistently represented across replicates based on GSEA of the various glioma cell state signatures (Appendix Fig. S10c, d). The frequency of this subpopulation among tumor cells was 0.1–0.5% across replicates and 0.08–0.4% among total cells. Furthermore, we observed more

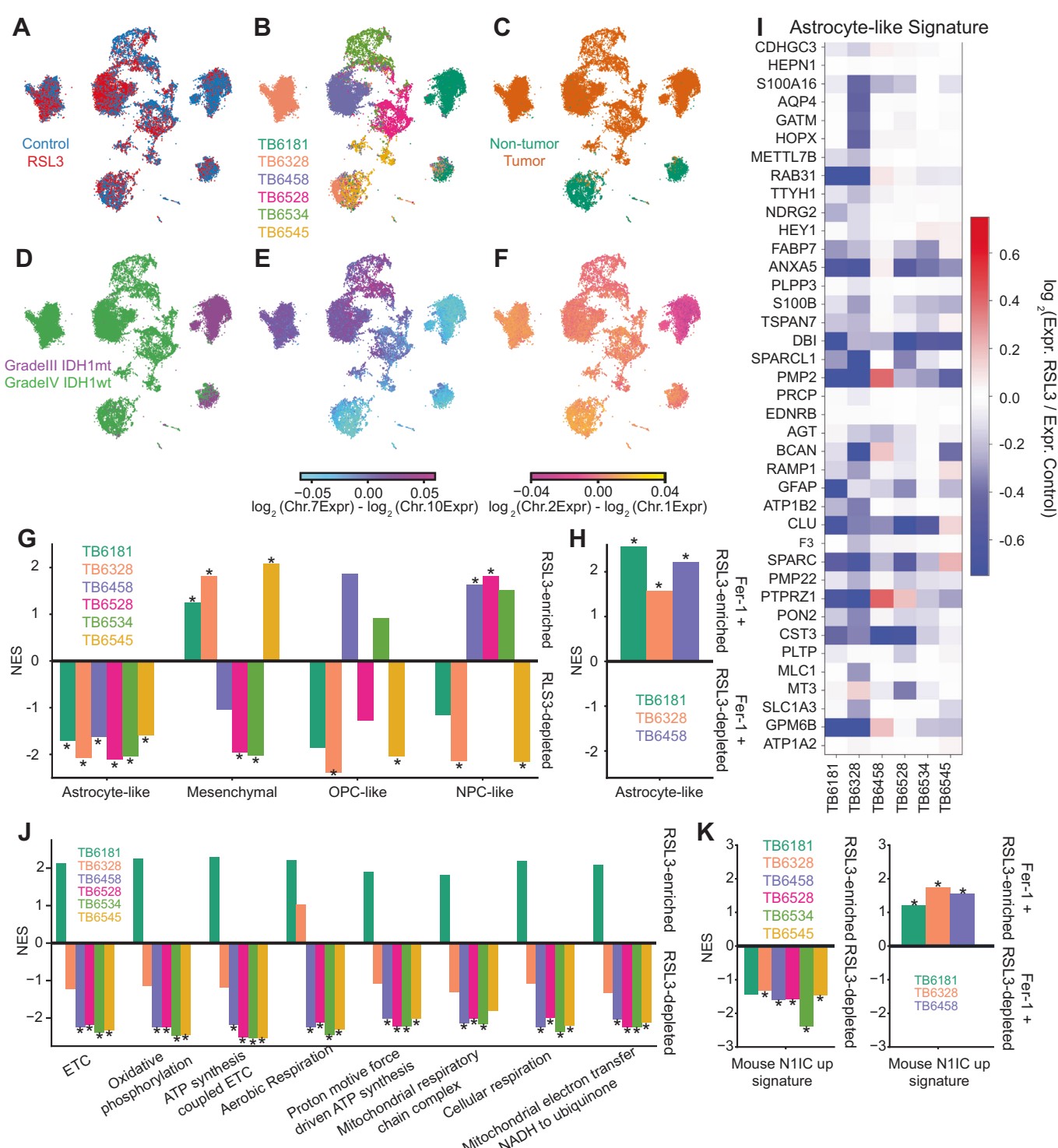

pervasive expression of the proliferative and mesenchymal signature genes across multiple clusters (Appendix Fig. S10e, f). These results are consistent with our expectation that the quiescent astrocyte-like state is replaced by proliferative or mesenchymal phenotypes at later time points in model systems that require extensive expansion of cells due to selection and transformation.

Thus, in vitro Notch overexpression or PDX models are unsuitable for testing our hypothesis that RSL3 targets the astrocyte-like state.

Given the limitations of patient derived cell models, to further test the translational impact of our findings we applied our recently reported approach of treating acute slice cultures of human glioma surgical specimens with drugs and deconvolving cell type-specific

**Figure 7. RSL3 targets quiescent AC-like transformed cell populations in acute slice cultures from human glioma.**

(A–F) UMAP embedding of scRNA-seq data from vehicle- and RSL3-treated slice cultures from six gliomas, including five primary GBMs and one primary IDH1 mutant adult-type diffuse glioma. Panel (A) shows the cells annotated by treatment; control (Blue), RSL3 (Red). Panel (B) shows the cells annotated by tumor. Panel (C) shows cells annotated as non-tumor (green) or tumor (orange). Panel (D) shows cells annotated by IDHmt (purple) or IDHwt (green). Panels (E) and (F) show cells annotated by chromosomal copy number alterations. Also refer to Appendix Fig. S11. (G) Normalized enrichment scores (NES) for glioma cell state-specific gene signatures comparing the vehicle treated vs. RSL3 treated slices for all 6 cases. (H) NES for the astrocyte-like gene signature, comparing RSL3 treated vs. RSL3+Ferrostatin-1 treated slices for 3 of the 6 cases. (I) Heatmap of the fold-change for each gene in the astrocyte-like gene signature across all six patients comparing the vehicle and RSL3 treated slices. (J) GSEA of mitochondrial metabolic signatures comparing the transformed glioma cells in the vehicle and RSL3 treated slices. (K) Normalized enrichment scores for the "N1IC_up" gene signature derived from the murine model comparing the vehicle treated vs. RSL3 treated slices for all 6 cases. Right panel depicts the same gene signature comparing RSL3 treated vs. RSL3+Ferrostatin-1 treated slices for 3 of the 6 cases. Panels (G), (H), (J), (K): Significant (FDR-corrected $p < 0.05$) NES marked with asterisk (*). Source data are available online for this figure.

responses with scRNA-seq (Zhao et al, 2021). This model preserves intratumoral heterogeneity with the diversity of neoplastic cell states, including the quiescent AC-like glioma cell population. We performed either DMSO (vehicle) or RSL3 treatments for 18 h on organotypic slice cultures from six gliomas, including five primary GBMs and an IDH1-mutant anaplastic astrocytoma (Fig. 7A–F, Table EV2). Cells from the RSL3-treated slice cultures co-clustered with subsets of cells from vehicle-treated slices, suggesting that RSL3 selectively depletes specific subpopulations (Fig. 7A). We used aneuploidies or other large copy number alterations, which cause significant increases or decreases in the relative expression of genes on amplified or deleted chromosomes, to identify transformed glioma cells (Yuan et al, 2018) (Fig. 7E,F, Methods). For the five GBMs we used the relative expression of Chr. 7 (amplified) and Chr. 10 (deleted) (Fig. 7E), while for the IDH1 mutant glioma we used Chr. 1 (deleted) and Chr. 2 (unaltered) (Fig. 7F). This analysis allowed us to perform differential expression analysis between the vehicle- and RSL3-treated slices specifically for the transformed glioma cells from each patient (Appendix Fig. S11). Using the gene signatures for the four-state model of glioma cell phenotype (Neftel et al, 2019), we performed GSEA to show that the astrocyte-like glioma cell state was depleted across all six patient samples (Fig. 7G), consistent with our findings in the animal model and the early passage patient-derived cell line. Interestingly, this effect was independent of patient's sex, MGMT methylation, p53 mutations or EGFR status. p53 has been directly linked to ferroptosis susceptibility in other cancer models (Chu et al, 2019; Jiang et al, 2015). As we and others have shown, this astrocyte-like glioma cell subpopulation is largely quiescent and resistant to conventional anti-proliferative therapies (Zhao et al, 2021). For three of the six patients, we also co-treated slice cultures with RSL3 and Ferrostatin-1, ferroptosis inhibitor which we would expect to reverse the effects of the GPX4 inhibitor. Indeed, differential expression analysis and GSEA show an enrichment of the astrocyte-like signature in the transformed cells that were co-treated relative to those treated with RSL3 alone (Fig. 7H). These results suggest that RSL3 depletion of astrocyte-like glioma cells occurs primarily through ferroptosis. Notably, RSL3 induced widespread depletion of the astrocyte-like signature in transformed glioma cells across patient samples, including several genes with important metabolic functions, such as *FABP7* (fatty acid binding protein 7), involved in fatty acid and mitochondrial metabolism, *DBI* (Diazepam Binding Inhibitor, Acyl-CoA Binding Protein) involved in lipid metabolism, and *GPM6B* (glycoprotein M6B), membrane proteolipid with important roles in membrane trafficking (Fig. 7I, Dataset EV9).

While these results are highly concordant with our findings in the N1IC glioma cells, which harbor an astrocyte-like phenotype and are highly sensitive to RSL3-induced ferroptosis, we wanted to know whether the RSL3-sensitive cells in human gliomas bore a metabolic resemblance to N1IC glioma cells. Thus, we performed GSEA on the differential expression analysis comparing the transformed glioma cells in the vehicle- and RSL3-treated slices using mitochondrial metabolic signatures that were enriched in the N1IC model relative to the p53 model (Figs. 7J and 2B). Indeed, signatures related to mitochondrial respiration and oxidative phosphorylation were depleted by RSL3 treatment in five of the six gliomas. Interestingly, the IDH1 mutant glioma was the tumor that did not conform to this trend, suggesting that it has a distinct metabolic profile from the primary GBMs, but still harbors an astrocyte-like subpopulation that is selectively vulnerable to ferroptosis. Lastly, we found that RSL3 depletes the "N1IC_up" gene signature in all patient samples except for the IDH1-mutant astrocytoma, an effect which is rescued with addition of Ferrostatin-1 (Fig. 7K). These findings demonstrate that the murine derived N1IC gene signature can be used to label ferroptosis responsive tumor cell populations in human IDH1-wt GBM. This signature could be used as biomarker for patient selection and response monitoring in future clinical trials employing GPX4 inhibitors. Taken together, these experiments support the potential clinical applicability of GPX4 inhibitors to eradicate quiescent astrocyte-like tumor cells in human GBM by targeting a mitochondrial vulnerability of this persister-like population which is resistant to standard of care anti-proliferative therapies.

## Discussion

### Effects of Notch signaling are time and context-dependent

Using a combination of genetically engineered mouse glioma models, transcriptomic, functional, and metabolomic studies, in vitro studies using patient-derived cell lines and ex vivo studies with acute slice cultures generated from patient-derived GBM samples, we show that Notch activation at tumor inception can induce a quiescent astrocyte-like persister glioma phenotype with a metabolic profile characterized by alterations in mitochondrial function leading to ROS generation and induction of lipid peroxidation with increased sensitivity to GPX4-driven ferroptosis. Our results are consistent with analysis of human GBM samples

showing high Notch signaling in a subpopulation of slowly cycling glioma cells, present both in treatment naïve samples and enriched after exposure to cytotoxic therapies (Liau et al, 2017). The importance of these findings extends beyond the specifics of Notch signaling in glioma cells and reveals a link between a quiescent AC-like glioma state, cellular metabolism, and therapeutic vulnerability. Notably, we found that the AC-like cell state is lost with serial passaging in patient derived cell models and xenografts, due to selection for more proliferative populations and due to transformation towards a mesenchymal phenotype. Therefore, we opted to use Notch activation in a genetically engineered mouse model as a method to induce astrocytic fate at tumor inception. Notch activation in the AC-like murine model induced upregulation of numerous mitochondrial complex I genes as well as compensatory upregulation of genes involved in FAO and the TCA cycle. Intriguingly, activation of Notch signaling in patient derived transformed human glioma cells did not recapitulate the transcriptional mitochondrial programs seen in the murine N1IC model and the AC-like populations in human GBM samples. Furthermore, Notch activation in transformed cells led to mesenchymal transformation rather than guiding glioma cells along an astrocytic cell fate. Timing of developmental program activation is therefore of critical importance in guiding cell fate trajectory and accompanying metabolic programs. Importantly, inhibition of baseline Notch signaling in fully transformed human cell lines did not alter sensitivity to GPX4 inhibition, further supporting the conclusion that Notch activation at tumor inception forces an astrocytic developmental trajectory with altered mitochondrial activity but Notch activation later in tumor development is not sufficient to induce similar cell states, metabolic programs, or therapeutic vulnerabilities. In line with these mechanistic studies, we did not find a direct correlation between Notch signaling and response to GPX4 inhibitors in human cell lines. Together these findings indicate that the astrocytic cell state with its associated mitochondrial dysfunction rather than Notch signaling per se is the main drivers of vulnerability to GPX4 inhibition. Importantly, while Notch signaling is not predictive of response to ferroptosis, the N1IC metabolic gene signature derived from the genetic murine model specifically labels ferroptosis-responsive populations in human glioma specimens.

## The clinical and therapeutic implications of cellular quiescence are context-specific

Genes that promote a quiescent phenotype can slow tumor growth and prolong survival, such as in the N1IC mouse glioma model. Genetically deleting Notch induces faster tumor growth and significantly shortens survival (Giachino et al, 2015). Thus, Notch signaling can function as a tumor suppressor in mouse models of gliomagenesis. In our murine model, high constitutive Notch signaling in end-stage N1IC tumors refutes a possible selection process of N1IC inactive clones as mechanism for driving tumor formation. Therefore, rather than acting as a pure tumor suppressor, Notch activation in our model leads to induction of a quiescent astrocyte-like cell state which recapitulates glioma cell populations present both in primary and recurrent GBM. Moreover, previous studies have shown that Notch-dependent quiescent glioma cells are resistant to cytotoxic/anti-proliferative therapies (Eyler et al, 2020; Jung et al, 2021; Liau et al, 2017), and analysis of

patient samples has also shown that quiescent cells are insensitive to drugs that preferentially target proliferating cells (Zhao et al, 2021; Spinazzi et al, 2022). Quiescent cells therefore likely contribute to resistance and tumor recurrence in GBM following treatments that target proliferating populations (Couturier et al, 2020; Hoang-Minh et al, 2018; Xie et al, 2022). Furthermore, treatment may also induce cells to acquire quiescent/senescent phenotypes that are distinct from the quiescent astrocyte-like glioma cells arising during gliomagenesis without therapeutic pressure. Most studies have considered quiescence through the conceptual framework of glioma stem cells. In contrast, our results highlight that quiescent glioma cells can have an astrocyte-like phenotype that is enriched in lineage restricted genes. It has been proposed that this is a more differentiated phenotype with relatively limited tumor propagation potential (Suva and Tirosh, 2020). However, analysis of the TCGA database revealed significantly shorter overall survival for patients whose tumors samples are enriched for the astrocyte-like signature (Levitin et al, 2019). Furthermore, scRNA-seq analysis of MRI-localized biopsies showed that astrocyte-like glioma cells are relatively abundant at the infiltrated margin, where they may escape surgical resection (Levitin et al, 2019), and due to their quiescent state may be resistant to standard of care radiation and chemotherapy. Importantly, a subset of quiescent tumor cells is capable of reentering the cell cycle and repopulating the proliferating cell pool, thereby reconstituting the heterogeneous cellular architecture of the tumor after cytotoxic treatment (Oren et al, 2021; Xie et al, 2022). Recently, AC-like cells have been phylogenetically linked to the mesenchymal cell state, which drives an aggressive glioma phenotype with resistance to standard therapeutic strategies (Chaligne et al, 2021; Schiffman et al, 2023). Together, these findings highlight the need to develop more effective ways to target quiescent AC-like glioma cells.

## The link between glioma cell state and metabolic programs remains an important question in GBM

The metabolic state is another key determinant of tumor behavior and resistance to therapy (Nguyen et al, 2021; Rabe et al, 2020; Torrini et al, 2022; Watson et al, 2023). One recent study proposed a functional dichotomy in primary GBM, with glioma cells following either a metabolic or a neurodevelopmental axis during gliomagenesis (Garofano et al, 2021). Our results further refine the link between transcriptional cell states and highly specialized metabolic programs. In addition to the enrichment in transcriptional signatures of oxidative phosphorylation and ETC in the N1IC glioma model, several genes associated with the Notch-induced astrocytic-like phenotype have metabolic implications. Notably, two genes upregulated in N1IC transformed cell populations, *Fabp7* and *Tspo*, have been implicated in alterations of mitochondrial function. Fabp7 is highly expressed in astrocytes with damaged mitochondria (Killoy et al, 2020) while Tspo is a translocator protein localized to the outer mitochondrial membrane that regulates mitochondrial respiration and generation of ROS by opening the mitochondrial permeability transition pore, which induces ROS induced ROS release (Daugherty et al, 2016; Milenkovic et al, 2019; Rupprecht et al, 2010). We show here that inhibiting Tspo with PK11195 significantly reduced mitochondrial lipid peroxidation and RSL3 induced cell death. Our metabolomic

and functional analyses highlighted metabolic differences between N1IC and p53 cells. Specifically, we found higher glycolysis in the rapidly proliferating p53 model, in line with several recent studies (Garofano et al, 2021; Hoang-Minh et al, 2018). In contrast, N1IC glioma cells showed altered mitochondrial activity at complex I with decreased OCR, increased ROS and lipid peroxidation. These results are consistent with recent studies demonstrating that complex I dysfunction in glioblastoma cells can result in a similar metabolic phenotype, characterized by reduced OCR and increased ROS production (Noch et al, 2024; Puca et al, 2021), and that ROS produced by complex I dysfunction causes lipid peroxidation and ferroptosis (Lyamzaev et al, 2023; Qiu et al, 2024). Notably, noncanonical Notch activity has been recently shown to directly modulate complex I function in the electron respiratory chain altering electron flow in the ETC (Ojha et al, 2022). Glioma cells are also capable of metabolic adaptation by acquiring mitochondria from the tumor microenvironment. This, in turn, increases OCR, ATP production, glutathione levels and activates cellular proliferation (Watson et al, 2023). Therefore, cellular phenotype and mitochondrial function are intimately linked. Intriguingly, the N1IC-driven quiescent AC-like model exhibited lower mitochondrial numbers and altered mitochondrial function, with decreased FAO, decreased biosynthesis, decreased OCR and decreased mitochondrial membrane potential, possibly due to ROS-induced damage. In turn, transcriptional programs regulating complex I biogenesis, FAO, TCA and the ETC along with redox protective programs were highly upregulated as compensatory mechanisms. As shown in other cancer models (Noch et al, 2024; Weiss-Sadan et al, 2023), the increased NADH/NAD ratio resulting from altered complex I activity in the N1IC model led to redox imbalance with increased ROS production and mitochondrial lipid peroxidation. In our model, the profound redox imbalance in the N1IC AC-like population in conjunction with ineffective cysteine/methionine metabolism leads to depletion of GSH stores, increasing vulnerability to ferroptosis. Lastly, lipidomic signatures demonstrated enrichment of glycerolipids in the AC-like N1IC model, mirroring the lipid profile recently described in astrocytes (Fitzner et al, 2020) as well as profound depletion of PUFA containing phospholipids, likely secondary to baseline lipid peroxidation and ferroptosis. In turn, this altered metabolic phenotype led to high dependency on GPX4 activity in astrocyte-like glioma populations. Modulating mitochondrial lipid peroxidation at complex I, but not at complex III, directly impacts response to GPX4 inhibitors. Thus, our studies suggest a mechanistic link between the altered mitochondrial state at complex I in the AC-like cell state and susceptibility to ferroptosis. Notably, another recent study showed that AC-like tumor cells in diffuse intrinsic pontine glioma have a metabolic vulnerability to ferroptosis (Mbah et al, 2022). Overall, similar to CNS development, ROS may play important roles in modulating the link between cell state and metabolic programs in glioma, thereby uncovering potential cell state-specific therapeutic liabilities. Further studies are needed to investigate this link.

## Developing therapeutic strategies based on cell-state-specific vulnerabilities remains an important goal in oncology

Tumor cell populations with specific metabolic affinities and potentially unique therapeutic vulnerabilities have been recently

identified in glioma (Garofano et al, 2021; Hoang-Minh et al, 2018; Rusu et al, 2019). In line with these findings, we show that chronic intratumoral topotecan leads to expansion of a quiescent AC-like tumor population enriched in the N1IC metabolic gene signature. In other solid cancers, treatment-resistant persister-like cells demonstrate acquired GPX4 dependency and specific sensitivity to ferroptotic cell death (Hangauer et al, 2017; Oren et al, 2021; Viswanathan et al, 2017). We found that the altered mitochondrial phenotype renders the N1IC cells more sensitive to GPX4 inhibitors and ferroptotic cell death (Yang et al, 2014). Intriguingly, modulating the electron flow in the ETC through complex I directly affected sensitivity to RSL3, a GPX4 inhibitor. In other cancer models, inhibition of complex I attenuated sensitivity to cysteine-deprivation but did not affect GPX4-driven ferroptosis (Gao et al, 2019), suggesting a highly cell type specific and context-dependent effect. Remarkably, treating early passage patient-derived cell lines and acute organotypic slice cultures of GBM tissue generated from surgical samples with RSL3 consistently depleted the quiescent AC-like glioma cell populations. Furthermore, RSL3 treatment caused a significant depletion in genes associated with mitochondrial respiration and oxidative phosphorylation in all five IDH-wild type tumors and a patient-derived cell line. Together these findings reveal a link between the quiescent AC-like glioma state and a specific metabolic/therapeutic vulnerability that translates from the N1IC mouse glioma model to human IDH-wild type GBM. Notably, the IDH-mutant glioma we tested also contained quiescent AC-like glioma cells that were selectively depleted by RSL3, although effects on the metabolic signatures were different, suggesting that sensitivity to ferroptosis is mechanistically distinct in IDH-mutant gliomas, possibly related to the effects of the IDH-mutation on cellular metabolism (Karpel-Massler et al, 2017; Fack et al, 2017). Lastly, the dependency on GPX4 yet not SLC7A11 in both mouse models may suggest that glioma cells obtain cysteine independent of the system $x_c^-$ antiporter, such as the transsulfuration pathway, while still depending on robust GPX4 activity to eliminate lethal concentrations of lipid peroxides. Notably, GPX4 has been shown to modulate peroxide-induced ferroptosis through thiol-independent selenolate-based catalysis in some cell types of the developing brain (Ingold et al, 2018). The roles of GPX4-independent mechanisms in governing ferroptosis susceptibility of various glioma cell states need further investigation.

## Our translational studies also address the need for models that recapitulate the astrocyte-like quiescent glioma population

One major barrier to developing new treatments that can effectively target the quiescent AC-like tumor cells is the lack of experimental models that recapitulate this phenotypic state. Both in vitro glioma cell culture models and in vivo cell transplantation glioma models commonly used to test drugs and other anti-glioma therapies are predominantly composed of proliferating tumor cells, and therefore are not well suited to study quiescent glioma cells (LeBlanc et al, 2022; Pine et al, 2020). In our previous studies, we showed that treating acute GBM slices with the topoisomerase poison etoposide selectively depletes subpopulations of proliferating glioma cells, consistent with its known mechanism of action (Hande, 1998; Zhao et al, 2021). Our results in the current study show that RSL3 can target quiescent astrocyte-like glioma cells, which are relatively resistant to etoposide

(Zhao et al, 2021). Future studies could use the slice culture model to test if combinations of drugs, such as Etoposide and RSL3, can more effectively eliminate both proliferating and quiescent glioma cell populations in order to obtain a durable response in GBM. Such models and platforms will be critical in testing personalized therapeutic strategies to effectively target a variety of cell states in each patient's tumor.

# Methods

## Murine glioma models

### Animals

All procedures were reviewed and approved by the Columbia University Institutional Animal Care and Use Committee (IACUC). C57BL/6 male and female mice were used as background for all experiments, including genetic and cell transplantation models. Wild-type C57BL/6 mice were obtained from Jackson Laboratories (strain #000664). Mice harboring a stop-flox N1IC construct (Buonamici et al, 2009) were crossed with $TP53^{fl/fl}$ mice (Lei et al, 2011). The resulting mouse line was maintained by breeding of $N1IC^+/TP53^{fl/fl}$ male mice with $TP53^{fl/fl}$ female mice. $N1IC^+/TP53^{fl/fl}$ mice were further crossed with $TP53^{fl/fl}$/mCherry-Luciferase$^{stop\text{-}flox}$ mice (Montgomery et al, 2020) to generate $N1IC^+/TP53^{fl/fl}$/mCherry-Luciferase$^+$ mice. Mice were bred in house and housed under standard conditions in pathogen-free facilities at Columbia University Medical Center. Correct expression of transgenes was confirmed by genotyping from genomic DNA.

### Viral and orthotopic cell injections

Mice of both sexes (6–8 weeks old) were used for viral injections. 1 μL of PDGF-BB – IRES – Cre retrovirus ($10^6$/mL titer) was injected into the subcortical white matter at the following coordinates (with bregma as reference): 2.2 mm lateral, 2.2 mm rostral, and 2 mm deep, at a flow rate of 0.33 μL/min. Retrovirally induced tumors from 2 male $N1IC^+/TP53^{fl/fl}$ and 2 male $TP53^{fl/fl}$ were dissected at end-stage for single cell RNA sequencing (scRNA-seq) analysis, as described below. For orthotopic cell transplantation experiments, 6-week-old C57BL/6 male mice were injected with 100,000 cells (p53-1 or N1IC-1, passage 10, isolated from littermates) resuspended in 2 μL PBS at the following coordinates (with bregma as reference): 2.5 mm lateral, 2 mm rostral, and 2.5 mm deep, and at a flow rate of 0.25 μL/min. Mice were sacrificed either at 30 days post injection or when endstage criteria were met, as indicated in the corresponding Figures.

For the human orthotopic injection model, TS543 cells ($10^5$ cells in 2 μL, passage 8) were injected in 8-week-old female nude mice (CrTac:NCr-$Foxn1^{nu}$, Taconic) using the same co-ordinates as above. At day 21 post-injection, tissue was harvested for scRNA-seq analysis.

## Monitoring of tumor growth

Tumor growth in mCherry-Luciferase$^+$ mice was monitored by bioluminescent imaging on an IVIS Spectrum Optical Imaging System (Caliper) once weekly as previously described (Lei et al, 2011). Growth of some tumors was also monitored using MRI. MR Images were obtained on a Bruker BioSpec 9.4T Magnetic Resonance Imager (Bruker Corp., Billerica, MA). The mice were anesthetized with 1–2% isofluorane mixed with medical air via a nose cone. The concentration of the isoflurane was adjusted during the procedure to maintain the respiration rate in the range of 40–70 breaths/min using a respiration pillow attached to a monitoring system (SA Instruments, Stony Brook, NY). Body temperature was maintained at 37 °C using a flowing water heating pad. Low-resolution T1 weighted scout images were obtained initially for localization of the brain. A T2 rapid acquisition with relaxation enhancement (RARE) sequence was used with the following parameters: repetition time (TR) = 3000 ms, echo time (TE) = 45 ms, field of view = 17 × 15 mm, matrix size = 225 × 198, slice thickness = 0.7 mm with 16 slices to cover the entire brain.

## Cell line isolation and culture

Primary murine glioma cells were isolated from virally induced end-stage tumors. Tumor tissue from the right hemisphere was minced and enzymatically dissociated in a cocktail of TrypLE and DNase at 37 °C in a shaking water bath for 5 min. Tissue was triturated by passing through a flame-polished glass Pasteur pipette, filtered through a 70 μm mesh filter and neutralized with 50% FBS. Cells were collected by centrifugation and resuspended in BFP media, composed of DMEM, 10 ng/mL PDGF-AA (Peprotech), 10 ng/mL bFGF (Peprotech), N2 supplement (Gibco), antibiotic/anti-mycotic and 0.5% FBS. Cells were propagated in poly-L-lysine (PLL) coated tissue culture dishes in a humidified incubator (5% $CO_2$) at 37 °C. Studies were performed in litter and viral batch matched pairs. Cells were regularly checked for mycoplasma.

Patient-derived TS543 cells were generously donated by Dr. Cameron Brennan (Memorial Sloan Kettering Cancer Center). TS543 were grown as neurospheres in NeuroCult™ NS-A Proliferation Kit (Human) supplemented with human recombinant bFGF (10 ng/ml), EGF (20 ng/ml), and heparin (2 μg/ml). All reagents were obtained from Stem Cell Technologies. TS543 passage 15 or lower were used for all experiments. In addition, high passage (P19-26) TS543 were used for scRNA-seq analysis to dissect presence of various glioma cell states over time.

293T cells were grown on PLL-coated dishes in DMEM supplemented with 10% FBS. For lentivirus production, 5 × $10^6$ 293T cells in 10 cm dishes were transfected with 3.5 μg pMD2.G (VSVg; Addgene #12259), 5.9 μg pMDLg/pRRE (gag/pol; Addgene #12251), 2.9 μg pRSV-Rev (Addgene #12253) and 11.7 μg lentiviral plasmid (pCCL.pgk.wpre vector encoding GFP, N1IC-GFP or dnMAML-GFP) using Lipofectamine-3000 as recommended by the manufacturer (ThermoFisher Scientific). Isolated virus was used to infect TS543 cells in the presence of sequabrene (Millipore Sigma; 8 μg/ml). Cell infection was confirmed via expression of EGFP.

## Ferroptosis compounds and in vitro screening

Drug screens were performed in 384-well plates (Greiner Bio-One 781075) pre-coated with 10 μg/mL PLL for 60 min followed by 3× washes in PBS. Cells were plated at a density of 1000 cells/well in 50 μL BFP medium and incubated overnight. Drugs diluted in DMSO were dispensed at the indicated concentrations using a HP Digital Dispenser. Cells were incubated for 24 h (RSL3, ML162, IKE, STS) or 48 h (ML210), where indicated, IACS-010759 (2 μM),

Antimycin (0.5 μM), Oligomycin (2 μM), FCCP (2 μM), mitoquinone (0.25 μM), SKQ1 (0.1 μM), Ferrostatin (2 μM), Z-VAD-FMK (50 μM), Nec-1 (10 μM) were also delivered to the cells. For sequential drug treatments, cells were plated overnight as above followed by delivery of PK11195 (10 μM), IACS-010759 (2 μM) or FCCP (2 μM) for 24 h. Following that, RSL3 was given for an additional 24 h. For drug combinations, wells were normalized by volume of DMSO added. The plates were subsequently allowed to equilibrate to room temperature for 30 min and 25 μL of Cytotox Glo solution (Promega) was added to the wells using a Wellmate plate filler. Plates were read in a Tecan Infinite F200 multi-label plate reader. Data were exported to Microsoft Excel and GraphPad Prism for further analysis.

## Flow cytometry staining and analysis

N1IC and p53 cells were seeded in 6-well plates at a density of $25 \times 10^4$ cells/well. After an 18-h incubation period, cells were labeled with 100 nM Mito Tracker Green FM, or 5 μM CM-H2DCFDA, or 2 μM BODIPY-C11 or 1.5 μM JC-1 (all from ThermoFisher Scientific) or 100 nM MitoCLOX (Lumiprobe) reagents dissolved in BFP media for 30 min. Cells were subsequently washed once with PBS, lifted with 10 mM EDTA in PBS (pH 7.4), and centrifuged for 5 min at $400 \times g$. The cell pellet was resuspended in flow buffer (0.1% w/v BSA, 1 μg/ml DAPI in PBS) and filtered into polystyrene flow tubes. Flow cytometry was performed on a BD LSRFortessa Cell Analyzer using Pacific Blue, FITC and PE or PE-Texas Red filters. Data were processed using FlowJo v10. Cells were first gated by forward and side scatter areas (FSC-A and SSC-A) to remove debris, followed by filtering of multiplets using forward scatter area and height (FSC-A and FSC-H). Dead cells were filtered out by gating on low DAPI (Pacific Blue channel). Mito Tracker Green FM and CM-H2DCFDA signals were captured on the FITC channel. BODIPY-C11 and MitoCLOX data were expressed as ratio of the oxidized (FITC) over reduced (PE-Texas Red) species. JC-1 was expressed as the ratio of the J1 aggregate (PE) over the monomeric form (FITC) of the dye. Resulting data were plotted as a density distribution and gated at the median of the p53 condition. The percentage of live cells above or below this threshold was recorded (e.g., 50% for p53 cells) from multiple independent experiments ($n \geq 3$) and plotted using Prism.

## Drug treatments and effects on ROS and lipid peroxidation

For analysis of drug effects by flow cytometry, cells were plated in 6-well plates at a density of $25 \times 10^4$ cells/well in BFP medium and incubated overnight. Cells were treated with DMSO (control), FCCP (2 μM), IACS-010759 (2 μM), mitoquinone (250 nM), Ferrostatin-1 (10 μM), deferoxamine (150 μM) or PK11195 (10 μM) for 24 h. Cells were subsequently lifted from the plates, resuspended in PBS and divided into aliquots that were stained with BODIPY C11, MitoCLOX or CM-H2DCFDA for 15 min in a humidified tissue culture incubator (37 °C, 5% $CO_2$). Cells were washed once with cold PBS, centrifuged, resuspended in flow buffer and analyzed by flow cytometry as detailed above. Resulting data were plotted as a density distribution and gated at the median of the DMSO-treated condition. The percentage of live cells above or

below this threshold was recorded from multiple independent experiments ($n \geq 4$) and plotted using Prism.

## GPX4 knockdown by siRNA

N1IC or P53 cells were plated in poly-L-lysine-coated 24-well plates ($6.2 \times 10^4$ cells/well) for 18 h in BFP media with or without 2 μM Ferrostatin. Cells were transfected with either control (ON-TARGETplus Non-targeting control siRNA, catalog number D-01810-01-05; Horizon Discovery) or Gpx4-targeting siRNA (ON-TARGET plus Mouse Gpx4, catalog number L-072959-01-0005; Horizon Discovery). For transfection, siRNAs were delivered to a final concentration of 10 nM using Lipofectamine RNAiMAX transfection reagent (ThermoFisher Scientific) as detailed by the manufacturer. Reduction of GPX4 protein levels was confirmed by western blotting. Phase contrast images of cells in the 24-wells were taken on a Nikon TE2000 microscope (5 independent fields per well) using a 10×Plan Fluor Ph1 objective. Images were exported to FIJI/ImageJ and live cells were counted manually. Subsequently, cells were stained with BODIPY C11 and processed for flow cytometry as detailed above.

## Western blotting

All materials for Western blotting were acquired from Fisher Scientific unless otherwise noted. Cells were lysed in cell extraction buffer (Cat#FNN0011) supplemented with Protease Inhibitor (Cat#78425), 1X Phosphatase Inhibitor Cocktail 3 (Cat#P004-5ml) and Phosphatase Inhibitor Cocktail 2 (Cat#P5726-1ml). Lysates were denatured by boiling in Nupage loading Buffer (Cat#NP0007) and Sample Reducing agent (Cat#NP0004) (10 min, 90 °C) and resolved in Nupage 4–12% gels (Cat#NP0321BOX) using MOPS SDS running buffer (Cat#NP0001). Ladders used were from Licor (Chameleon Duo Pre-Stained Protein Ladder, Cat#928-60000). Resolved proteins were transferred to LI-COR Odyssey nitrocellulose membranes (926-31090) using Nupage transfer buffer (NP0006-1). Membranes were blocked with 5% non-fat dry milk PBS-T solution followed by primary antibody incubation overnight at 4 °C with gentle agitation. Following PBS-T washes (3 × 10 min), membranes were incubated with secondary antibodies (Licor IRDye 680RD Goat Anti Mouse- Cat#925-68070 and Licor IRDye 800CW Goat anti-Rabbit- Cat#926-32211), washed in PBS-T (3 × 10 min) and developed on a Licor Odyssey FC machine with western channel 700 and 800 and exposure time of 2 min. The following primary antibodies were used: anti-β-actin (AC-15; Sigma, cat. No. A5441), anti-GPX4 (Abcam, ab125066), and anti-SLC7A11 (Cell Signaling Technology, 98051). Full Western blots are provided as source data files.

## C¹³ Isotope tracing/flux analysis

p53 or N1IC cells were plated in triplicate in 10 cm dishes ($10^6$ cells/dish) in BFP media and incubated for 24 h. Cells were subsequently cultured in DMEM containing 100 μM (U-$^{13}$C16) palmitic acid (Cambridge Isotope Laboratories, Inc) for 24 h in the presence of 1.5% dialyzed FBS (Thermo Fisher, A3382001). Polar metabolites were extracted, read, and analyzed by the Metabolomics Core Facility at Weil Cornell (New York).

## Liquid chromatography-mass spectrometry (LC-MS/MS) lipidomic profiling and analysis in primary cell cultures

Lipids were extracted using previously described methods (Ye et al, 2020). $2 \times 10^6$ cells (p53 or N1IC of similar passage, $n = 4$ replicates per condition) were grown in 10-cm dishes overnight, as detailed above. Cells were subsequently collected by scraping on dry ice with a microtip homogenizer and 132 µL of solution containing ice-cold methanol and 0.01% (w/v) butylated hydroxyl toluene (BHT) were added to 100 µL of each cell pellet in a glass vial with a Teflon-lined cap. Samples were vortex mixed for 15 s. 668 µL of methyl tert-butyl ether (MTBE) was added and samples were again vortex mixed vigorously for 30 s. Samples were then incubated for 20 min in a cold room at 4 °C on an orbital shaker and incubated for an additional 5 min on ice. The samples then underwent centrifugation at 3000 rpm in a cold room at 4 °C for 15-min for phase separation. The organic top layer was collected in a glass vial, dried under nitrogen stream, and stored at −80 °C until used for analysis. Dried samples were reconstituted in 2-propanol/MeCN/$H_2O$ (4:3:1, v/v/v) for LC-MS analysis. 20 µL of each sample were used as quality control (QC). The protein pellet was used to measure protein concentration for normalization using the colorimetric Bradford assay (ThermoFisher Scientific).

Separation of extracted lipids was done at 55 °C on ACQUITY UPLC HSS C18 Column over a 20-min gradient elution (1.7 mM, 100 × 2.1 mm; Waters). The column was eluted isocratically for 1 min with 80% mobile phase A (MeCN/water 60:40; v/v) with 10 mM ammonium acetate/0.1% acetic acid followed by 2-min linear gradient 40% mobile phase B (2-propranol/MeCN/water 85:10:5; V/V/V) with 10 mM ammonium acetate/0.1% acetic acid followed by 70% at 2 min and 90% at 18 min. The flow rate was set to 400 µL/min and injection volumes were set at 6 µL e. Analysis was performed using a Synapt G2- Q-ToF mass spectrometrer (Water, Manchester, UK) in positive and negative electrospray ionization (ESI). Technical details were as follows: positive mode—+3 kV capillary voltage/32 V sampling cone voltage/source temperature 120 °C/desolvation temperature 500 °C; negative mode—−2 kV capillary voltage/30 V sampling cone voltage/source temperature 120 °C/desolvation gas flow 900 L/h. The traveling wave velocity was set to 650 m/s and wave height was 40 V. The helium gas flow in the helium cell region of the ion-mobility spectrometry cell was set to 180 mL/min to reduce the internal energy of the ions and minimize fragmentation. Nitrogen as the drift gas was held at a flow rate of 90 mL/min in the IMS cell. The low collision energy was set to 4 eV, and high collision energy was ramping from 25 to 65 eV in the transfer region of the T-Wave device to induce fragmentation of mobility separated precursor ions. All raw data were converted to netCDF using the MassLynx software v4.1. XCMS package in R was used for subsequent analysis. The CenWave algorithm was used for peak picking (CenWave algorithm, peak width window 2–25 s) and peak grouping (bandwith 2 s, $m/z$-width 0.01 Da). Peaks with variations larger than 30% in QCs were eliminated.

All the extracted features were normalized to measured protein concentrations measured by Bradford assay. Statistical analysis was performed in MetaboAnalyst. Group differences were calculated using Welch's t-test with FDR-corrected $p$-value < 0.05. Lipid identities were determined based on available online databases such as METLIN, Lipid MAPS, and HMDB and matching with $MS^E$

fragments. Lipid ontology analysis was performed using the LION webtool (lipidontology.com), as previously described (Molenaar et al, 2019). The ranking mode was used for input and lipids were ranked based on $\log_2$FC (Dataset EV4). One-tailed $t$-test was used to assess statistical significance of the comparison.

## Untargeted metabolomics by LC-MS/MS in primary cell cultures

Cells were processed for untargeted metabolomics as described elsewhere (Nguyen et al, 2021). Briefly, $2 \times 10^6$ cells (p53 or N1IC of similar passage) were plated in 10 cm dishes overnight. The next day plates were washed twice with ice-cold PBS, placed on dry ice and 1 mL 100% HPLC grade methanol was added to the dish. Cells were then scraped and transferred to cold Eppendorf tubes centrifuged at $14,000 \times g$ for 20 min at 4 °C and the supernatant was transferred to a fresh Eppendorf tube and dried using SpeedVac. The dried sample was reconstituted in water/acetonitrile (1:1; v/v) before LC-MS analysis. Protein was extracted from the pellets after centrifugation using cell extraction buffer with protease and phosphatase inhibitors. A colorimetric Bradford assay (ThermoFisher Scientific) was read at 740 nm for evaluation of protein content.

LC-MS analyses were performed on a Q Exactive Orbitrap mass spectrometer (ThermoFisher Scientific) coupled to a Vanquish UPLC system (ThermoFisher Scientific). The Q Exactive operated in polarity-switching mode. A Sequant ZIC-HILIC column (2.1 mm i.d. × 150 mm, Merck) was used for separation of metabolites. Flow rate was set at 150 µL/min. Buffers consisted of 100% acetonitrile for mobile B, and 0.1% $NH_4OH$/20 mM $CH_3COONH_4$ in water for mobile A. Gradient ran from 85% to 30% B in 20 min followed by a wash with 30% B and re-equilibration at 85% B. Initial data analysis was done using TraceFinder 4.1 (ThermoFisher Scientific). Metabolites were identified based on exact mass within 5 ppm and matching the retention times with the standards. Relative metabolite quantitation was performed based on peak area for each metabolite. Samples were normalized by protein content using a Bradford assay. MetaboAnalyst version 5.0 (metaboanalyst.ca) was used for differential assessment analysis, partial least square discriminant analysis (PLS-DA), metabolite set enrichment analysis (MSEA) and quantitative pathway analysis. Differentially enriched metabolites in N1IC or p53 cells were identified based on the Volcano plot analysis using the standard $t$-test with a cutoff $p < 0.05$ and no specific cutoff for the $\log_2$FC. MSEA analysis was performed using a KEGG based overrepresentation analysis (Dataset EV3). For joint pathway analysis, we used significant differentially expressed genes with $\log_2$FC > 0 based on the scRNA-seq analysis (Dataset EV1) and differentially enriched metabolites in the N1IC cell line (Dataset EV3). The pathway analysis was performed using a hypergeometric test with degree centrality and combined queries as integration method.

## Global metabolite profile extraction and LC-MS analysis on murine tumors

For metabolic analyses of murine tumors, tissue samples of ~20 mg were measured and hydrated with 400 mL of chilled MilliQ water. Samples were homogenized with Misonix XL-2000 Ultra-liquid

sonicator (Misonix Inc., Farmingdale, NY, USA) at 40 A for 30 s. A 20 µL aliquot was set aside for each sample to perform protein quantification using the Pierce bicichoninic acid assay (BCA, ThermoFisher). 100 µL of 50% MeOH(aq) spiked with qualitative internal standards (qIS), 0.125 µg/mL nitrodracylic acid and 0.125 µg/mL isocaramidine sulfate was added to the extract solution, and then vortexed and incubated on ice for 10 min. Samples were extracted using an adapted Bligh and Dyer biphasic liquid extraction at 2:0.5:1 chloroform/water/methanol. 100 µL of chloroform reagent spiked with qIS, 0.06 µg/mL phenyl-N-pyridinyl acrylamide (PNPA) was added to the extract solution. Samples were vortexed on BenchMixer (Benchmark Scientific, Edison, NJ, USA) at mid-speed (6) for 15 s and incubated on ice for 20 min. The samples were centrifuged at $12,000 \times g$ for 15 min at 4 °C which resulted in two (upper hydrophilic and lower hydrophobic lipid) phases, while discarding the remaining protein layer. Extracts were concentrated under $N_2$ gas vapor stream until complete dryness (~2 h), snap-frozen with dry ice and stored at −80 °C until the time of analysis.

Only LC-MS grade solvents and additives (Covachem, Loves Park, IL, USA) were used to prepare reagents, mobile phases, and wash solutions unless otherwise indicated. Polar extracts were resuspended in 60% methanol (aq) and analyzed on the Agilent 6545 Quadrupole time-of-flight (QToF-MS) with Infinity II 1290 Ultra High-Performance Liquid Chromatography (Agilent Technologies, Wilmington, DE, USA). Global metabolic profiling and relative quantification for steady-state analyses were performed using a HILIC assay specialized for broad detection and high resolution of polar metabolites (energy metabolism and other central carbon metabolites) with the AdvanceBio Glycan Map 2.1 × 100 mm 2.7 µm column with Infinity II in-line filter (Agilent Technologies, Wilmington, DE, USA). Glycan Map HILIC (Agilent Technologies, Wilmington, DE, USA) acquisition was performed in two experiments: positive and negative electrospray ionization (ESI) modes. Compounds were resolved over mobile phase A (10 mM ammonium acetate in 88% water/12% acetonitrile (MeCN), pH 6.85) and mobile phase B (10 mM ammonium acetate in 90% MeCN, pH 6.85) with column temperature 30 °C using flow rate 0.25 mL/min under gradient conditions: 100% B, 0.5 min; 95% B, 2.0 min; 60% B, 3.0 min; 35% B, 5 min; hold 0.25 min; 0% B, 6 min; hold 0.5 min; 100% B, 7.8 min. The mass analyzer parameters included: drying gas temperature, 250 °C; sheath gas temperature, 325 °C; nebulizer, 45 psi; skimmer, 50 V; octopole radio frequency, 750 V; scan rate, 5 spectra/s. For the ESI positive mode experiment, MS spectra were acquired over a voltage gradient of capillary 3000 V, nozzle 2000 V, and fragmentor 125 V. For ESI negative mode experiment, mass spectra were acquired over a voltage gradient of capillary 3000 V, nozzle 1000 V, and fragmentor 90 V. We used wash cycles consisting of strong wash (50% methanol, 25% isopropanol, and 25% water), weak wash (90% acetonitrile and 10% water), and seal wash (10% isopropanol and 90% water) to eliminate carryover between consecutive injections.

Hydrophobic extracts were reconstituted in 100 µL 5:4:1 EtOH/MeOH/water. The lipids were resolved using reverse-phase Acquity CSH 2.5 µm, 2.1 × 100 mm column (Waters Corp) utilizing a gradient composed of mobile phase A, 70:30 Water/MeCN with 5 mM ammonium formate (aq) + 0.1% formic acid (FA); and mobile phase B, 90:10 Isopropanol/MeCN with 5 mM ammonium formate (aq) + 0.1% FA. An isothermal column temperature of

40 °C and static flow rate of 0.200 mL/min was maintained. Real-time mass correction was applied with 0.2 mL/min infusion of external standard (containing TFA/PURINE/HP921) in 95:5 ACN/water. Electrospray injection (ESI) negative (−) ion acquisition was applied with the following MS parameters: injection volume, 8 µL; drying gas temperature (temp), 175 °C; drying gas flow, 8 L/min; nebulizer pressure, 45 psi; sheath gas temp, 350 °C; sheath gas flow, 12 L/min; capillary voltage, 3000 V; nozzle voltage, 25 V; fragmentor, 90 V; skimmer, 50 V; scan rate, 3.0 spectra/s; mass range $m/z$ 100–2850. The following gradient timetable was applied: 0–0.1 min, 1% B; 0.8 min, 5% B, 2 min 35% B; 4 min, 38%; 4.25 min, 40% B; 7 min, 98% B; 7.5 min, 99%; hold, 0.75 min; 8.75 min, 35%; 10.5 min, 90% B; 11.5 min, 38% B; 12.5 min, 1%; equilibrate, 1.5 min. Alternatively, ESI positive ion acquisition applied the following MS parameters: injection volume, 6.5 µL drying gas temperature (temp), 250 °C; gas flow 8 L/min; nebulizer, 45 psig; sheath gas temp, 350 °C; sheath gas flow, 12 L/min; capillary voltage, 3500 V; nozzle voltage, 1000 V; fragmentor, 170 V; skimmer, 50 V; scan rate, 3.5 spectra/s; mass range, $m/z$ 100–2850. During ESI+, following gradient timetable was applied: 0–0.1 min, 1% B; 0.8 min, 5% B, 2 min 35% B; 4.5 min, 38%; 4.75 min, 40% B; 7.5 min, 71% B; 8.5 min, 80.5% B; 9.5 min, 81.5%; 11 min, 98% B; 11.25 min, 99%; 11.25 min, 99%; 12.25 min, 100% B; hold 0.25 min; 13 min, 75% B; 14 min, 90% B; 15 min, 1% B; equilibrate, 1.5 min.

Prior to pre-processing each dataset, the extracted ion (XIC) and total ion (TIC) chromatograms were examined for the pooled quality control (QC) samples to inspect the consistency of retention time and ionization levels throughout. Following the acquisition, mass feature bins were defined by targeted selection using a compound reference library which allowed partitioning of the $m/z$ versus retention time (RT) matrices into fixed width using Agilent Masshunter Profinder B.08.00 (Agilent Technologies, Wilmington, DE, USA). The bins were inspected manually to confirm consistency of the integration for each compound of interest across all samples. Targeted ion selection and alignment parameters were restricted for logical binning of the input data to ion mass range ±2.0 mDa and retention time ±0.4 min. These parameters were used to select and annotate detected neutral mass features based on in-house Personal Compound Data Library (PCDL). Following pre-processing, the ion area for each compound was corrected to the area of sample-dependent qIS followed normalization by tissue-specific protein concentrations (mg/mL).

Data was normalized to the sum and log-transformed for partial least square discriminant analysis (PLS-DA) and orthogonal PLS-DA using the MetaboAnalyst platform version 5.0. Next, we used the unpaired one-tailed parametric $t$-test with equal variance to validate model-specific metabolites. Individual metabolites were selected based on VIP scores from the PLS-DA and OrthoPLS-DA analyses (Dataset EV3). Bar graphs of cohort mean with SEM (represented in error bars) were generated in GraphPad Prism v 10.

## Seahorse analysis of cellular respiration and extracellular acidification

Murine glioma cells (similar low passage N1IC-1/2 and p53-1/2) were seeded in PLL-coated XFe24 cell culture microplates (Agilent TEchnologies) at $1.8 \times 10^4$ cells per well in 250 µL BFP and were allowed to attach overnight. For the cysteine/methionine

deprivation (CMD) experiment, media was aspirated and replaced with either BFP or CMD BFP media after 4 h and cells were grown in respective media for an additional 18 h. Cells were then washed twice in PBS before changing to Seahorse XF base media (Agilent, 102353-100). Mitochondrial stress tests were run with the following concentrations of media: 10 mM glucose, 2 mM glutamine, and 1 mM pyruvate in assay medium, and 2 μM oligomycin, 2 μM trifluoromethoxy carbonylcyanide phenylhydrazone (FCCP), and 0.5 μM rotenone/antimycin A. The assay involved injection of glucose (10 mM), followed by oligomycin (1 μM), followed by 50 mM 2-deoxy-D-glucose. Cells were incubated at 37 °C in a $CO_2$-free atmosphere for 1 h. Oxygen consumption rate (OCR) and extracellular acidification rate (ECAR) were analyzed using the XFe24 Extracellular Flux Analyzer (Agilent, Santa Clara, CA).

## Glutathione measurement

GSH concentration was calculated using the GSH Detection Assay Kit (Abcam; ab138881) as detailed in the manufacturer's protocol. Briefly, $2.5 \times 10^5$ cells (p53 or N1IC of similar passage) were plated per well of a six-well plate and grown for 24 h as detailed above. Cells were washed with cold PBS and lysed with 100 μL ice-cold 1× Mammalian Lysis Buffer. Lysates were centrifuged at $21,000 \times g$ for 15 min. The supernatant was collected, mixed with 20 μL ice-cold trichloroacetic acid, incubated on ice for 10 min and centrifuged at $12,000 \times g$ for 5 min. The supernatant was neutralized by addition of $NaHCO_3$ to a final pH of 6, centrifuged at $13,000 \times g$ for 5 min, and supernatants collected. Samples were corrected for the diluted volumes, mixed with GSH Assay Mixture and GSH was calculated by measuring in a microplate reader (Promega GloMax) using 490/520 nm excitation/emission filter and fitting into a GSH standard curve.

## qRT-PCR

2.5 μg of RNA were used for first-strand cDNA synthesis using the SuperScript Vilo cDNA synthesis kit (ThermoFisher, 11754050). cDNA was diluted to 250 ng/μL and the RT-qPCR reactions were conducted using Thermo Scientific ABsolute Blue qPCR SYBR mix (ThermoFisher, AB4322B). Reactions were performed in triplicate samples per condition on an Applied Biosystems QuantStudio 3 qPCR instrument and all experiments were repeated 3 independent times. Samples were normalized to β-actin, used as reference, and fold change between conditions was calculated using the ΔΔCT method. The following primers against murine transcripts were used (shown in 5' → 3' direction):

 β-actin forward: CGAGGCCCAGAGCAAGAGAG
 β-actin reverse: CTCGTAGATGGGCACAGTGTG
 Slc7a11 forward: GGCACCGTCATCGGATCAG
 Slc7a11 reverse: CTCCACAGGCAGACCAGAAAA
 Chac1 forward: CTGTGGATTTTCGGGTACGG
 Chac1 reverse: CCCTATGGAAGGTGTCTCC
 Ptgs2 forward: TTCAACACACTCTATCACTGGC
 Ptgs2 reverse: AGAAGCGTTTGCGGTACTCAT

## Immunofluorescence and immunocytochemistry

Immunohistochemistry was performed on 4% PFA-fixed, paraffin-embedded sections or OCT-frozen tissue. Paraffin-embedded

sections were first deparaffinized in xylene (35 min) and rehydrated with sequential incubations in 100% ethanol (25 min), 95% ethanol (25 min), and 75% ethanol (15 min). Following a wash in distilled water, antigen retrieval was performed in 10 mM Citrate (pH 6.0) solution in a pressure cooker for 10 min. Slides were allowed to equilibrate to room temperature for 30 min, washed in PBS and incubated in blocking buffer (10% normal goat serum, 0.5% Triton X-100 in PBS) for 30 min at room temperature. For obtaining frozen sections, 4% PFA-fixed tissue was incubated in 30% sucrose solution in PBS followed by immersion and freezing in OCT. Frozen sections (12 μm thickness) were obtained in a cryostat (Leica), mounted on glass slides, and stored at −80 °C. Sections were washed in PBS before incubation in blocking buffer for 30 min at room temperature. For immunocytochemistry, cells were plated at a density of $3 \times 10^4$ cells on 12 mm glass coverslips in 24-well plates. Cells were fixed with 4% PFA for 15 min, permeabilized with 0.5% Triton X-100 in PBS for 5 min and blocked with blocking buffer (10% normal goat serum in PBS) for 30 min. Primary antibody incubations were performed at 4 °C overnight. After three 10-min washes in PBS, sections were incubated with the appropriate Alexa Fluor conjugated antibodies for 1 h. Sections were washed extensively with PBS, incubated with DAPI for 10 min (Thermo Scientific, D1306, 0.5 μg/mL), and mounted using Fluoro-Gel with TES buffer (Electron Microscopy Sciences). The following primary antibodies were used: rat anti-HA (Millipore Sigma; 11867423001), rabbit anti-HA (Covance; PRB101), rabbit anti-Ki67 (Cell Signaling Technology; 9129), goat anti-Clu (ApoJ; Thermo Fisher; PA5-46931), rabbit anti-Top2a (Abcam; ab52934), mouse anti-TfR (ThermoFisher; 13-6800), rabbit anti-N1IC (Abcam; ab8925). All Alexa-Fluor conjugated secondary antibodies were from ThermoFisher Scientific.

### Confocal imaging

Confocal images of fixed samples (1024 × 1024 pixels) were acquired on a Zeiss LSM800 confocal microscope equipped with GaAsP point-detectors, using a Plan-Neofluar 40×/1.3 NA oil DIC objective. Fluorophores were excited using 405 nm, 488 nm, 561 nm, and 639 nm wavelength lasers. Confocal stacks were acquired using 1 AU pinhole size and at 0.58 μm steps. Images were exported to ImageJ for further analysis. A minimum of three fields were obtained from each biological sample.

### Live cell imaging

For time-lapse imaging, $1.3 \times 10^5$ N1IC-1 cells were plated on poly-L-lysine-coated 35 mm glass-bottom dishes (MatTek Corporation) for 24 h. Cells were incubated for 30 min in BFP media containing 2 μM BODIPY-C11. Cells were washed once with warm PBS and media were replaced with fresh BFP. Cells were imaged on a Nikon A1RMP confocal microscope at 37 °C in a humidified chamber with 5% $CO_2$. Time-lapse images (512 × 512 pixels) were acquired using a 40×/1.3 NA oil immersion objective and focus was maintained using the Perfect Focus System. Excitation was achieved using 488 nm and 561 nm laser illumination; emission of the oxidized and reduced forms of BODIPY-C11 was captured using a 525/50 and a 595/50 filter, respectively. At time 0, RSL3 (500 nM) was added and images were acquired every 30 s for a total of 30 min. Images were exported to ImageJ and ratio images of the 488 over the 568 channels were obtained using a pipeline detailed elsewhere (Kardash et al, 2011).

For direct imaging of MitoRed stain in live cells, $1.3 \times 10^4$ p53 or N1IC cells were plated on 35 mm poly-L-lysine-coated glass-bottomed dishes (MatTek Corporation) overnight and labeled with 25 nM MitoTracker Red CMXRos dissolved in BFP for 45 min. Cells were washed once with PBS and incubated with fresh BFP media. Cells were imaged on a Nikon Ti Eclipse spinning disk confocal microscope (Yokogawa CSU-X1) at 37 °C in a humidified chamber with 5% $CO_2$. Images (voxel size: $0.0903 \times 0.0903 \times 0.5$ µm$^3$) were acquired using a Plan Apo TIRF 60×/1.49 NA DIC oil immersion objective and an Andor Xyla 4.2 camera. Excitation was achieved using 561 nm laser illumination and emission was captured using a 595/50 filter. Images were exported to ImageJ. For quantitation of MitoTracker Red signal, fields were background subtracted using a rolling ball radius of 50 pixels, and mean fluorescence intensity of positive pixels was measured for each cell.

## Ex vivo organotypic murine slice experiments

Six-week-old C57BL/6 male mice were injected with 100,000 cells (N1IC-1, passage 10) resuspended in 2 µL PBS at coordinates and rate described above. Mice were sacrificed at 30 days post injection, and tumor-bearing brains were dissected and placed into artificial CSF (aCSF). Murine tumor slices were generated and cultured as previously described (Zhao et al, 2021). Briefly, a McIlwain tissue chopper was used to generate 500 µm coronal sections of the tumor-bearing murine brains, starting at the anterior aspect of visible tumor. Slices were plated on Millicell cell culture inserts (catalog # PICM0RG50) in 6-well dishes and incubated in slice culture media (DMEM/F12, 1% N2 supplement, 1% anti-mycotic/anti-bacterial) with either DMSO control or 500 nM RSL3 for 6 h. After treatment was completed, slices were fixed in 4% PFA, embedded in paraffin, and cut into 5-µm-thick sections.

## Fluorescent in situ hybridization/RNAscope

In situ hybridization of murine Clu and Ptprz1 from paraffin-embedded 5 µm-thick sections was performed using the RNAscope Fluorescent v2 assay (ACDbio), as detailed in the manufacturer's protocol. Of note, we used Ptprz1 as a cell-state agnostic tumor specific marker as it was not differentially expressed between the two murine models (Dataset EV1). Briefly, sections were baked for 1 h at 60 °C, followed by deparaffinization in xylene and 100% ethanol. Slides were dried for 5 min at 60 °C, followed by incubation with $H_2O_2$ for 10 min at room temperature. Antigen retrieval was performed in boiling 1 Target Retrieval reagent for 15 min. Slides were then washed in water, dehydrated in 100% ethanol, and incubated with Protease Plus for 30 min at 40 °C. The C3 probe (Clu) was diluted 1:50 in the C1 probe (Ptprz1) and hybridized on the sections for 2 h at 40 °C. Ptprz1 was detected with TSA-Cy5 and Clu with TSA-Cy3. Nuclei were stained with DAPI and the slides were mounted with Fluoro-Gel with TES buffer (Electron Microscopy Sciences). Images were obtained from Ptprz1 high signal areas to identify tumor. For quantification of RNAscope images, nuclei were first segmented using the Huang intensity threshold followed by a watershed filter. For identification of Clu+ signals, images were thresholded using the "Sambhag white" filter to generate masks. Positive particles were identified using the "Analyze Particles" function and integrated density of Clu from the entire field was obtained. Clu intensity was expressed as weighted

Integrated Density by dividing the sum of Clu integrated densities over the sum of cells per biological sample.

## Drug treatment of patient-derived tissue slices

Approval was obtained from the Columbia University Irving Medical Center Institutional Review Board (IRB) before commencing the study. We strictly adhered to regulations imposed by the IRB. Informed consent was obtained from all participants prior to surgery. Tumor specimens were procured from surgeries at New York Presbyterian Hospital/Columbia University Irving Medical Center. Patient-derived tissue slices were generated and cultured as previously described (Zhao et al, 2021). Deidentified patient diagnosis information can be found in Table EV2. From each individual patient tumor resection, we performed an 18-h drug perturbation with 50 nM RSL3 or drug vehicle (DMSO) on spatially adjacent slices. For three patient tumor resections (TB6181, TB6328, and TB6458), we also co-treated slice cultures with 50 nM RSL3 and 10 µM Ferrostatin-1.

## scRNA-seq analysis

### Tissue dissociation

Murine brain tumors and patient-derived tissue slices generated from TB6181, TB6328, and TB6458 were collected for dissociation using the Adult Brain Dissociation kit (Miltenyi Biotec) according to the manufacturer's instructions. PDX specimen and patient-derived tissue slices generated from TB6528, TB6534, and TB6545 were collected for dissociation using Papain Dissociation System (Worthington) as previously described (Mizrak et al, 2020; Mizrak et al, 2019) with the following modifications. Briefly, minced tissue was digested with papain (Worthington, 10 units per sample) in PIPES solution (120 mM NaCl, 5 mM KCl, 20 mM PIPES (Sigma), 0.45% glucose, DNase I (Worthington, 100 units per sample), 1× Antibiotic/Antimycotic (GIBCO, pH adjusted to 7.6) for 30 min at 37 °C on gentleMACS™ Octo Dissociator with Heaters (Miltenyi Biotec, program 37C_ABDK). After digestion, the cell suspension was applied to a MACS SmartStrainer (70 µm, Miltenyi Biotec) and centrifuged at $300 \times g$ for 10 min at 4 °C. The cell pellets were then subjected to red blood cell removal (Red Blood Cell Lysis Solution (10×), Miltenyi Biotec) and debris removal (Debris Removal Solution, Miltenyi Biotec) according to the manufacturer's instructions.

### Microwell-based scRNA-seq

Dissociated cells from each mouse brain tumor specimen, PDX specimen or patient-derived slice cultures were profiled using microwell-based single-cell RNA-seq (Yuan and Sims, 2016) as previously described (Zhao et al, 2021). Each cDNA library was barcoded with an Illumina sample index. Libraries with unique Illumina sample indices were pooled for sequencing on an Illumina Nextseq500/550 with an 8-base index read, a 26-base read 1 containing the cell barcode (CB) and unique molecular identifier (UMI), and a 58-base read 2 containing the transcript sequence (mouse brain tumors, PDX specimen, TB6181, TB6328, and TB6458) or on an Illumina NovaSeq 6000 with an 8-base index read, a 26-base read 1 containing CB and UMI, and a 151-base read 2 containing the transcript sequence (TB6528, TB6534, and TB6545).

### scRNA-seq data pre-processing

scRNA-seq data were preprocessed as described previously (Zhao et al, 2021). Briefly, raw sequencing data were trimmed and aligned (Levitin et al, 2019; Yuan et al, 2018). Data obtained from the Illumina NovaSeq 6000 were corrected for index swapping to avoid crosstalk between sample index sequences using the algorithm described by (Griffiths et al, 2018) before assigning read addresses for each sample. Reads with the same CB, UMI and aligned gene were collapsed and sequencing errors in the CB and UMI were corrected to generate a preliminary matrix of molecular counts for each cell. We applied the EmptyDrops algorithm (Lun et al, 2019) to recover cell-identifying barcodes in the digital gene expression matrix and filtered out CBs as described by (Zhao et al, 2021) to remove low quality cells.

### Identification of neoplastic glioma cells and differential gene expression analysis in murine tumors

We identified transformed glioma cells from the murine scRNA-seq data by performing unsupervised clustering with Louvain community detection as implemented in Phenograph (Levine et al, 2015) followed by differential enrichment analysis using the binomial test to identify genes that are statistically enriched in each cluster as described previously (Shekhar et al, 2016). The pipeline for performing this analysis is previously described (Levitin et al, 2019) and the corresponding code can be found at https://github.com/simslab/cluster_diffex2018. We defined the transformed glioma cells as cells in clusters that exhibit statistical enrichment of the Cre recombinase transcript. We performed differential expression analysis between transformed glioma cell populations in the two mouse glioma models using the Mann–Whitney U-test as previously described (Zhao et al, 2021) after subsampling the relevant groups to the same cell numbers and the same average number of unique transcripts per cell and normalizing the resulting count data with scran (Lun et al, 2016). Genes were defined as differentially expressed between the two models if they $\log_2 FC > 1$ and FDR $< 0.05$ in at least three of the four pairwise comparisons of the two sets of biological replicates for each model. Based on these parameters, these significantly upregulated and significantly downregulated genes therefore comprise the "N1IC_up" and "N1IC_down" signatures, respectively.

### Gene set enrichment analysis (GSEA) in the murine model

Differential gene expression analysis was performed as described above and $\log_2 FC$ was then used to rank genes for Preranked GSEA. Preranked GSEA was performed in GSEA version 4.3.2 (Subramanian et al, 2005) using MSigDB C2 (KEGG, REACTOME, etc) and C5 (Gene Ontology) gene set libraries (Liberzon et al, 2011). Only statistically significant gene sets with a nominal $p$ value $< 0.05$ were depicted. Each tumor cell was also scored using the package AUCell in R (version 1.16.0) (Aibar et al, 2017) using the "Astrocyte-like" (AC) gene signature described in (Neftel et al, 2019), the "quiescence" gene signature described in (Shin et al, 2015), and the "persister" gene signature described in (Liau et al, 2017) and visualized as Violin plots using Seurat (version 4.1.0) (Hao et al, 2021).

### Identification of neoplastic glioma cells and non-tumor cells in patient-derived slice cultures

The malignant glioma cells and non-tumor cells were identified as previously described (Zhao et al, 2021). Briefly, we first merged

scRNA-seq data of all samples derived from the same patient for unsupervised clustering analysis. We used Louvain community detection as implemented in Phenograph for unsupervised clustering with k = 20 for all k-nearest neighbor graphs. The marker genes were identified using the drop-out curve method as previously described (Levitin et al, 2019) (https://github.com/simslab/cluster_diffex2018) for each individual sample, and we took the union of the resulting marker sets to cluster and embed the merged dataset. We defined putative malignant cells and non-tumor cells using the genes most specific to each cluster. Putative tumor-myeloid doublet clusters were removed first. Next, we computed the average gene expression of each somatic chromosome as previously described (Yuan et al, 2018). For data obtained from IDH1-wild type glioma tissues (TB6328 and TB6458, TB6528, TB6534, and TB6545), we define the malignancy score to be the log-ratio of the average expression of Chr. 7 (amplified) genes to that of Chr. 10 (deleted) genes (Zhao et al, 2021). For data obtained from the IDH1-mutant glioma (TB6181), we define the malignancy score to be the log-ratio of the average expression of Chr. 2 (unaltered) genes to that of Chr. 1 (deleted) genes. We plotted the distribution of malignancy scores and fit a double Gaussian to the malignancy score distribution and established a threshold at 1.96 standard deviations below the mean of the Gaussian with the higher mean (i.e., 95% confidence interval). Putative malignant cells with malignancy scores below this threshold and putative non-tumor cells with malignancy scores above this threshold were discarded as non-malignant or potential multiplets (Appendix Fig. S11).

### Whole-genome sequencing and analysis

Genomic DNA was extracted from a piece of fresh-frozen tissue from each patient tumor resection using the DNeasy Blood & Tissue Kit (Qiagen) according to the manufacturer's instructions. Libraries for whole-genome sequencing were constructed using the Nextera XT kit (Illumina) with unique i7 indices for each patient sample according to the manufacturer's instructions. Sequencing was performed with an Illumina NextSeq 500/550 using a 150-cycle High Output Kit (Illumina) with an 8-base index read, 80-base read 1, and 80-base read 2. Raw sequencing data were aligned to the human genome (GRCh38) using bwa mem as described previously(Yuan et al, 2018; Zhao et al, 2021). Briefly, we computed the number of de-duplicated reads that aligned to each chromosome for each patient and divided by the number of de-duplicated reads that aligned to each chromosome for a diploid germline sample from patient TB6545 (peripheral blood mononuclear cells) after normalization by total reads. We then normalized this ratio by the median ratio across all somatic chromosomes and multiplied by two to estimate the average copy number of each chromosome.

### Cell-type specific differential expression analysis in patient derived slice cultures

We identified differentially expressed genes for RSL3- vs. vehicle-treated tumor cells as previously described (Zhao et al, 2021). Briefly, we first randomly sub-sampled number of cells in each comparison to have the same number of cells as the condition with fewer cells. Next, we sub-sampled the count matrices for the two conditions such that they had the same average number of molecules per cell and normalized the resulting count matrix using scran (Lun et al, 2016). We then conducted differential expression

analysis for protein-coding genes using the two-sided Mann-Whitney U-test as implemented by the "mannwhitneyu" command in the Python module "scipy". The resulting *p*-values were corrected for false discovery using the Benjamini–Hochberg procedure as implemented in the "mutlipletests" command in the Python module "statsmodels". We used the same approach for the differential expression analysis for RSL3 and Ferrostatin-1 co-treated vs. RSL3-treated tumor cells.

### Gene set enrichment analysis (GSEA) in patient derived slice cultures

GSEA was performed using GSEA version 4.3.2 by ranking differentially expressed genes by +/−log-ratio of q-values obtained from comparing the RSL3- vs. vehicle-treated tumor cells or the RSL3 and Ferrostatin-1 co-treated vs. RSL3-treated tumor cells. Gene signatures used for the four-state model of glioma cell phenotype were described by Neftel et al (2019). Gene signatures used for identifying the metabolic signatures were the gene sets from the GO Biological Process ontology.

### Single cell hierarchical Poisson factorization (scHPF) analysis

The scHPF model was generated by combining scRNA-seq profiles from all patient slice culture data, randomly subsampling to an even number of cells from each patient, and randomly down-sampling the resulting count matrix so that the average number of unique transcripts per cell was equal for each patient. We subjected the processed count matrix to probabilistic matrix factorization with scHPF (https://github.com/simslab/scHPF) (Levitin et al, 2019). To generate Fig. 7A–F, we embedded the resulting cell score matrix in two dimensions with UMAP. The relative expression of chromosomes exhibiting aneuploidy shown in Fig. 7E,F was computed using scHPF-imputed average expression values for the genes on each chromosome as described in Zhao et al (2021).

### scRNA-seq for TS543 in vitro RSL3 treatment

TS543 cells were plated at density $1 \times 10^4$ viable cells/cm$^2$ and cultured with NeuroCult™ NS-A basal medium supplemented with NeuroCult™ NS-A proliferation supplement, 20 ng/ml EGF, 10 ng/ml bFGF, and 2 µg/ml heparin (Stem Cell Technologies) in a humidified incubator with 5% $CO_2$ at 37 °C. When the diameters of neurospheres reached to approximately 50 to 100 µm after about 5 days' culture, the culture medium of neurospheres were replaced with pre-warmed medium containing RSL3 with desired concentration or corresponding volume of vehicle (DMSO). We used 0.2 µM RSL3 for 1-day (24 h) treatment and 0.1 µM RSL3 for 3 days treatment. At the end of the drug treatment, neurospheres were collected and dissociated to single cells by pipetting. The dissociated TS543 cells with different treatment conditions were stained with TotalSeq™-A hashtag antibodies (BioLegend) before pooling for scRNA-seq. Briefly, 1 million dissociated TS543 cells at each treatment condition were resuspended in 100 µl staining buffer (2% BSA, 0.01% Tween in PBS). Then 10 µl Human TruStain FcX™ (Fc Receptor Blocking Solution, BioLegend) were added to each cell resuspension to be incubated at 4 °C for 10 min. Then each sample were independently stained with one of the TotalSeq™-A anti-human hashtag antibodies (BioLegend) at 4 °C for 30 min followed by 3 times washing with staining buffer. Samples stained with unique hashtag antibodies were pooled together, washed once with staining buffer, and then resuspended in PBS for scRNA-seq.

The microwell-based scRNA-seq was performed as described previously (Zhao et al, 2021) with modifications in the cDNA amplification and purification steps to generate both the transcriptome and Hashtag oligonucleotide (HTO) libraries according to the Cell Hashing protocol (Stoeckius et al, 2018). Briefly, HTO additive oligonucleotides were spiked into the cDNA amplification PCR at a final concentration of 2 nM. And cDNA amplification PCR was performed as previously described (Zhao et al, 2021). Following cDNA amplification, 0.6X SPRI (Beckman Ampure XP beads) was used to separate the large cDNA fraction from the Cell Hashtag (HTO)-containing fraction. The cDNA fraction retained on beads was processed as previously described (Yuan and Sims, 2016) to generate the transcriptome library and barcoded with an Illumina i7 index. An additional 1.2X reaction volume of SPRI beads was added to the supernatant containing HTO fraction to bring the ratio up to 1.8X. The beads were washed with 80% ethanol, eluted in water, and an additional round of 1.8X SPRI performed to remove excess single-stranded oligonucleotides from cDNA amplification. After eluting HTO library to 90 µl water, we added 100 µl 2x KAPA Hifi PCR master mix, 5 µl of 10 µM TruSeq DNA D7xx_s primer (containing i7 index) and 5 µl of 10 µM P5-SMARTpcr (Yuan and Sims, 2016) to amplify HTO library using PCR amplification cycles of following conditions: 95 °C 3 min, 8 cycles of (95 °C 20 s, 64 °C 20 s, 72 °C 20 s), 72 °C 5 min. Amplified HTO library with sequencing index were purified by adding 1.6x SPRI, washed twice with 80% ethanol and eluted in 50 µl water. HTO library with sequencing index were spiked in at 5% when loading indexed cDNA libraries for sequencing on an Illumina Nextseq500/550 with an 8-base index read, a 26-base read 1 containing CB and UMI, and a 58-base read 2 containing the transcript sequence or the hashtag barcodes. Sequencing data of transcripts was preprocessed as other samples described in the scRNA-seq data pre-processing section of the Method section. Hashtag barcodes was preprocessed and extracted using DropSeq-Pipeline8 (https://github.com/simslab/DropSeqPipeline8). To demultiplex hashtag barcodes, we normalized HTO raw counts by centered log ratio (CLR) transformation and performed k-means clustering and statistical identification of singlets by fitting a negative binomial model as previously described (Stoeckius et al, 2018).

Differential expression analysis for RSL3- vs. vehicle-treated cells and the gene set enrichment analysis (GSEA) are same as described in the method section for the patient derived slice cultures.

### Identification of neoplastic cells and cell state analysis in TS543 PDX mouse model

We identified transformed TS543 cells from the TS543 PDX scRNA-seq data with the fraction of sequencing reads aligned uniquely to human greater than 90%. We performed unsupervised clustering of transformed TS543 cells using Louvain community detection as implemented in Phenograph (Levine et al, 2015) (https://github.com/simslab/cluster_diffex2018). Gene signatures of four GBM states and cell cycles were derived from Neftel et al (2019), and the single cell gene signature score was calculated as previously described (Neftel et al, 2019). In brief, for a given set of genes, reflecting an expression signature of a specific cell type or biological function, we identified the control genes by first binning all analyzed genes into 100 bins of aggregate expression levels and

then, for each gene in the gene-set, randomly selecting 100 genes from the same expression bin. The gene signature score for each cell is calculated as the average expression of the set of genes minus the average expression of the control genes.

## Single-cell RNA sequencing analysis of published IDH1-wild type human glioma dataset

Single-cell RNA sequencing data was obtained from (Yuan et al, 2018), which is an annotated scRNA-seq dataset comprising tumors from 8 patients with IDH-wild-type glioblastoma. Cells that were annotated as "transformed" tumor cells were subsetted for each patient, marker genes were identified for each patient using the drop-out curve method as described in Levitin et al (2019) (www.github.com/simslab/cluster_diffex2018), and we took the union of the resulting marker sets to cluster and embed the merged 8-patient dataset into a UMAP. Merged scRNA-seq count data was loaded into R and normalized using the Seurat package.

Each tumor cell was scored using the package AUCell in R (version 1.16.0) (Aibar et al, 2017) for each of the six Neftel lineage gene sets: Mesenchymal 1 (MES1), Mesenchymal 2 (MES2), Astrocyte-like (AC), NPC-like 1 (NPC1), NPC-like 2 (NPC2), OPC-like (OPC). For each transformed cell, the highest of the 6 lineage scores via AUCell defined its lineage classification, and NPC1/2 and MES1/2 were then aggregated, resulting in 4 unique lineage classifications: MES, AC, NPC, OPC. Then, each cell also underwent cell cycle scoring using the "CellCycleScoring" function in Seurat (Butler et al, 2018) and the proliferation gene sets G1/S and G2/M described in Neftel et al (2019), which classified each tumor cell into "G0", "G1/S", and "G2/M" groups. Those assigned to G1/S or G2/M were defined as "proliferative", while those assigned to G0 were defined as "quiescent". These classifications were then projected on to the merged UMAP.

AUCell scoring for this dataset was also performed on each gene set in MSigDB C2 (KEGG, REACTOME, etc) and C5 (Gene Ontology) gene set libraries (Liberzon et al, 2011), as well as using the "N1IC_up" and "N1IC_down" gene signatures described above. AUCell scores for each of the 8 Neftel gene sets (6 lineage sets and 2 proliferation sets) across all patients underwent pairwise Spearman correlation analysis with AUCell scores for select C2 and C5 metabolic gene sets, as well as the "N1IC_up" and "N1IC_down" signatures, and these correlation results are depicted in heatmaps using the ComplexHeatmap (version 2.10.0) package in R (Gu et al, 2016).

## Bulk RNA sequencing analysis of convection-enhanced delivery clinical trial dataset

Transcript count data was obtained from (Spinazzi et al, 2022), which describes a clinical trial in recurrent glioblastoma in which 86 MRI-localized biopsies were taken before or after intratumoral delivery of topotecan across 5 patients. Differential gene expression analysis comparing pre- and post-topotecan biopsies was performed using the DESeq2 package in R and was done both on a patient-by-patient basis as well as across all patients. Genes were ranked using $\log_2 FC$, and Preranked GSEA was performed using the GSEA application as described above using the "N1IC_up" gene signature.

### Cancer Cell Line Encyclopedia (CCLE) analysis

Public processed data from the Cancer Cell Line Encyclopedia (CCLE) was extracted and downloaded from the open-source data portal https://depmap.org/portal, release "23Q2". Cell lines tagged as "CNS/Brain → Diffuse Glioma → Glioblastoma" were utilized for downstream analysis in R. Only IDH-wild type glioma cell lines with gene expression and drug sensitivity data, specifically area under the curve (AUC) for ferroptosis-inducing drugs RSL3, ML162, FIN56, and Lovastatin were used. Additionally, metabolomic data, specifically GSH, GSSG, NAD, and NADP abundance were extracted for each cell line.

Normalized gene expression profiles for each Glioblastoma cell line were z-normalized from FPKM values using the "zFPKM" R package (version 1.20.0). GSVA analysis was subsequently performed on each cell line as previously described, applying the eight Neftel cell state gene signatures, as well as the quiescence gene signature and the KEGG Notch signaling pathway. Based on these normalized enrichment scores (NES) for each cell line, cells were categorized as either "Notch high" if the NES > 0 and "Notch low" if NES < 0 for the KEGG Notch pathway.

Spearman correlations were calculated between the NES of each Neftel cell state and the quiescence and Notch pathway scores for all GBM cell lines. Additionally, Spearman correlations between the AUC of different ferroptosis-inducing drugs, as well as the normalized expression of *TSPO*, were inspected. Cell lines were stratified by "Notch high" and "Notch low" and compared for sensitivity to RSL3, as well as GSH/GSSG ratio, relative abundance of NAD+, and NADP+.

## Statistical analysis

Statistical analysis and generation of graphs were performed using GraphPad Prism (v9.0 and v10.0) and RStudio. *P* values were calculated using unpaired one- or two-tailed t-tests or paired two-tailed t-tests, where appropriate. A *P* value of.05 was the cutoff for statistical significance. Log rank/Mantel–Cox test was used for analysis of survival in the mouse model. Kruskal Wallis with uncorrected Dunn's post hoc test was used for multiple comparisons. Multiple hypothesis correction was performed using the Benjamini–Hochberg method or the 2-stage linear step-up procedure of Benjamini, Krieger, and Yekutieli, where indicated. Dose–response curves for ferroptosis drugs were generated using sigmoidal 4 parameter nonlinear regression. Data are reported as mean ± standard error of the mean (SEM). Number of replicates and details of statistical tests used are described in the figure legends and text. No statistical methods were used to predetermine the sample sizes. Experimenters were not blinded to experimental conditions. No data were excluded from analysis. Biological samples were excluded if sample preparation or data acquisition failed due to technical reasons. Data points used for analysis in each figure are provided in Source data files.

# Data availability

Datasets generated for this study including murine scRNA-seq data and the human cell line and slice culture scRNA-seq data are deposited in the Gene Expression Omnibus (https://www.ncbi.nlm.nih.gov/geo/query/acc.cgi?acc=GSE224727).

The source data of this paper are collected in the following database record: biostudies:S-SCDT-10_1038-S44318-024-00176-4.

## Code and materials availability

The code used for analysis and generation of figures is available on Github (https://github.com/simslab). All unique/stable reagents and mouse/cell lines generated in this study are available within reasonable request from the lead contact with a completed Materials Transfer Agreement. Further information and requests for resources and reagents should be directed to and will be fulfilled by the lead contact, PC (pc561@cumc.columbia.edu).

## Peer review information

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

## Acknowledgements

We would like to thank all patients who generously donated tissue for these studies. We would like to thank Dr. M Picard for his valuable input regarding mitochondrial phenotype and functions. We would like to thank Dr. Y Sun for technical assistance in obtaining murine MRIs. We thank members of the

Stockwell group and members in the High Throughput Screening Facility for excellent technical assistance, O Adeuyan for optimizing in vitro experiments, Dr. M Chiriac for assistance with figure editing, Drs. X Yang and G Zhang from the Proteomics and Metabolomics Core Facility at Weill Cornell Medicine for LC-MS metabolomics analysis, Dr. M Sisti. Dr. GM McKhann and Dr. B Youngerman for assistance with tissue acquisition. We thank Dr. Cameron Brennan for donating the TS543 cell line. MAB was partially supported by the NREF&AANS/CNS Section on Tumors Research Fellowship Grant and the R38CA231577 StARR Kirschstein NRSA award. JNB was partially supported by William Rhodes and Louise Tilzer Rhodes Center for Glioblastoma, the Khatib Foundation and The Gary and Yael Fegel Foundation. This study was supported by the Emerson Health Collective Cancer Research Fund and the NIH/NCI/ NINDS/NHLBI (JNB, PAS, and PC—R01NS103473, BRS—R35CA209896, JK— 5R01HL112626, Nikon A1RMP confocal microscope—S10RR02568). This study used instrumentation from Cancer Center Flow Cytometry, Oncology Precision Therapeutics and Imaging Core (OPTIC), Confocal Microscopy Core, and the Genomics and High Throughput Screening Shared Resource (in part funded by the NIH/NCI Cancer Center Support Grant P30CA013696).

## Author contributions

**Matei A Banu**: Conceptualization; Software; Funding acquisition; Investigation; Methodology; Writing—original draft; Writing—review and editing. **Athanassios Dovas**: Conceptualization; Investigation; Methodology; Writing— original draft; Writing—review and editing. **Michael G Argenziano**: Investigation; Methodology. **Wenting Zhao**: Investigation; Methodology. **Colin P Sperring**: Investigation. **Henar Cuervo Grajal**: Conceptualization; Investigation; Methodology; Writing—review and editing. **Zhouzerui Liu**: Investigation. **Dominique MO Higgins**: Investigation. **Misha Amini**: Investigation. **Brianna Pereira**: Investigation. **Ling F Ye**: Investigation. **Aayushi Mahajan**: Investigation. **Nelson Humala**: Investigation. **Julia F Furnari**: Investigation. **Pavan S Upadhyayula**: Investigation. **Fereshteh Zandkarimi**: Investigation; Methodology; Writing—review and editing. **Trang TT Nguyen**: Investigation. **Damian Teasley**: Investigation. **Peter B Wu**: Investigation. **Li Hai**: Investigation. **Charles Karan**: Software; Investigation; Methodology. **Tyrone Dowdy**: Investigation. **Aida Razavilar**: Investigation. **Markus D Siegelin**: Resources; Supervision; Methodology; Writing—review and editing. **Jan Kitajewski**: Conceptualization; Resources; Supervision; Funding acquisition; Investigation; Methodology; Writing—review and editing. **Mioara Larion**: Supervision; Methodology; Writing—review and editing. **Jeffrey N Bruce**: Conceptualization; Supervision; Funding acquisition; Methodology; Writing— review and editing. **Brent R Stockwell**: Supervision; Funding acquisition; Methodology; Writing—review and editing. **Peter A Sims**: Conceptualization; Supervision; Funding acquisition; Investigation; Methodology; Writing—original draft; Writing—review and editing. **Peter Canoll**: Conceptualization; Supervision; Funding acquisition; Investigation; Methodology; Writing—original draft; Writing—review and editing.

Source data underlying figure panels in this paper may have individual authorship assigned. Where available, figure panel/source data authorship is listed in the following database record: biostudies:S-SCDT-10_1038-S44318-024-00176-4.

## Disclosure and competing interests statement

BRS is an inventor on patents and patent applications involving small molecule drug discovery, and the 3F3-FMA antibody, co-founded and serves as a consultant to Exarta Therapeutics, and ProJenX Inc., holds equity in Sonata Therapeutics, and serves as a consultant to Weatherwax Biotechnologies Corporation and Akin Gump Strauss Hauer & Feld LLP. PAS receives patent royalties from Guardant Health. Columbia University has filed a patent application on the microwell single-cell RNA-seq technology used in this study, and PAS is listed as a co-inventor.

