## [Peer Review File · The EMBO Journal]

A cell state-specific metabolic vulnerability to GPX4-dependent ferroptosis in glioblastoma

Matei Banu, Athanassios Dovas, Michael Argenziano, Wenting Zhao, Colin Sperring, Henar Cuervo Grajal, Zhouzerui Liu, Dominique Higgins, Misha Amini, Brianna Pereira, Ling Ye, Aayushi Mahajan, Nelson Humala, Julia Furnari, Pavan Upadhyayula, Fereshteh Zandkarimi, Trang Nguyen, Damian Teasley, Peter Wu, Li Hai, Chuck Karan, Tyrone Dowdy, Aida Razavilar, Markus Siegelin, Jan Kitajewski, Mioara Larion, Jeffrey Bruce, Brent Stockwell, Peter Sims, and Peter Canoll

Corresponding authors: Peter Canoll (pc561@cumc.columbia.edu), Peter Sims (pas2182@cumc.columbia.edu)

Review Timeline:

Submission Date:	23rd Apr 24
Editorial Decision:	14th May 24
Editorial Correspondence:	17th May 24
Revision Received:	12th Jun 24
Accepted:	1st Jul 24

Editor: Daniel Klimmeck

Transaction Report:

This manuscript transferred to The EMBO Journal following peer review at another journal. The peer review comments and authors' responses were made available as agreed with the authors and the other journal, and were taken into account for the decision process at The EMBO Journal.

RESPONSE TO FIRST ROUND OF REVIEWS

We greatly appreciate the reviewers' input and broad expertise and have made an extensive effort to address all of their comments. We have performed multiple additional experiments which we believe have significantly advanced our work while further supporting the conclusions from our original submission. We now provide a revised version of our manuscript with new data and new figures, specifically focused on two main aspects raised by the reviewers:

1. The relevance and limitations of our two original AC-like cell state models. First, we provide analyses of human GBM scRNA-seq data to demonstrate that the quiescent AC-like glioma cells are present in various proportions in all human GBM samples, and that the AC-like gene signature is significantly correlated with the "N1IC_up" metabolic gene signature derived from our mouse model (**Figure S8d**). We have added additional text to further detail the importance of using a genetically engineered murine glioma model as well as organotypic slice cultures to study the AC-like cell state. Second, we have now added a second human glioma model, an *early* passage patient derived cell line which we treated with RSL3 and analyzed via single cell RNA sequencing to demonstrate targeting of the AC-like population (**Figure 6b,c**). Third, we use data from the Cancer Cell Line Encyclopedia (CCLE) as well as Notch overexpression and genetic inhibition in a human cell line to demonstrate that the effects of Notch signaling are cell-context dependent in driving cell state changes and therefore difficult to dissect mechanistically in human models (**Figure S9**). Lastly, we analyzed a patient derived cell line model via scRNA-seq over serial passages as well as a PDX model and demonstrate the absence of the quiescent AC-like population, which gets outcompeted by the highly proliferative populations or undergoes mesenchymal transformation and is therefore selected out as the cell line is passaged over time *in vitro* (**Figure 6a**), or when transplanted *in vivo* (**Figure S10**). Overall, we provide strong evidence that the genetic mouse model as well as the patient derived explant model are accurate and relevant models and in fact indispensable models to study this elusive yet important cell state.

2. In depth mechanistic studies to explain the differential sensitivity of AC-like glioma cells to GPX4 inhibitors and ferroptosis. We have performed more detailed mechanistic studies and now include a new main figure (**Figure 5**) to describe the role of mitochondrial lipid peroxidation in driving sensitivity to GPX4 inhibition and ferroptosis. We also describe an altered mitochondrial phenotype in the AC-like glioma cell state, characterized by decreased mitochondrial membrane potential, lower biosynthetic function with lower fatty acid oxidation and lower oxygen consumption rate, complex I dysfunction with increased levels of reductive substrates and increased ROS production and lipid peroxidation. These findings are presented in a new main **Figure 3**. We show that the AC-like N1IC model has higher mitochondrial lipid peroxidation secondary to this redox imbalance. Notably, decreasing baseline mitochondrial lipid peroxidation by modulating mitochondrial complex I activity, inhibiting the outer mitochondrial membrane transporter TSPO (involved in ROS induced ROS release and upregulated in the AC-like population) or using mitochondrial ROS scavengers decreased vulnerability to GPX4 inhibition. Furthermore, we show that RSL3 induces mitochondrial lipid peroxidation and that this event is required in our glioma model for effective induction of ferroptosis but not apoptosis. Lastly, we show that ATP levels do not affect response to ferroptosis in our model. Overall, we now provide a novel mechanistic link between an altered mitochondrial phenotype in AC-like cells driving increased lipid peroxidation and response to GPX4 inhibitors and ferroptosis.

Attached, we provide a point-by-point response to the reviewer comments, including each specific figure containing new data to address the respective comment. New sections added to the manuscript are highlighted in yellow. We believe these extensive revisions, with more in-depth mechanistic studies and additional human glioma models are a strong and compelling response to the concerns raised by the reviewers, supporting our novel findings that the AC-like quiescent glioma cell state harbors an altered mitochondrial phenotype which renders this population sensitive to GPX4 inhibition and ferroptosis. We hope the reviewers are satisfied with our efforts and find the data compelling, within the limitations of the existing models.

Point-by point rebuttal:

Reviewer #1 (Remarks to the Author):

In this manuscript (NCB-A50794), Banu, Dovas, Argenziano, Zhao et al. leveraged mouse models to demonstrate that glioma cells that are quiescent have a metabolic dependency (high lipid catabolism, increased ROS, high mitochondria uncoupling) that could be leveraged for therapeutic benefit via inhibition of GPX4. The heterogeneity in glioblastoma (GBM) is well appreciated but how specific populations can be effectively targeted is less clear, especially a slow cycling population. Therefore, this work is of great interest to the field and has the potential to make a big impact but providing much needed insight into this question. In general, the studies are well done and the manuscript is well written. The authors make good use of a new mouse model (integrating constitutive active notch 1 activity with PDGF-B overexpression and p53 deletion) and some sophisticated profiling to identify targetable alterations. However, I have a series of technical and narrative issues that reduce my enthusiasm (but could be addressed to ensure the claims made are supported by the data provided).

Response: We thank the reviewer for highlighting the significance of our work. In the revised manuscript we have addressed all of the points raised and believe we provide additional and strong evidence to support our claims.

1. In general, all the findings in this manuscript are made in a single mouse model with validation in limited patient tissue (and that itself is descriptive). It would be useful to functionally validate the findings of this paper in a series of human GBM models with the N1IC up and down properties.

Response: Unfortunately, patient derived cell lines do not maintain the AC-like cell state *in vitro*, likely because the quiescent AC-like population is selected out and/or transformed with serial passaging (**Figure 6a**), also shown in recent studies (PMID: 32253265). We also show using data from the Cancer Cell Line Encyclopedia (CCLE) that cell states encountered in patient-derived glioma specimens are not clearly delineated *in vitro*. As such it is not surprising that RSL3 sensitivity does not correlate with Notch signaling *in vitro* in the CCLE data set. We provide this data in a new **Supplementary figure S9a-c**. Importantly, activating Notch signaling in transformed cell lines induces MES but not the AC-like cell state, demonstrating the importance of Notch activation at tumor inception, as we have done in our mouse model, to drive this specific phenotype (**Figure S9d**). Inhibiting Notch signaling genetically *in vitro* also did not affect sensitivity to ferroptosis inducers (FINs), demonstrating that it is the AC-like phenotype and not the Notch signaling *per se* driving sensitivity to ferroptosis (**Figure S9g-j**). These findings further refute the possibility that Notch activity alone is responsible for modulating the altered mitochondrial activity outside of the AC phenotype and this is an important finding of our study which we now highlight. To further functionally validate our findings, we now use an early passage patient derived cell line and using scRNA-seq we find a small quiescent AC-like population which is sensitive to RSL3 (**Figure 6b**). Importantly, this population has a similar metabolic profile to the ferroptosis sensitive population in the mouse model and slice cultures (**Figure 6e**). The RSL3 depleted population is also enriched in the “N1IC_up” gene signature derived from the Notch mouse model (**Figure 6c**) which identified ferroptosis sensitive subpopulations of cells in human glioma specimens. Interestingly, this slower cycling population is selected out with cell passaging (**Figure 6a**), and when implanted in nude mice to generate a xenograft model (**Figure S10**). This further demonstrates the utility of genetically engineered murine models and organotypic slice cultures to study the quiescent AC-like cell state in human gliomas.

2. There appears to also be a limited amount of in vivo targeting validation of the proposed inhibitors. These types of studies (intracranial models with given inhibitors) are standard in the field and these studies should be included.

Response: We have previously validated *in vivo* efficacy of RSL3 using genetically engineered mouse models (PMID: 36864031). Unfortunately, as we now show, it is difficult to capture the AC-like cell population using human cell line models, including PDX models. As previously reported by other groups, PDX models select for rapidly proliferating cells. We provide similar findings in a new set of experiments (**Figure S10**). The quiescent AC-like population is present *in vitro* in early passage patient derived cell lines but only present in a very small proportion *in vivo*. Late passage human neurospheres also appear to lack this population while also undergoing a mesenchymal drift. Overall, our new *in vitro* data from early-passage, patient-derived glioma neurospheres provides valuable additional corroboration of the vulnerability of AC-like glioma cells to ferroptosis. However, the PDX data highlight the challenges of modeling this cellular state and support the two alternative approaches – a genetically engineered mouse model and patient-derived acute slice cultures of primary GBM resections – that were employed in our study.

3. It is nice that a RSL3 inhibitor is being used, but specificity would be strengthened with some companion genetic loss of function approaches (sh/sgRNA of the p53 and N1IC cell lines). It would also be worth considering some overexpression and/or rescue studies.

Response: We thank the reviewer for this suggestion. We have now included genetic GPX4 inhibition via siRNA and rescue with ferrostatin in both the p53 and N1IC cell models and present several key findings: 1) the N1IC model is more sensitive to GPX4 deletion than p53; 2) ferrostatin can partially rescue deletion of GPX4 and 3) genetic deletion of GPX4 induces lipid peroxidation as measured via BODIPY-C11, a well-established marker of ferroptosis. These new results are now included in **Figures 4d and S5f-h**.

4. In the abstract, the suggestion that “the link between tumor cell state and metabolic programs is poorly understood in glioma” is an overstatement as there are multiple examples of the link between cell state (such as the cancer stem cell state) and metabolic programs. In fact, there was a recent series of papers in Nature Cancer using single cell data discussing cell states (PMIDs: 33644767, 33681822, 35122077) and one of them was enhanced metabolic activity. This statement should be either revised or removed.

Response: We agree with the reviewer and in fact cite several of the studies mentioned above. We have removed this statement from the abstract.

5. In the introduction, the second paragraph is essentially a summary of the key findings and adds little to the introduction. I appreciate a short introduction (but I have an interest in this field) but for a broad readership, it may be useful to address some key concepts needed to appreciate the findings of this paper.

Response: We agree with the reviewer and have added an additional paragraph to discuss aspects of metabolism in GBM and have added more details regarding the astrocytic cell state. We have also added additional details in the Discussion.

6. In the introduction, I would recommend naming a specific number of cell states in GBM as it is evolving (and will continue to do so) but rather just state that there are multiple cell states.

Response: We agree with the reviewer that the paradigm of glioma cell states is evolving, and we have removed the naming of a specific number of cell states from the introduction.

Reviewer #2 (Remarks to the Author):

Banu et al. showed that N1IC tumors (with constitutively active Notch signaling) harbored quiescent astrocyte-like transformed cell populations while p53 tumors (with no constitutively active Notch signaling) were predominantly comprised of proliferating progenitor-like cells. In addition, N1IC neoplastic

cells exhibited high lipid catabolism, mitochondrial uncoupling and increased ROS production, rendering them more sensitive to inhibition of GPX4 and induction of ferroptosis. Treating patient-derived organotypic slices with a GPX4 inhibitor induced selective depletion of quiescent astrocyte-like glioma cell populations.

The authors provided an interesting link between the activated Notch signaling with sensitivity to induction of ferroptosis by GPX4 inhibition. However, the mechanism underlying this observation is largely unexplored.

Response: We thank the reviewer for this comment as it helped us clarify several important points in our manuscript by performing additional mechanistic studies. The sensitivity to GPX4-driven ferroptosis is not directly driven by Notch signaling *per se*. Rather, we show that Notch activation at tumor **inception** drives the formation of tumors with a more quiescent astrocyte-like transcriptional phenotype which is associated with a specific metabolic vulnerability. We performed several additional mechanistic studies to explore the role of Notch signaling in more depth. Activating Notch signaling in an already transformed cell line did not induce the astrocytic phenotype but induced mesenchymal transformation instead (**Figure S9d**). Importantly, activating Notch in a transformed cell line also did not induce the astrocytic metabolic phenotype that is seen both in our murine model and in the AC-like tumor cell population in human specimens and slice cultures (**Figure 9e**). Genetically inhibiting baseline canonical Notch signaling in a human cell line using a dominant negative MAML construct also did not change sensitivity to either GPX4-driven or SLC7A11-driven ferroptosis (**Figure S9g-j**), demonstrating that Notch signaling alone is not responsible for driving sensitivity to ferroptosis inducers. Data analysis on patient derived glioma cell lines from the Cancer Cell Line Encyclopedia (CCLE) did not demonstrate a significant correlation between Notch status and response to GPX4 inhibition (**Figure S9b-c**). We conclude that timing and context of Notch activation play an important role in guiding neoplastic cells along a specific fate, including activating specific metabolic programs. It is the metabolic alterations of the quiescent AC-like cells state that account for the sensitivity to ferroptosis. Therefore, as recommended by the reviewer, we now further probed the underlying metabolic mechanisms driving sensitivity to ferroptosis, including TSPO. In human cell lines from the CCLE, TSPO expression correlated with response to GPX4 inhibitors (**Figure S9b**). TSPO is upregulated in our N11C murine glioma model as well as in the human quiescent tumor populations and it is part of the “N11C_up” gene signature. Our new mechanistic studies demonstrate that TSPO modulates mitochondrial lipid peroxidation, and inhibiting TSPO using a pharmacologic inhibitor reduced sensitivity to GPX4 inhibitors and ferroptosis (**Figures 5h-j and S7g,h**). Lastly, we show that the astrocytic population has an altered mitochondrial state, which leads to high mitochondrial and global lipid peroxidation, redox imbalance and depletion of GSH (**Figures 3, 5 and S7**). We propose that these mechanisms underlie the observed sensitivity to ferroptosis.

The findings that active Notch signaling is linked to quiescent cell state and that inactive Notch signaling is linked to progenitor-like cell states are inconsistent to the publications demonstrating that Notch signaling is highly active in GSCs, where it suppresses differentiation and maintains stem-like properties (PMID: 30832246). This raised the question that how much the relatively simplified murine glioma models in the current report recapitulates the heterogeneity of human glioma. In addition, the mechanism underlying Notch signaling-induced metabolic alteration is unclear.

Response: Our findings, and the referenced review article (PMID: 30832246) both show that the effects of Notch signaling are cell-context dependent. The studies discussed in the referenced review article were investigating Notch signaling in fully transformed Glioma Stem Cells. In contrast, we showed that Notch activation at tumor inception will induce an AC-like phenotype. These two findings are not inconsistent. We have revised our discussion to better highlight this point. Also, we acknowledge that the murine glioma model does not recapitulate the heterogeneity seen in human gliomas. However, the model does recapitulate key features of the astrocyte-like glioma cell state and associated metabolic vulnerability that we also identified within the heterogeneous populations seen in the acute slices of human glioma samples. We have also included additional analyses of scRNA-seq of human GBM samples to show that the quiescent AC-like glioma population is present in various proportions in all

patient tumors (**Figure S8a**), and that the “N1IC_up” gene signature derived for the Notch murine glioma model is significantly correlated with the AC-like glioma cell state in the same scRNA-seq dataset (**Figure S8d**). Importantly, we show that RSL3 selectively depletes both the AC-like gene signature and the “N1IC_up” gene signature in early passage human cell lines (**Figure 6b,c**) and acute slices generated from patient samples (**Figure 7g,k**).

Specific points

1. Figure 2. To verify the high FAO and TCA cycle activity in N1IC cells, flux analyses of labeled fatty acid (PUFA) in both p53 and N1IC cells should be performed.

Response: We thank the reviewer for this suggestion. We have now performed this experiment and found that FAO/TCA cycle activity was in fact downregulated in the N1IC cell lines. The altered mitochondrial function in AC-like cells appears to involve decreased metabolic activity, lower OCR and high redox imbalance. The high FAO transcriptional signature is compensatory in the setting of this altered mitochondrial state. This helped us further characterize the AC mitochondrial state to better understand this important yet understudied driver of ferroptosis. We provide this data in a new **Figure 3** describing mitochondrial activity and phenotype in the two models.

2. TSPO facilitates cholesterol transport across the mitochondrial intermembrane space and required for biogenesis and membrane maintenance. Does depletion of Tspo affect mitochondrial membrane potential, mitochondrial respiration, and generation of ROS in N1IC cells?

Response: We thank the reviewer for this helpful question and suggestion. Indeed, TSPO is upregulated in the N1IC mouse model. TSPO has been shown to decrease the membrane potential and increase ROS production, the mitochondrial phenotype seen in N1IC cells. When we performed TSPO inhibition with the TSPO inhibitor PK11195 we noted decrease in mitochondrial lipid peroxidation, whereas total lipid peroxidation or total ROS production were not significantly altered (**Figure 5h, S7g**). Furthermore, inhibition of TSPO for 24 hours had little effect on mitochondrial membrane potential. This may be related to the length of treatment with PK11195 but it is beyond the scope of our manuscript as it does not directly relate to ferroptosis sensitivity. TSPO appears to have a very specific role in the AC-like population by controlling mitochondrial lipid peroxidation and therefore sensitivity to GPX4 inhibitors. Importantly, sequential treatment (but not concomitant treatment) with PK11195 and RSL3 led to a decrease in sensitivity to ferroptosis (**Figure 5j**). Therefore, we provide a mechanistic link between the expression of TSPO in glioma cells and sensitivity to ferroptosis via mitochondrial lipid peroxidation. The role of cholesterol transport in mediating membrane stability and ferroptosis is intriguing and needs further exploration but these studies are beyond the scope of this manuscript, which is focused on redox imbalance.

3. Fig. 2J, 2K. IF analyses should be conducted by using a mitochondrial protein as a control. Is the difference attributed to different amounts of mitochondria in the cells? The mitochondrial membrane potential should be measured by alternate ways.

Response: We thank the reviewer for this suggestion. We have performed several additional experiments to address this. We have now included a mitoGREEN stain and found that there was indeed a significant difference in mitochondrial numbers between models, with fewer mitochondria in the N1IC cells (**Figure 3e**). We have also included two alternate ways to measure MMP: the ratiometric dye JC-1, which is independent of mitochondrial numbers (**Figures 3f and S7e,h**), and spinning disk microscopy with quantification of mitoRED mean fluorescence intensity and confirm that there is indeed more depolarization in the N1IC mitochondria (**Figure S4a**).

4. Fig. S5 shows that p53 cells have higher OCR than N1IC cells, suggesting that p53 cells have more mitochondrial ROS production than N1IC cells. This is against the authors' conclusion that N1IC cells have more ROS production. The total cellular ROS and mitochondrial ROS should be measured to clarify this issue.

Response: These are critical points that needed to be clarified and we thank the reviewer for these comments. We have performed several additional mechanistic studies to explore this. First, we found that N1IC cells have fewer mitochondrial numbers with lower FAO, as assessed via palmitate flux analysis (**Figure 3g,i,j**). These findings alone can explain the lower OCR despite the consistently higher ROS production in the Notch model. We also now show that N1IC cells have higher total ROS production, higher cellular lipid peroxidation and higher mitochondrial lipid peroxidation (**Figures 2, 3**). We have confirmed that N1IC have higher ROS in several ways including demonstrating higher total ROS via H2DCFDA, higher lipid peroxidation via BODIPY-C11 and higher mitochondrial peroxidation via mitoCLOX (**Figure 3**). Notably, mitoCLOX is a ratiometric dye measuring lipid peroxidation independent of mitochondrial numbers. Importantly, we show that N1IC cells have an altered mitochondrial phenotype, with higher levels of reductive substrates including higher NADH/NAD, decreased mitochondrial membrane potential, lower OCR and higher mitochondrial lipid peroxidation as well as decreased GSH levels which leads to redox imbalance and higher response to GPX4 inhibitors and ferroptosis (**Figure 5**). We thank the reviewer for suggesting these additional experiments as this has now led us to more fully characterize the mitochondrial phenotype in AC-like cells which we link to ferroptosis susceptibility via GPX4 inhibition.

5. Fig. 3C. Chac1 and Slc7a11 are not markers specific to ferroptosis and can be upregulated by growth factor receptor activation and general ROS elevation.

Response: We agree that these are not highly specific markers for ferroptosis, but they do provide additional evidence for ferroptosis induction after RSL3. Ultimately, we show that we induce ferroptosis via GPX4 inhibition in several ways, including showing higher BODIPY levels, rescue with ferrostatin, and upregulation of these transcriptional markers. We have added a clarification in the Results section mentioning that these markers are nonspecific.

6. Figure S7 shows that both cell models were resistant to IKE, a potent system Slc7a11 inhibitor and ferroptosis inducer. Does IKE treatment decrease the total amount of reduced GSH compared to GPX4 inhibition? It is well known that inhibition of Slc7a11 or GPX4 can induce ferroptosis. What is the mechanism underlying the observed differential effects elicited by Slc7a11 and GPX4 inhibition?

Response: While we agree this is an interesting finding, the mechanisms of differential sensitivity to Slc7a11 and Gpx4 are beyond the scope of our current study. This may be due to different mechanisms of susceptibility to the different FIN types. N1IC cells have higher redox imbalance and higher baseline ROS, which we put forward as a key mechanism for the higher sensitivity to RSL3 and other GPX4 inhibitors. We suspect that these glioma cells are insensitive to system xc- inhibitors because they may use the transsulfuration pathway to obtain cysteine. Indeed, IKE minimally decreased the GSH/GSSG ratio. Lack of system xc- expression or strong over expression were not the basis for lack of sensitivity to xc- inhibitors based on WB (**Figure S6c**). Since we do detect transsulfuration pathway metabolites such as cystathionine in both cell types (**Table S4**), we think this may explain the lack of effect of blocking system xc- in these cells, although there are other possibilities, such as additional pools of glutathione or other thiols that circumvent the need for system xc-.

7. Fig. 3d shows that RSL3-induced GPX4 inhibition results in ferroptosis of about 50% p53 cells. However, IACS-mediated inhibition of complex 1 completely abolished RSL3-induced cell death. Inhibition of complex 1 blocks ATP synthesis. How is ATP depletion helpful for cell survival? In addition, inhibition of complex 1 by IACS and uncoupling of mitochondrial electron transport chain by FCCP both results in ATP synthesis inhibition. But, the data showed that IACS alleviated the cell death and that FCCP exacerbated the cell death. Why can ATP deficiency-induced cell death be rescued by ROS inhibition? Finally, mitochondria-produced ROS is linked more to apoptosis than to ferroptosis. These results are controversial to known knowledge and the authors' conclusion.

Response: We appreciate the reviewer's comment and have now included additional data (**Figure 5**) detailing the effects of mitochondrial electron transport chain (ETC) modulation on sensitivity to RSL3 and ferroptosis in our models. In this new figure we use various ETC inhibitors and mitochondria-specific ROS scavengers to reverse the N1IC mitochondrial phenotype and thereby increase resistance to ferroptosis. We now show that modulating mitochondrial ROS production at complex I impacts response to RSL3. These effects are independent of ATP levels. Notably, both IACS and FCCP reduce ATP production, but have opposite effects on response to RSL3. IACS, a specific complex I inhibitor, decreased mitochondrial and cellular lipid peroxidation and total ROS levels, and also decreased RSL3-induced ferroptosis. In contrast, FCCP did not impact ROS or lipid peroxidation in our cell model. Notably, FCCP does induce significant cell death, either in combination with RSL3 or on its own (**Figure 5f, 7d, f**), presumably via mechanisms that are independent of ROS or lipid peroxidation. In addition, we now show that modulating ATP production via Oligomycin, an ATPase inhibitor, does not affect response to RSL3 (**Figure 5g**). Furthermore, we now also show that there is no difference in ATP levels between the p53 and N1IC models, which have different sensitivities to GPX4 inhibition, therefore further supporting the concept that it is the difference in mitochondrial ROS production and lipid peroxidation (not ATP levels) that affects ferroptosis in our model. Equally important, modulating ROS production at complex III does not affect response to RSL3, demonstrating that it is complex I activity that is linked to ferroptosis in our models. Lastly, we also performed an experiment to address the reviewer's excellent point regarding apoptosis. While mitochondria-specific ROS scavengers directly affected response to RSL3 they did not affect response to the apoptosis inducer staurosporine (**Figure 5m**). In conclusion, we now provide in depth mechanistic studies to implicate complex I lipid peroxidation in modulating response to GPX4 driven ferroptosis in our glioma model.

8. Fig. 4. The results from Fig. 1-3 are based on the difference of activation of Notch signaling in glioma. However, the human sample studies did not provide connected mechanistic information between Notch signaling and metabolic reprogramming/ferroptosis. Has the gene expression profiling reflected the status of activated Notch signaling? Does activated Notch signaling induce the mitochondrial metabolic signatures observed in N1IC cells? if so, what is the underlying mechanism?

Response: To address the first question, we now show that the "N1IC_up" gene signature derived from the mouse model is depleted after RSL3 treatment in an early passage patient derived cell line (**Figure 6c**) and acute slice cultures generated from patient samples (**Figure 7k**). In regards to the second question, we present new data showing that the timing and context of Notch activation are critical to induce the AC-like glioma cell state and associated metabolic phenotype (**Figure S9d-f**). As detailed in previous responses, we used Notch signaling as a method to induce astrocytic transformation in the mouse model. However, activating Notch signaling in patient derived glioma cell lines did not induce a similar phenotype, and Notch activation in human cell lines did not correlate with differences in redox metabolites (**Figure S9c**). Together, our results suggest that the metabolic programs are tightly linked to the quiescent AC-like glioma cell state, but that the link to Notch signaling is indirect and complex.

Minor points:

Fig. S2B, C. Figure legends do not match the figures.

Response: We have fixed the figure legends.

Fig. S3A, E. The indication of the blue and red colors should be clarified.

Response: We have fixed the figure legend and added a color legend in the figure.

Reviewer #3 (Remarks to the Author):

Banu and coworkers compared side by side two genetic mouse models of gliomas (PDGFB O.E., p53fl/fl, ± constitutively active Notch, N1IC), with the Notch-tg model (NCKI) being more quiescent and consisting of astrocyte-like type of cells, whereas those expressing only endogenous Notch appeared to be more

proliferative and aggressive. Multi-omics analysis revealed that the NCKI model is characterized by a higher dependence on lipid metabolism and increased ROS generation, which rendered this type of tumor cells more vulnerable to ferroptosis. Accordingly, treating patient-derived organotypic slice cultures with GPX4 inhibitors specifically ablated the population of quiescent, astrocyte-like type of cells, indicating that a subset of cells in a tumor may account for tumor regression after standard therapy.

Although the study contains a series of mostly omics-based data ranging from mice to patient samples, they seem to be rather correlative with the role of Notch in the human specimens being less clear.

Response: We thank the reviewer for these comments and have now included much more detailed mechanistic studies to demonstrate the link between the astrocytic cell state, associated metabolic programs, the altered mitochondrial activity and susceptibility to ferroptosis. We believe this has significantly improved the manuscript. Importantly, we cannot directly link Notch signaling to ferroptosis susceptibility. Rather, we use Notch signaling as a way to model the AC-like phenotype in a mouse model. Notch is known to drive astrocytic cell fate in glial progenitors. In addition, we provide analysis of scRNA-seq data from human GBM samples to show that the N1IC_up gene signature derived from our mouse glioma model is significantly correlated with the AC-like gene signature (**Figure S8d**). We found that high Notch is not necessarily predictive of sensitivity to ferroptosis in all models (**Figure S9b,c**). Furthermore, inhibiting canonical Notch signaling did not impact response to GPX4 inhibition (**Figure S9g,h**). Therefore, we conclude that the effects of Notch signaling are highly context and timing dependent hindering in depth mechanistic studies in human models. Nonetheless, we demonstrate in two independent human models, using an *in vitro* patient derived cell line and patient derived slice cultures, that the “N1IC_up” gene signature derived from the mouse model is consistently depleted with RSL3, therefore labeling the ferroptosis sensitive cell populations. We now include new data with an early passage patient-derived human glioma line to demonstrate the link between the AC-like population and sensitivity to ferroptosis in an additional human glioma model and again demonstrate that the “N1IC_up” gene signature derived from the murine model properly labels the ferroptosis susceptible populations (**Figure 6b**).

Moreover, the functional part of the study as presented is rather perfunctory requiring further in-depth experimentation.

Response: We agree with the reviewer about the need for further mechanistic studies and have now performed a much more detailed functional analysis with interesting and compelling findings. We addressed two important mechanistic questions: 1) Does Notch signaling alone increase sensitivity to ferroptosis? And 2) What are the metabolic abnormalities in the AC-like population/N1IC model driving redox imbalance? To address the first question, we showed that activating Notch at tumor inception in a murine model induces an AC-like phenotype and a ferroptosis responsive population. However, activating Notch in human glioma cell lines does not induce a similar AC-like or metabolic phenotype and genetically inhibiting canonical Notch signaling does not alter sensitivity to ferroptosis (**Figure S9**). We conclude that it is the AC-like phenotype and the linked metabolic programs that drive susceptibility to RSL3. To address the second question, we performed in depth functional studies and find an altered mitochondrial metabolism in the AC-like population, with decreased metabolic activity including low fatty acid oxidation and low OCR, redox imbalance, high lipid peroxidation, high levels of reductive substrates and low GSH levels (**Figure 3**). The resulting redox imbalance, likely due to altered activity at complex I, leads to increased susceptibility to GPX4 inhibitors. Blocking mitochondrial lipid peroxidation with complex I inhibitors or mitochondrial ROS scavengers decreased sensitivity to RSL3 (**Figure 5**). Therefore, we propose that the mitochondrial redox imbalance is a key mechanism responsible for the increased susceptibility to GPX4 inhibition.

- The authors need to provide more conclusive data on the relevance of the transgenic models in the development of glioblastoma in patients.

Response: The relevance of the Notch mouse model to glioblastoma in patients rests on our finding that the glioma cells in this model resemble the quiescent AC-like glioma cell state that is seen in human GBM samples, but is not well represented in other models. We have provided additional analyses of a scRNA-seq dataset from human GBM samples to demonstrate the presence of the AC-like glioma population in all samples, and the phenotypic similarity to the N1IC glioma cells (**Figure S8a,d**). Importantly, the N1IC glioma cells retained an AC-like phenotype in culture, which enabled us to perform a series of *in vitro* functional studies to characterize the metabolic phenotype and identify a specific therapeutic vulnerability to ferroptosis inducing drugs (**Figures 3-5**). We also show that the AC-like glioma cell population is present in acute slice cultures generated from GBM patient samples, and treating the slices with RSL3 depleted the AC-like glioma cell state and the “N1IC_up” gene signature derived from the Notch mouse glioma model (**Figure 7g,k**). Furthermore, we also validate the presence of this population in an early passage human derived cell line and show that RSL3 depletes the cell subpopulation labeled by the “N1IC_up” gene signature (**Figure 6b,c**). Thus, we provide several lines of data highlighting the relevance of the Notch mouse glioma model to human GBM, and its utility in identifying a specific therapeutic vulnerability of a pervasive glioma cell state.

- As a direct comparison and important control, why didn't the authors include a constitutively inactive Notch in their in vivo studies?

Response: We used Notch activation at tumor inception to induce an AC-like cell fate in the transformed cell population. Ablating baseline Notch would not provide an accurate comparison. We do appreciate the reviewer's suggestion about assessing the role of baseline Notch signaling in driving sensitivity to ferroptosis. Therefore, we have now included an additional set of experiments using a dominant negative MAML construct to block canonical Notch signaling in a patient derived human cell line (**Figure S9**). As hypothesized, we show that inhibiting baseline Notch in transformed cells does not induce a more progenitor-like phenotype, with stable levels of SOX2 or OLIG2 expression. Furthermore, inhibiting baseline Notch activity also does not change sensitivity to ferroptosis, either via GPX4 or via SLC7A11 inhibition. This further demonstrates that modulating Notch signaling alone in already transformed cells *in vitro* is insufficient to drive the phenotype we see in both genetic mouse models and human organotypic slice cultures.

- The functional data in terms of ferroptosis sensitivity are solely based on RSL3 (and similarly reactive compounds). Though RSL3 is a good proxy and an easy tool for studying a possible involvement of ferroptosis in cell death processes, it is not specific for GPX4 (see earlier works by Wang et al, Beijing). As such, the authors need to experimentally address at least in cultured cells or tissues whether the genetic modulation of GPX4 (and perhaps other ferroptosis players) would give similar results.

Response: As suggested, we have now performed genetic, more specific GPX4 ablation in the murine cell lines using siRNA (**Figures 4, S5**). Using this method, we first demonstrate that GPX4 knockdown induces higher lipid peroxidation, as assessed via BODIPY-C11 flow cytometry. Next, we show that these effects can be rescued with the specific ferroptosis inhibitor Ferrostatin. Lastly, we show that N1IC cells are more sensitive to GPX4 inhibition compared to p53 cells. Therefore, we provide an additional, more specific, method to demonstrate the higher susceptibility of the Notch AC-like cells to GPX4 inhibition and ferroptosis.

- While lipidomics data may provide a first indication whether there was increased remodeling /repair of phospholipids in response for instance to oxidative stress, a more detailed analysis of the oxidized fraction of esterified fatty acyl chains in phospholipids is required. The oxidized fraction seems to be rather subtle (see also Fig 2f). Importantly, the authors need to check for proliferation rates to analyze whether the observed effects are only due to an altered proliferation.

Response: We thank the reviewer for the suggestion and perspective on the lipidomics data. We acknowledge that the two models have different proliferation rates as this is part of the cell state-metabolic phenotype link we describe in this study. Therefore, we cannot uncouple the lipid profile from

proliferation. Nonetheless, the trends we observe are in the opposite direction – the slower proliferating cell population has a higher oxidized lipid fraction as assessed by BODIPY-C11. We have replaced the panel in **Figure 2h** to better illustrate the degree of lipid peroxidation in the N1IC model – notably this is on a logarithmic scale and therefore the difference in lipid peroxidation between models is quite significant. We now also provide a quantification across experiments with statistical analysis. Furthermore, we have normalized the lipidomic data using protein concentrations to account for potential differences in cell numbers. We also now demonstrate that there is higher lipid oxidation both at the global level using BODIPY-C11 and at the mitochondrial level, using mitoCLOX. We agree that it would be interesting to look at oxidized fatty acyl side chains to define specific oxidation products that accumulate during ferroptosis or other oxidative stress conditions. However, detecting oxidized fatty acyl tails is challenging to perform technically, as the oxidized products are only present in a small percent of the parent lipid species, and since various oxidation products are typically formed, each is quite low in abundance and hard to detect. Moreover, defining the identity of specific oxidized lipids is challenging, as it is difficult to identify where the oxidation takes place. We have attempted to do oxilipidomics, but found it was not reproducible for the above reasons. We know there are a few labs that can reliably detect oxidized lipids, but they have developed deep expertise and may have specialized instrumentation not available to us. Therefore, while we agree this would be an excellent area of future collaboration, we respectfully propose that this analysis would be beyond the scope of what can be done for this study and is not critical for our conclusions.

- Total and reduced levels of glutathione should be shown side by side. A 20% drop of reduced GSH likely bares any substantial biological meaning.

Response: We thank the reviewer for the suggestion and have now included a GSH/GSSG ratio alongside the GSH quantification in both the mouse model and cell lines isolated from the mouse model (**Figure 3d**). In both instances we show a decrease (>50%) in the GSH/GSSG ratio and this is consistent with the increased sensitivity to GPX4 inhibition of the N1IC model.

- Fig. S2c – the differences are hard to discern.

Response: We have added a quantification of the nuclear N1IC levels in the two cell models (**Figure S2d**).

- Besides ferroptosis markers, the number of dying cells needs to be determined in Fig. 2f-i. Are these effects sensitive to ferroptosis inhibitors?

Response: When we performed the flow cytometry experiment (now shown in revised **Figure 2h**) we gated on live cells, as we describe in the Methods section and in Extended data figure 1 demonstrating our gating strategy. The levels of BODIPY in N1IC cells are significantly lowered with a ferroptosis inhibitor (Ferrostatin), as we now depict in **Figures S7j**, further demonstrating that N1IC cells undergo baseline lipid peroxidation and ferroptosis secondary to an altered mitochondrial state with higher mitochondrial lipid peroxidation.

- Fig. S4e,f: These differences are minor - H2DCFDA is likely the most unspecific ROS indicator. Can these effects be blunted by ROS scavengers including mito-targeted ones, iron chelators etc.?

Response: We thank the reviewer for the suggestion to perform more detailed mechanistic studies into the role of specific organelles in driving ROS production in these models. We have now performed these experiments and our new results provide valuable mechanistic insight. We agree that H2DCFDA is the most nonspecific ROS indicator, and we used it to measure global ROS production. ROS differences are quite significant between models and we provide a better figure to showcase this (**Figure 3c**). We have now performed more in-depth experiments to study the role of mitochondrial ROS in controlling sensitivity to RSL3-induced ferroptosis in these cell models. We found that the AC-like population exhibits altered complex I activity driving higher mitochondrial lipid peroxidation and therefore increasing

sensitivity to GPX4 inhibition (**Figure 3, 4 and 5**). In fact, using two different mitochondrial ROS scavengers (Mitoquinone and SKQ1) we demonstrate that we are capable of rescuing cells from ferroptosis quite effectively (**Figure 5i**). Mitochondrial ROS scavengers do not affect total ROS production but do impact cellular lipid peroxidation as assessed via BODIPY C-11. We have also investigated other ROS scavengers as suggested and provide interesting findings. Iron chelators, such as DFO, decrease mitochondrial lipid peroxidation while ferrostatin, a ROS scavenger that acts at the ER level, does not impact mitoCLOX. Nonetheless, both ferrostatin and DFO change cellular lipid peroxidation and neither of them affect global ROS production. Ferrostatin rescues cells from undergoing ferroptosis at a downstream level from the mitochondria. Importantly, we show that RSL3 induces mitochondrial lipid peroxidation and that interfering with mitochondrial lipid peroxidation via mitochondrial ROS scavengers, by inhibiting complex I or by blocking TSPO, changes sensitivity to ferroptosis in this model.

- Since inhibition of SLC7A11 had no effect, one may wonder whether SLC7A11 is expressed at all in these cells. How does this relate to an earlier study by Savaskan and coworkers (see PMID: 18469825)?

Response: This is an excellent point. We have now performed a Western blot experiment and demonstrate that these cells indeed express Slc7a11 at the protein level (**Figure S6c**). The previous study by Savaskan et al. focused on the role of Slc7a11 in regulating glutamate secretion and associated neuronal cell death and brain edema. These findings are very interesting, but are not directly related to our studies.

- As cell culture-based studies including organoids are artefact-prone (which is a main concern in the ferroptosis field), one may wonder whether the “specific” ablation of astrocyte-like quiescent tumor cells may also hold true in patient-derived xenograft (PDX) models of glioblastoma – a recent study reported on a metabolically more stable version of RSL3 (Randolph et al. J Med Chem 2023).

Response: To address this comment we assessed the effects of RSL3 on a human glioma cell line (TS543) both *in vitro* using early passage cells, and *in vivo*, in a PDX model as suggested. This experiment revealed two important findings. First, we showed by scRNA-seq that early passage TS543 neurospheres have a subpopulation of cells with an N1IC-like/AC-like phenotype. Next, we showed that treating the early passage TS543 neurospheres with RSL3 selectively depleted the AC-like and N1IC_up gene signatures (**Figure 6b,c**). Unfortunately, the AC-like cell state, a quiescent population, gets outcompeted by the highly proliferative populations and is selected out as the cell line is passaged over time *in vitro* or when transplanted *in vivo*. Additionally, cells in culture appear to undergo a mesenchymal drift. Therefore, we detected only a minute fraction (less than 0.5%) of AC-like tumor cells in the orthotopic PDX tumors (**Figure S10d-f**), making this model unsuitable for testing if RSL3 selectively targets the AC-like tumor cells *in vivo*. This finding highlights the challenges of modeling this cellular state in PDX models, and supports the two alternative approaches – a genetically engineered mouse model and patient-derived acute slice cultures of primary GBM resections – that were employed in our study.

- Only one short paragraph introduces to the topic. This should be more elaborated.

Response: We have now included an additional paragraph in the introduction.

- P3, Line 31: What does it mean that astrocyte-like glioma cells “preferentially used” lipid peroxidation?

Response: We have removed this vague statement from the text.

- Fig. S4: Mitochondrial ROS are usually generated at complexes I and III.

Response: We have now included a more detailed depiction of the mitochondrial ROS generation sites/mechanisms (**Figure 5a**).

RESPONSE TO REVIEWERS' COMMENTS FROM SECOND ROUND OF REVIEWS:

Reviewer #1:

In general, the **authors have been very responsive to previous comments** and the addition of new data is acknowledged. The additional modeling limitations are also acknowledged and I am glad to see another low passage model was provided for additional validation. **At this point, there are 2 minor issues that can be addressed in the discussion/text. No outstanding experimental issues remain.**

1. What is the IDH status from data obtained from the Cancer Cell Line Encyclopedia (CCLE) analysis? This should be included in the manuscript.

Response: We agree with the reviewer on this point and can easily provide this data.

2. The authors have appropriately addressed genetic manipulation of GPX4 inhibition with siRNA and ferrostatin rescue. The authors have demonstrated that rescuing cells from ferroptosis with mito-specific ROS scavengers have been quite effective. Why is ferrostatin only partially rescuing GPX4 inhibition? At which level does ferrostatin interfere downstream of mitochondria?

Response: The partial rescue may be explained by either an off-target effect of the siGPX4 reagent or by a source of lipid peroxidation that is not scavenged by ferrostatin. While mitochondria can generally be targeted by ferrostatin, different compartments within mitochondria have not yet been explored in detail. Therefore, it may be that the ROS driving cell death in these cells in response to siGPX4 are not completely blocked by ferrostatin – we can test this hypothesis by testing other ferroptosis-blocking radical trapping agents, such as liproxstatin-1, Trolox, and idebenone. Of note, as pointed out by the reviewer, mitochondrial ROS scavengers were particularly effective at rescuing cells from ferroptosis.

This is exciting work, congratulations.

Response: We thank the reviewer for his excellent suggestions and positive review of the manuscript.

Reviewer #2:

The revised manuscript has addressed some previous concerns. However, several key questions remain unanswered. The **phrase "beyond the scope of our manuscript" has repeatedly appeared in responses to main points 2, 6, and others from different reviewers. The manuscript, at its current stage, remains descriptive and correlational.**

Response: We respectfully disagree with this comment. Our revised manuscript addressed all of the reviewers' major points. In our responses, we used the phrase "beyond the scope of our manuscript" only 3 times, each time in reference to a point that was truly outside the focus of our manuscript. These included the following: 1) reviewer 2 asked us to investigate the possibility that TSPO affects mitochondrial membrane stability via its effects on cholesterol transport. While this is an interesting possibility, it is not the mechanistic focus of our paper. 2) What is the mechanism underlying the observed differential effects elicited by Slc7a11 and GPX4 inhibition? We show that our glioma cell models are relatively insensitive to Slc7a11 inhibitors although, as we show in the revised manuscript, they do express Slc7a11. We suspect that these glioma cells are insensitive to system xc- inhibitors because, as identified based on our metabolomic data presented in the manuscript, they use the transsulfuration pathway to obtain cysteine. To support this we show that IKE, an SLC7a11 inhibitor, minimally decreased the GSH/GSSG ratio. While these findings are also intriguing and warrant further investigation, they are not the focus of the manuscript and are not directly linked to the differential

sensitivity of the astrocyte-like population to GPX4 inhibitors. 3) Lastly, reviewer 3 asked for a more detailed analysis of the oxidized fraction of esterified fatty acyl chains in phospholipids. As we explain in our response, we do not have the technical capability to perform this analysis which we do not believe is critical for our conclusions. Notably, reviewer 3 did not object to this response. In our revised manuscript we provide a clear mechanistic explanation of the increased susceptibility to GPX4 inhibition in the astrocyte-like glioma populations. Specifically, astrocyte-like glioma cells have an altered mitochondrial phenotype with dysfunction at complex I which leads to increased ROS and lipid peroxidation. We have performed extensive mechanistic studies to demonstrate this important link. While other mechanisms may be at play, the goal of our paper was not to provide an exhaustive mechanistic investigation but instead to identify therapeutic vulnerabilities in the quiescent astrocyte-like glioma population resistant to standard of care therapies which could be translated into clinical trials. As such, we respectfully disagree that the manuscript is “descriptive and correlational” as we provide a strong mechanistic explanation.

Original general point:

The authors did not clarify how and why Notch signaling activation at the early or late stage of tumorigenesis has different effects on the astrocytic metabolic phenotype or response to the induction of ferroptosis by GPX4 inhibition. They also did not clarify why mouse tumor cells behaved differently from human tumor cells upon Notch signaling activation. A comparison of astrocyte-like mouse tumor cell lines with astrocyte-like human tumor cell lines may help.

Response: Our study focuses on the metabolic and therapeutic vulnerabilities of quiescent astrocyte-like glioma cells. The use of Notch activation in this study was to generate an animal model that resembles the astrocyte-like glioma state in order to study therapeutic vulnerabilities. We had no intention of studying the role of Notch signaling in early vs late tumorigenesis or in driving response to ferroptosis. Notch activation in other cancer models has been shown to directly induce the metabolic phenotype we see at tumor induction in the astrocyte-like glioma (PMID 24474689); however, we also show that simply activating Notch in human glioma cells *in vitro* is insufficient to induce either the astrocyte-like cell state or the associated metabolic programs proving that timing and context of Notch activation are crucial in driving this phenotype. We show that the mouse glioma model that includes activated Notch at the time of tumor induction generates tumors that more closely resembles the quiescent astrocyte-like glioma cell phenotype found in human tumors. This model is necessary to study the astrocyte-like glioma population in conjunction with slice cultures and early passage human cell lines due to its high plasticity and quick transformation in other models, as we show in the revised manuscript. The astrocyte-like populations are a small percentage of early passage human cell lines and are selected out and/or transformed with passaging or once transplanted to generate xenograft models, as we show in the revised manuscript. There are no pure astrocyte-like human tumor cells available, which are only maintained in organoid glioma models (as shown, for example, in this study: PMID 32253265). While we use this model to assess the metabolic alterations that are associated with the vulnerability to ferroptosis, we make no claims that Notch signaling alone drives the metabolic phenotype. In fact, we demonstrate that simply manipulating Notch signaling in fully transformed human glioma cells does not affect sensitivity to ferroptosis and induces a mesenchymal cell state with different metabolic programs, showcasing the limitations of these models. Therefore, we conclude that the astrocyte-like glioma cell phenotype is correlated with vulnerability to ferroptosis through an altered mitochondrial complex I activity and not increased Notch signaling *per se* and that timing of Notch activation may be important in driving metabolic programs which drive susceptibility to GPX4 inhibitors. Notably, since Notch signaling alone is insufficient to identify ferroptosis sensitive tumor populations, we derive a metabolic gene signature from the Notch astrocyte-like murine glioma model that can be used for this purpose. We show that this signature identifies ferroptosis susceptible tumor cell populations both in early passage cell lines and in patient-derived slice cultures, and that these cells are present in human GBM specimens irrespective of genetic background.

Original specific points:

1. Original Point 1. "If ' FAO/TCA cycle activity was in fact downregulated in the N1IC cell lines ' with decreased metabolic activity and lower OCR, how are the higher ROS and mitochondrial lipid peroxidation induced? These results are controversial.

Response: We strongly disagree with the reviewer that these findings are controversial. Reduced FAO and OCR concomitant with increased ROS and lipid peroxidation have been shown in other studies and are linked mechanistically to a complex I redox dysfunction (PMID: 38366593, 37262067, 34039636, 38359293, 33463549), as we also demonstrate in our study. One well documented mechanism which causes elevated ROS levels in the setting of reduced OCR is the reverse electron flow from complex II to complex I in the setting of reductive stress whereby oxygen becomes the electron acceptor, thus resulting in ROS generation (PMID: 38359293, 37262067, 34039636). Notably, lineage cell trajectories in other organs have been directly linked to reductive stress (PMID: 37558881). As a novel finding, we demonstrate here that this phenotype is linked to increased response to GPX4 inhibitors in the quiescent astrocytic-like glioma cell state. We have revised our discussion to better explain these points.

2. Original Point 2. The authors showed that the TSPO inhibitor PK11195 did not affect mitochondrial membrane potential. This finding is inconsistent with published reports demonstrating that TSPO knockout cells exhibit reduced mitochondrial membrane potential and decreased respiratory function (PMID: 32102369). This raises concerns regarding the specificity of the TSPO inhibitor PK11195 and its potential off-target effects. Furthermore, the authors claimed to have revealed 'a mechanistic link between the expression of TSPO in glioma cells and sensitivity to ferroptosis via mitochondrial lipid peroxidation,' but essential results supporting this claim are lacking, especially considering that the role of TSPO in cellular bioenergetics, associated mitochondrial function, and cell apoptosis (rather than ferroptosis) has been well documented.

Response: The TSPO experiments were performed in response to the reviewer's comments in the first round of reviews. Notably, we used the TSPO inhibitor at low doses and for a short time in order to demonstrate that inhibiting ROS-induced ROS release via TSPO/mitochondrial permeability transition pore can also impact response to ferroptosis. As a result, there were no effects on mitochondrial membrane potential, as these effects on TSPO have been shown to be dose and time-dependent. As stated previously, the effects of TSPO on cellular bioenergetics including the effects on cholesterol synthesis and mitochondrial membrane potential are not central to our story. The main mechanism driving GPX4 dependency in the astrocyte-like cell state is redox imbalance secondary to complex I dysfunction leading to increased ROS and lipid peroxidation which is further exacerbated by TSPO through ROS-induced-ROS release.

3. Original Point 3. If N1IC cells have decreased mitochondrial number and membrane potential and lower OCR, how is total ROS increased in the cells?

Response: Please see our answer to point 1 above. These results are consistent with multiple other studies demonstrating that complex I dysfunction results in a similar metabolic phenotype, namely reduced mitochondrial membrane potential, reduced OCR and increased ROS production (PMID: 38359293, 32941802, 38359293) We have revised our manuscript to better explain this point.

4. Original Point 7. Why does the reduction of ROS production by inhibition of complex III not affect the response to RSL3, as does complex I inhibition? Both complexes produce ROS. Is there any difference in lipid peroxidation caused by the ROS produced by these two different sources?

Response: As requested by the reviewer originally, we have investigated the role of complex I and III in driving sensitivity to ferroptosis and show that only complex I modulation affects this response. Our results are consistent with recent studies showing that ROS produced by complex I, but not complex III induce lipid peroxidation leading to ferroptosis (PMID: 38366593, 37262067 and 36831278). Furthermore, a recently published paper (PMID: 38366593) by one of the senior co-authors of this study demonstrated that interfering with electron flow at complex I induced mitochondrial stress, decreased

OCR, increased mitochondrial ROS and lipid peroxidation with induction of ferroptosis but that inhibiting complex III activity did not impact lipid peroxidation. We have revised our discussion to better explain these points

8. Original Point 8. The only distinction between the two mouse cell lines lies in the status of Notch signaling. However, the observed difference in the quiescent AC-like phenotype among the mouse cell lines could not be replicated using human cell lines. The authors' assertion that 'the metabolic programs are tightly linked to the quiescent AC-like glioma cell state, but that the link to Notch signaling is indirect and complex' appears weak and lacks persuasive evidence.

Response: We strongly disagree with the reviewer on this point. There are significant metabolic and phenotypic distinctions between the two cell line models, as shown in the manuscript. We used Notch at tumor inception to induce an astrocytic lineage trajectory and to model the astrocyte-like glioma cell state. Notably, Notch is a transcriptional master regulator with numerous downstream targets. Furthermore, secondary and tertiary effects of Notch signaling accumulate throughout tumor evolution. Therefore, it is not surprising that cell lines isolated from the two mouse models at end-stage show many differences in cell state and metabolic programs. Simply attributing these phenotypic differences to the status of Notch signaling in the cell lines is conceptually flawed and does not account for all of the divergent changes that accumulate throughout tumor evolution. To directly demonstrate this point, we overexpressed Notch in human glioma cells. However, since effects of Notch signaling are highly dependent on timing and context of Notch activation, simply activating Notch in transformed cells did not have the same effects as activating Notch *in vivo* at the time of tumor initiation. This is not surprising, and further points to the limitations of classical models such as human cell lines in modeling the phenotypic cell diversity seen in GBM. We provide evidence that both in the mouse model and in an early passage patient-derived cell line, in human slice cultures, and in human GBM specimens, glioma cell populations with an astrocytic phenotype have specific transcriptional metabolic programs and that these programs can be used to predict response to GPX4 inhibition.

Reviewer #3:

The authors **have substantially revised the manuscript and have addressed several of my concerns**. In particular, they show that lowering GPX4 expression in neuronal cells sensitizes to ferroptosis. Although they failed to provide a direct molecular link between Notch signaling and ferroptosis susceptibility, they provide evidence that scRNAseq signatures derived from human GBM samples resemble the mouse glioma N1IC-up signature that correlates with the AC-like signature (i.e. RSL3-sensitive). Accordingly, they found that the N1IC-up/AC-like cell population decreases after RSL3 treatment.

Response: We thank the reviewer for the succinct summary of our findings. Of note, we show that lowering GPX4 expression in both neural-progenitor like and in astrocyte-like glioma cells induces ferroptosis and cell death but that this effect is more pronounced in the astrocyte-like population, which is more susceptible to both pharmacologic and genetic GPX4 inhibition. Furthermore, we showed that the susceptibility to ferroptosis cannot be augmented by merely overexpressing activated Notch in fully transformed glioma cells, further demonstrating that it is the mitochondrial alterations and increased production of ROS seen in astrocyte-like glioma cells, rather than Notch signaling *per se*, that accounts for the increased sensitivity to ferroptosis.

Despite all these efforts, it remains rather vague what is causal and what is correlative.

Response: To address this comment, we have revised our discussion to explain more clearly which of our key findings are correlative (the correlation between the Astrocyte-like glioma cell phenotype and sensitivity to RSL3) and which are causal (complex I dysfunction with increased mitochondrial production

of ROS and increased lipid peroxidation leading to increased sensitivity to GPX4 inhibitors in the astrocyte-like glioma populations).

For instance, RSL3 treatment causes profound changes in redox enzymes and mitochondrial energy metabolism, although these changes are not consistent (new Fig. 3). Professional enzymes such as TXNRD1, SRXN and GCLM are down- while KEAP1, TXN1, TXN2 and those involved in ETC are upregulated.

Response: The reviewer has incorrectly interpreted the results shown in new Figure 3. We do not show that RSL3 induces these changes. Rather, the new Figure 3 shows the differentially expressed genes between the p53 glioma cells and the N1IC glioma cells. Notably, TXNRD1, SRXN and GCLM are downregulated in the N1IC glioma cells while KEAP1, TXN1 and TXN2 are upregulated along with many other transcripts for components of complex I. These findings point to differences in the mitochondrial phenotype and redox imbalance between the N1IC and p53 cells, which we then extensively dissect in the next panels of this figure and in supplementary figures. These are not induced by RSL3 treatment but rather are baseline differences in transcriptional metabolic programs between the two models that led us to investigate the mitochondrial activity and subsequently the role of ferroptosis to target this cell state.

Moreover, the link between the altered OXPHOS and the increased sensitivity to GPX4 inhibition is unclear and requires further mechanistic investigations. In other words, there is a lack of a clear picture that would allow to draw firm conclusions based on the data presented. At least a graphical summary is required.

Response: We respectfully disagree with this statement. We provide a clear mechanistic link between mitochondrial OXPHOS dysfunction and increased susceptibility to GPX4 inhibitors in the Notch astrocyte-like model. Specifically, we show that altered activity at complex I leads to increased ROS production and increased mitochondrial lipid peroxidation, which are mechanistically linked to increased sensitivity to GPX4 inhibition in the astrocyte-like glioma state. A graphical summary illustrating this mechanistic link can easily be provided.

Dear Dr Canoll,

Thank you again for the submission of your amended manuscript (EMBOJ-2024-117686-T) to The EMBO Journal. We have carefully assessed your manuscript and the point-by-point response provided to the referee concerns that were raised during review at a different journal. In addition, and as mentioned before, we decided to involve an arbitrating expert to evaluate the revised version of your work, with respect to technical robustness, conceptual advance and overall suitability of your work for publication in The EMBO Journal.

As you will see from his/her comment enclosed below, the advisor is in favour of the work stating the interest and value of your results and s/he is supportive of publication at The EMBO Journal, pending minor revision.

We are thus pleased to inform you that we can offer to swiftly move forward towards acceptance of this work at The EMBO Journal, pending minor revision of the following remaining issues, which need to be adjusted in a re-submitted version.

Based on the positive views of the advisor together with our own assessment, we decided to proceed with publication of your work at The EMBO Journal pending the above points in a time frame of two weeks.

Once we have received the revised version, we should then be able to swiftly complete formal acceptance and expedited production of the manuscript.

We also need you to take care of a number of minor issues related to formatting and data annotation, which I will share shortly in a separate message, together with additional changes and requests by our production team and my colleague H. Sonntag (CC'ed) for Source Data provision.

Please submit a revised version of the manuscript using the link enclosed below, addressing the advisor's comments.

As you might have seen on our web page, every paper at the EMBO Journal now includes a 'Synopsis', displayed on the html and freely accessible to all readers. The synopsis includes a 'model' figure as well as 2-5 one-short-sentence bullet points that summarize the article. I would appreciate if you could provide this figure and the bullet points.

Thank you again for giving us the chance to consider your manuscript for The EMBO Journal, I look forward to hearing from you and receiving your final revised version of the manuscript.

Kind regards,

Daniel Klimmeck

Daniel Klimmeck PhD
Senior Editor
The EMBO Journal.

EMBOJ-2024-117686-T, arbitrating advisor's comment:

I have now carefully reviewed the manuscript and the two rounds of revisions. I don't see any reasons why this would not be published as it is or with minor modifications (perhaps emphasizing more on the limitation of this mouse model) in EMBO.

This study provides valuable data linking cell states (notably the astrocyte-like state) to altered metabolic programming. The authors have done a commendable job of adding mechanistic studies and addressing several key issues raised by reviewers in the first round. While some concerns brought up by reviewers 2 and 3 are valid, I do agree that several points raised by reviewer 2 are indeed beyond the scope of this study (for example, the role of Notch in early vs late stages of gliomagenesis).

Some of my perceived limitations:

Lack of correlation between Notch signaling and RSL3 sensitivity in vitro: GBM cells cultured as neurospheres should retain plasticity. Yet, it is surprising that activating Notch in these cells was insufficient to guide the TS543 neurospheres toward the AC-like phenotype. This somewhat dampens my enthusiasm regarding the relevance of their N1IC mouse model. That said, all in vivo models of gliomagenesis suffer from limitations.

The notion that Notch activation at tumor inception forces an astrocytic developmental trajectory, but Notch activation later in tumor development is insufficient to induce similar cell states, is questionable, given what is provided. I feel that a distinction between inherent (for ex. Notch-driven or AC-like related) vs. acquired quiescence (for ex, induced by radiation therapy or other stress-inducing insults) is lacking.

Dear Dr Canoll,

Further to below, please find the mentioned additional formatting requirements and requests by our office team enclosed.

Please let us know any time should there be additional questions related.

We are looking forward to your final submission.

Best regards,

Daniel Klimmeck

Daniel Klimmeck PhD
Senior Editor
The EMBO Journal.

>> Please add up to five keywords to your study.

>> Author Contributions: Remove the author contributions information from the manuscript text. Note that CRediT has replaced the traditional author contributions section as of now because it offers a systematic machine-readable author contributions format that allows for more effective research assessment. and use the free text boxes beneath each contributing author's name to add specific details on the author's contribution.

More information is available in our guide to authors.
<https://www.embopress.org/page/journal/14602075/authorguide>

>> Adjust the title of the 'Conflict of Interest' section to 'Disclosure and Competing Interests Statement' and move after Acknowledgements.

>> Provide a completed Author Checklist.

>> Please provide the main manuscript text as .doc file.

>> Figure callouts: Table S7 needs to be called out.

>> Figures in separate files: Figures should be removed from the manuscript text and uploaded as individual, high resolution figure files. Legends should be placed after the References.

>> Please provide source data for the study as to the separate request e-mail by my colleague Hannah Sonntag.

>> Appendix file with ToC: the file with the supplementary figures should be renamed "Appendix". A table of contents with page numbers should be added on the first page, yellow highlights should be removed. Please correct the nomenclature in the appendix file and in the manuscript text to "Appendix Figure S1" etc. Please remove the table legends and add each legend to the corresponding table on a separate worksheet. Please rename Tables S2, 9 and 10 "Table EV1" - EV3, and rename tables S1, 3 - 8 and 11 "Dataset EV1" - EV8. Table S12 is source data and will need to be uploaded as such; Hannah Sonntag will email authors about this.

>> Funding: please enter the funding information into the list of funders in our online system. we don't need the green checkmark in case the funding source is not already present in our online system.

>> References: adjust reference format to EMBO Journal format, 10 authors et al, and place References after the Discussion, before figure legends.

>> Data availability section: provide a URL for the GSE dataset.

>> Consider additional changes and comments from our production team as indicated below:

- DAS:

Please note that the specific URL for GSE224727 dataset is not provided in the data availability statement.

- Figure legends:

1. Please note that the exact p values are not provided in the legends of figures 1b, g; 4c-d; 5d, f, h-i, k-l.
2. Please indicate the statistical test used for data analysis in the legends of figures 2g; 3g-h.
3. Please note that information related to n is missing in the legend of figure 1g.
4. Please note that n=2 in figure 4b.
5. Please note that the measure of center for the error bar needs to be defined in the legend of figure 2e.

Dear Dr Canoll,

Thank you again for the submission of your amended manuscript (EMBOJ-2024-117686-T) to The EMBO Journal. We have carefully assessed your manuscript and the point-by-point response provided to the referee concerns that were raised during review at a different journal. In addition, and as mentioned before, we decided to involve an arbitrating expert to evaluate the revised version of your work, with respect to technical robustness, conceptual advance and overall suitability of your work for publication in The EMBO Journal.

As you will see from his/her comment enclosed below, the advisor is in favour of the work stating the interest and value of your results and s/he is supportive of publication at The EMBO Journal, pending minor revision.

We are thus pleased to inform you that we can offer to swiftly move forward towards acceptance of this work at The EMBO Journal, pending minor revision of the following remaining issues, which need to be adjusted in a re-submitted version.

Based on the positive views of the advisor together with our own assessment, we decided to proceed with publication of your work at The EMBO Journal pending the above points in a time frame of two weeks.

Once we have received the revised version, we should then be able to swiftly complete formal acceptance and expedited production of the manuscript.

We also need you to take care of a number of minor issues related to formatting and data annotation, which I will share shortly in a separate message, together with additional changes and requests by our production team and my colleague H. Sonntag (CC'ed) for Source Data provision.

Please submit a revised version of the manuscript using the link enclosed below, addressing the advisor's comments.

As you might have seen on our web page, every paper at the EMBO Journal now includes a 'Synopsis', displayed on the html and freely accessible to all readers. The synopsis includes a 'model' figure as well as 2-5 one-short-sentence bullet points that summarize the article. I would appreciate if you could provide this figure and the bullet points.

Thank you again for giving us the chance to consider your manuscript for The EMBO Journal, I look forward to hearing from you and receiving your final revised version of the manuscript.

Kind regards,

Daniel Klimmeck

Daniel Klimmeck PhD
Senior Editor
The EMBO Journal.

EMBOJ-2024-117686-T, arbitrating advisor's comment:

I have now carefully reviewed the manuscript and the two rounds of revisions. I don't see any reasons why this would not be published as it is or with minor modifications (perhaps emphasizing more on the limitation of this mouse model) in EMBO.

This study provides valuable data linking cell states (notably the astrocyte-like state) to altered metabolic programming. The authors have done a commendable job of adding mechanistic studies and addressing several key issues raised by reviewers in the first round. While some concerns brought up by reviewers 2 and 3 are valid, I do agree that several points raised by reviewer 2 are indeed beyond the scope of this study (for example, the role of Notch in early vs late stages of gliomagenesis).

Some of my perceived limitations:

Lack of correlation between Notch signaling and RSL3 sensitivity in vitro: GBM cells cultured as neurospheres should retain plasticity. Yet, it is surprising that activating Notch in these cells was insufficient to guide the TS543 neurospheres toward the AC-like phenotype. This somewhat dampens my enthusiasm regarding the relevance of their N11C mouse model. That said, all in vivo models of gliomagenesis suffer from limitations.

The notion that Notch activation at tumor inception forces an astrocytic developmental trajectory, but Notch activation later in tumor development is insufficient to induce similar cell states, is questionable, given what is provided. I feel that a distinction between inherent (for ex. Notch-driven or AC-like related) vs. acquired quiescence (for ex, induced by radiation therapy or other stress-inducing insults) is lacking.

The authors addressed the minor editorial issues.

Dear Dr Canoll,

Thank you for submitting the revised version of your manuscript. I have now evaluated your amended manuscript and concluded that the remaining minor concerns have been sufficiently addressed.

I am thus pleased to inform you that your manuscript has been accepted for publication in the EMBO Journal.

On a different note, I would like to alert you that EMBO Press offers a format for a video-synopsis of work published with us, which essentially is a short, author-generated film explaining the core findings in hand drawings, and, as we believe, can be very useful to increase visibility of the work. Please see the following link for representative examples and their integration into the article web page:

<https://www.embopress.org/doi/full/10.15252/emj.2019103932>

Finally, we have noted that the submitted version of your article is also posted on the preprint platform bioRxiv. We would appreciate if you could alert bioRxiv on the acceptance of this manuscript at The EMBO Journal in order to allow for an update of the entry status. Thank you in advance!

Best regards,

Daniel Klimmeck

Daniel Klimmeck, PhD
Senior Editor
The EMBO Journal
EMBO
Postfach 1022-40
Meyerohofstrasse 1
D-69117 Heidelberg
contact@embojournal.org
Submit at: <http://emboj.msubmit.net>